

# Characteristics and evolution of bedrock permafrost in the Sisimiut mountain area, West Greenland

Marco Marcer[1,2], Pierre-Allain Duvillard[3,4], Soňa Tomaškovičová[1], Steffen Ringsø Nielsen[2,5], André Revil[4], and Thomas Ingeman-Nielsen[1]

[1]DTU Sustain,Bygningstorvet, Bygning 115, 2800 Kgs. Lyngby, Denmark
[2]Arctic DTU, Siimuup Aqqutaa 32, B-1280, 3911 Sisimiut, Greenland
[3]Styx4D, 12 Allée du lac de Garde, Le Bourget du lac, France
[4]EDYTEM, Université Savoie Mont-Blanc, CNRS (UMR 5204), 73370 Le Bourget du Lac, France
[5]KTI Råstofskolen - Greenland School of Minerals and Petroleum, Adammip Aqq. 2, 3911 Sisimiut, Greenland

**Correspondence:** Marco Marcer (marcma@byg.dtu.dk)

**Abstract.** Bedrock permafrost is a feature of cold mountain ranges that was found responsible for the increase of rock fall and landslide activity in several regions across the globe. In Greenland, bedrock permafrost has received so far little attention from the scientific community, despite mountains are a predominant feature on the ice-free coastline and landslide activity is significant. With this study, we aim to move a first step towards the characterization of bedrock permafrost in Greenland.
Our study area covers 100 km$^2$ of mountain terrain around the town of Sisimiut – 68° N on the West Coast. We first acquire surface ground temperature data from 2020-2021 to model bedrock surface temperatures time series from weather forcing on the period 1850 - 2022. Using a topographical downscaling method based on digital elevation model, we then create climatic boundary conditions for 1D and 2D heat transfer numerical simulations at the landscape level. In this way we obtain permafrost distribution maps and ad-hoc simulations for complex topographies. Our results are validated by comparison with temperature
data from two lowland boreholes (100 m depth) and geophysical data describing freezing/unfreezing conditions across a mid-elevation mountain ridge. Finally, we use regional carbon pathway scenarios 2.6 and 8.5 to evaluate future evolution of ground temperatures to 2100. Our results indicate a sporadic permafrost distribution up to roughly 400 m.a.s.l., while future scenarios suggest a decline of deep frozen bodies up to 800 m.a.s.l., i.e. the highest summits in the area.

## 1 Introduction

Mountain permafrost is a typical cryospheric feature of cold mountain regions, defined as perennially frozen ground occurring in complex topography. In the past decades, the scientific community focused on understanding the role of mountain permafrost in slope stability, in order to determine the rise of new natural hazards linked to the changing climate (Haeberli et al., 2010). Several studies proved a relationship between ground mechanical properties and permafrost characteristics, indicating that an increase in rock temperature – involving increasing water content and decreasing ice content - generally causes a deterioration
in mechanical stability. This deterioration consists of a decrease in compression and shear resistance (Davies et al., 2001;



Yamamoto and Springman, 2019), as well as lower resistance to fracture initiation (Krautblatter et al., 2013), resulting in the breaking of rock bridges (Mamot et al., 2020).

Several field studies describe a significant correlation between warming climate and an increase in slope instability, observed as rockfall frequency (Ravanel and Deline, 2011; Gallach et al., 2020), large rockslide occurrence (Patton et al., 2019; Guerin

et al., 2020; Frauenfelder et al., 2018; Walter et al., 2020), high elevation infrastructure destabilization (Duvillard et al., 2019) and debris permafrost creep rate increase (Marcer et al., 2021). Overall, the scientific community agrees on the effects of climate change on slope instability in mountain permafrost, and several countries started comprehensive programs to monitor this phenomena as a basis for risk assessment.

In Greenland, the scientific community still does not have a precise quantification of mountain permafrost distribution.

Available models are based on numerical simulations at kilometer scale, and either not calibrated, or calibrated using data representative for sedimentary terrain only. Indeed, permafrost temperature data from Greenland are limited to few low-land boreholes that are not representative for higher elevation and complex terrain (Obu et al., 2019). This knowledge gap challenges our understanding of mountain hazards and their evolution, preventing a reliable hazard assessment that Greenland urgently requires, as most of the population resides in proximity of mountain areas. As a result, the scientific community does not have a

clear idea about the role of permafrost warming in recent events, such as the landslide-triggered tsunami that tragically ravaged the settlement of Nugaatsiaq (Svennevig, 2019; Svennevig et al., 2020) or the debris flows that hit Siorapaluk (Walls et al., 2020).

The first step to gather information on permafrost conditions in mountain terrain is based on the standard approach developed in Switzerland in the early 2000's (Gruber et al., 2004) relying on a network of permanent surface temperature loggers.

These data are then used for transient modelling of ground temperatures across 1D profiles in relation with depth (Westermann et al., 2016), or in 2D (Magnin et al., 2017a) and 3D complex geometries (Noetzle et al., 2007). Several studies model ground temperatures using numerical approaches, as TEBAL (Stocker-Mittaz et al., 2002; Gruber et al., 2004) and CryoGrid (Westermann et al., 2016; Czekirda et al., 2019; Gisnås et al., 2017; Myhra et al., 2017). Both models have a numerical approach to the evaluation of the Surface Energy Balance (SEB), i.e. the transfer from weather parameters to surface energy flux as upper

boundary condition for the heat transfer module. Other studies have handled the SEB problem using an empirical approach based on correlating weather data and measured Surface ground temperatures (Magnin et al., 2017a; Etzemüller et al., 2021; Rico et al., 2021; Legay et al., 2021). The advantage of this approach is that it does not require complex climatic input which may not be always available, still reaching good performances.

Using this latter modeling approach, we aim to provide a first quantification of mountain permafrost conditions in Greenland

by focusing in the Sisimiut area (68° N on the west coast). Instead of accurate weather data, we dispose of a large number of Ground Surface Temperature (GST) measurements, as in fall 2020 we installed 28 surface temperature loggers in the area. Using these data, we train a statistical model to evaluate the correlation between weather variables and measured GST. Weather data are downscaled using a basic elevation-gradient and solar exposure approach based on a Digital Elevation Model. Once the statistical model is trained, we use it to predict time series of GST at the landscape scale and longer temporal frames,

allowing creating the boundary conditions for a heat transfer model. In this study, we use COMSOL heat transfer module,





connected to Matlab using LiveLink. We test our model for 1D simulation, which we compare to temperature data obtained by two 100m deep boreholes drilled in the area in 2019 and 2021. To obtain field data on ground temperature at high elevation, we used the approach proposed by Duvillard et al. (2020) consisting in geophysical surveys and calibration of resistivity - temperature dependencies in laboratory experiments. This methodology allows to obtain a bidimensional transect of ground

freezing conditions at a given survey date.

## 2 Study site

Our study site is located around the West Greenland town of Sisimiut, which is located on the coastline of the widest non-glaciated area in West Greenland, about 200 km from the Greenland Ice sheet (see figure 1). Sisimiut is the second largest city in Greenland, counting 5582 inhabitants in 2020 and experiencing a rapid development. The landscape is characterized by

narrow fjords, alpine summits and isolated coastal glaciers. The dominant lithology is amphibolitic gneiss (Ljungdahl, 1967). The mountains of the region typically have pyramid-shaped summits and steep rockwalls. The Sisimiut town is surrounded by two main mountain ridges: the Nasaasaaq – Appillorsuaq ridge to the south, summiting at 784 m.a.s.l., and the Palasip Qaqqa– Sammisoq ridge to the North, summiting at 605 m.a.s.l. Climatically, Sisimiut is located in the low arctic oceanic area (Jensen 1999), with a mean annual temperature of -1.8 °C for the period 2000 - 2020 (Cappelen and Jensen, 2021) and mean annual

total precipitation of 382 mm at sea level (Period 1961 – 1990). The warmest month is July (6.3 °C on average), while the coldest is March (-14.0 °C). Mean annual air temperature increased by 2.2 °C from the period 1980 – 2000 to the period 2000 – 2020, while precipitation remained substantially unchanged. This temperature increase is believed to have caused significant glacial retreat in the coastal glaciers in the area, which lost about a fourth of their volume in the past three decades (Marcer et al., 2017) . The coastal region around Sisimiut is situated in the sporadic permafrost zone (Obu et al., 2019; Biskaborn et al.,

2019). In the area are observable some morphologically active rock glaciers reaching sea level elevation.

## 3 Methods

Our study is based on field data acquisition of ground temperatures, which are then used to calibrate a model, used to describe present and future permafrost conditions in the study area.

### 3.1 Field data

#### 3.1.1 Ground temperature monitoring

As part of this study, we established a ground temperature monitoring network comprising both ground surface temperature (GST) monitoring and borehole temperature (BT) monitoring to a depth of 100 m.b.g.s. All sensors used for the temperature data acquisition were custom zero-point calibrated using a Fluke 7320 compact bath with a manufacturer specified temperature stability and uniformity better than 0.01°C. The bath temperature was measured using a Fluke PRT 5610 secondary standard

temperature probe, and each sensor was immersed in the bath for 40 minutes while logging every 30 seconds. After the sensor



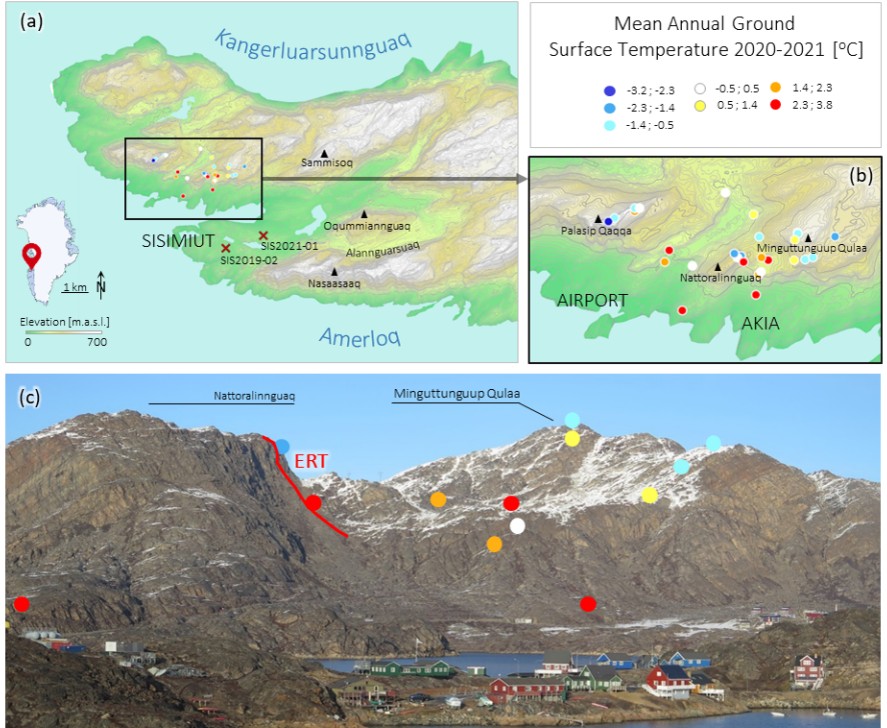

**Figure 1.** Study site summary. In panel (a) is shown the map of the entire study area, with location of deep boreholes SIS2019-02 and SIS2021-01. On panel (b) is shown the detail of the Nattoralinnguaq area, where most of the ground surface sensors are installed. On panel (c) is shown the south face of Nattoralinnguaq and Miguttunguup Qulaa (picture taken from Sisimiut in October 2020) with surface loggers location and ERT location. Loggers are colored based on their measured mean annual surface ground temperature (fall 2020 to fall 2021). Elevation data belong to the Arctic DEM (Porter, 2018).

temperature stabilized, the sensor offset was calculated as $\Delta T = (\sum_{(i=1)}^{n}[T_{ref,i} - T_{s,i}])/n$, where $T_{s,i}$ [°C] is the $i^{th}$ sensor temperature measurement in the calibration period, $T_{ref,i}$ [°C] is the corresponding bath temperature measured by the PRT sensor at the same time, and $\Delta T$ [°C] is the average calculated sensor offset, which was applied as a correction to each field temperature measurement collected by that sensor. The total combined uncertainty of the calibration bath reference temperature is better than 0.02 °C (k=2).

### 3.1.2 Ground Surface Temperature

We established a GST monitoring network consisting of 28 individual monitoring locations, covering as evenly as possible the range of aspects, elevations and slopes at the study site, and data were acquired for one year, from fall 2020 to fall 2021. The technical information about loggers used are summarized in table 1. Both DS1922L and M5-Rock are sensors widely used in permafrost studies and the community has previous experience in their strength and weaknesses (Gruber et al., 2004;





| Nb | Brand | Type | Sensor | Resolution | Logging interval | Installation depth | Terrain |
|----|-------|------|--------|-----------|-----------------|-------------------|---------|
| 28 | Maxim integrated | iButton | DS1922L | 0.06 °C | 4 hrs | 5cm | Bedrock (13), Steep bedrock (4) Soil (11) |
| 5 | Geoprecision | M5-Rock | TNode | 0.01 °C | 1 hr | 30cm | Steep bedrock |
| 4 | Geoprecision | M5-Rock | 2xTNode | 0.01 °C | 1 hr | 30cm, 90 cm | Steep bedrock |

**Table 1.** Summary of sensors and their specifications used to measure GST in the study area form fall 2020 to fall 2021.

Gubler et al., 2011). This combination of loggers is common as a trade-off between equipment costs and data quantity/quality. The DS1922L are low-cost digital chip temperature sensors, which provide a reasonable temperature resolution of 0.0625°C. However, since they are primarily designed for indoor use, they are prone to failure when used in harsh environment and about 5-10% of the deployed loggers can be expected to fail (Gubler et al., 2011). Protecting the logger with plastic film (as we did) helps to reduce failures. The M5-Rock on the other hand are more expensive and reliable loggers, which provide a better quality measure. Geoprecision loggers can be accessed by radio, allowing remote download of data (10-20 m range), which becomes handy in steep terrain. Given these characteristics, we decided to install nine M5-Rock loggers in steep bedrock, as these data are strategically more relevant in the context of the study. The iButtons (28 loggers in total), were installed in other more accessible conditions, such as flat bedrock (13), soil (11) and easy-access rockwalls (4) (See figure 1).

### 3.1.3 Deep Ground Temperature

Two 100 m deep boreholes, SIS2019-02 and SIS2021-01, were drilled in bedrock outcrops in relatively flat terrain at 50 and 70 m.a.s.l. (see Figure 1). The boreholes were drilled using a Sandvik DE130 compact core drill owned and operated by the Greenland School of Minerals and Petroleum, with wireline NQ drilling tools (outer diameter: 70 mm). The holes were installed with a 100 m long PE casing (outer diameter 32 mm, inner diameter 26 mm), closed at the bottom with a heavy duty heat shrink end cap with heat activated glue. Borehole SIS2019-02 is logged manually at 2 m depth intervals, using a HOBO U12-015-02 logger (zero-point calibrated in a triple-point of water cell). The logger uses a 10 sec sampling interval and rests at each depth for two minutes. In the post processing, temperatures are averaged only over the last minute to obtain the temperature at a particular depth, thereby ensuring the sensor has equilibrated to the new temperature. The borehole SIS2021-01 is equipped with a GeoPrecision thermistor string with 28 sensors (TNode, digital chip with 0.01 °C resolution). The upper-most sensor is located at 0.1 m.b.g.s., the lower-most at 99 m m.b.g.s. The sensor spacing progressively increases with depth from 0.4 m in the top to 10 m at depth, and the logging interval is 1 hr. The two boreholes are drilled in similar conditions of elevation and exposure to solar radiation, while snow conditions are different. SIS2019-01 is located in a drift accumulation area and the snow depth can reach 2 m, while SIS2021-01 is on a wind-exposed hill, which ensures snow-free conditions most of the winter.





### 3.1.4 Geophysical profiles


The geophysical investigations were conducted on Nattoralinnguaq (353 m a.s.l). The summit presents a steep and rocky South face about 100 m high with a debris slope underneath and a more gentle North face characterized by small vegetation patches and some short steeper sections. This summit was chosen due to its proximity to town infrastructures, as well as for easy accessibility thanks to a path that leads to a popular viewpoint at the summit.

The geophysical investigations were conducted early October 2020 using Electrical Resistivity Tomography. Five 100-m-long cables (500-m-long profile) and a total of 100 electrodes (5-m-spacing) were connected to a resistivity meter (GuidelineGeo Terrameter LS2 powered by a 12 V external battery). We used 10 mm x 100 mm stainless steel electrodes, inserted in pre-drilled holes with a paste of salty bentonite to improve electrical grounding and prevent freezing (Krautblatter and Hauck, 2007; Magnin et al., 2015b). The Wenner configuration was used because of its best signal-to-noise ratio in complex

environments due to its particular electrode configuration since the voltage electrodes MN are located in-between the current electrodes AB (Dahlin and Zhou, 2004; Kneisel, 2006). Topography along the profile was obtained thanks to a handheld GPS and altitude were obtained using a metric digital elevation model (DEM).

We cleaned 28% of the data point acquired before the inversion (734 points acquired, 528 inverted) base and the standard deviation and the pseudo section. The data were inverted with the RES2DINV-4.8.10 software using a smoothness-constrained

least-squares method and the standard Gauss–Newton method (Loke and Barker, 1996). The inversion was stopped at the third iteration when the convergence criterion was reached.

### 3.1.5 Laboratory experiences and temperature analysis

In addition to the field measurements, we performed a laboratory electrical conductivity experiment on two a rock sample collected in the field from an outcrop of the rockface and the middle of the south north face. Three granite cubic core samples

(sample G-RF, G-LR and G-DA) was characterized by a porosity of $\Phi = 0.046$ for G-RF, $\Phi = 0.032$ for G-LR and $\Phi = 0.012$ for G-DA. Before performing the laboratory measurements, the samples was dried during 24 hr then saturated under vacuum with degassed water from melted snow taken in the field. The samples was left several weeks in the solution to reach chemical equilibrium before performing the laboratory measurements. The water conductivity at 25 °C and at equilibrium was 9.56·10-3 Sm$^{-1}$ for G-RF and 1,157 10-3 Sm$^{-1}$ for G-LR and G-DA. The sample holder was placed in a heat-resistant insulating bag

and placed in a freezer for seven days at -22°C. Measurements were then made during the temperature rise. The (in-phase) conductivity measurements shown here are obtain at a frequency of 1 Hz. We moved the freezing point temperature $T_F = 0$°C based on direct observations on instrumented boreholes for G-RF. The measurements with $T_F = 2$°C reflect the fact that the measurements were made only in the upward direction of the temperatures and not in the downward direction. These analyses define the relation between resistivity collected in the field and freeze-thaw conditions of the ERT transect.





| Dataset | Text reference | Period available | Period used | Variables | Type | Location |
|---|---|---|---|---|---|---|
| Cappelen et al 2021a | a | 1784-2021 | 1850 – 1958* | Air temperature | Weather station | Nuuk , Ilulissat |
| Cappelen et al 2021b | b | 1958 - 2021 | 1958 – 1979* | Air temperature | Weather station | Sisimiut |
| a and b, merged | c | 1784 – 2021 | 1850 – 1979 | Air temperature, solar radiation | Ad hoc model | Sisimiut |
| Herbasch et al 2019 | d | 1979 – present | 1979 – 2022 | Air temperature, solar radiation, cloud cover, dew point, wind speed and direction, total precipitation | Reanalysis | Global, Gridded 0.5 degs |
| Hofer et al 2020, RCP2.6, | e | 2006 – 2100 | 2022 – 2100 | Air temperature, solar radiation | Model | Global, Gridded 0.5 degs |
| Hofer et al 2020, RCP8.5 | f | 2006 – 2100 | 2022 – 2100 | Air temperature, solar radiation | Model | Global, Gridded 0.5 degs |

**Table 2.** Summary of climatic databases used to cover the investigation period (1850 -2100). Datasets a,b (*) are used to create dataset c by calibrating an ad hoc model. Therefore, only datasets c,d,e,f are directly used for GST models calibration.

## 3.2 Modeling

Our modeling approach is based on a mixed statistical-numerical methodology, which is conceptually similar to the study developed by Magnin et al. (2017a) and the modelling section in Etzemüller et al. (2021). The methodology consists in evaluating Ground Surface Temperature (GST) time series with an empirical approach, which are then used as upper boundary conditions for a heat transfer numerical model. This modelling methodology can be divided in a three-steps workflow: (i) acquisition of climatic forcing data and downscaling , (ii) statistical modeling and prediction of GST data and (iii) numerical modeling of heat transfer in bedrock.

### 3.2.1 Forcing data and downscaling

The weather input data were retrieved form different sources covering different periods - summarized in table 2. Concerning datasets a and b, since only data from Nuuk (300 km South) and Ilulissat (250 km North) are available prior to 1958, we evaluated the regression between datasets a and b over the overlap period 1958-2020. The regression predicts air temperature in Sisimiut using dataset a for the period 1850-1958, creating air temperature for dataset c. Since datasets a and b did not have measurement of solar radiation, we assigned them a synthetic estimation equal to the average year over the period 1979-2022 (dataset d). Dataset d was downloaded by the Copernicus database, and we selected the standard set of predictor variables used by CryoGrid SEB module. In order to understand future evolution of permafrost in the area, we simulated future GST forced by the Norwegian Erath System Model version 1 (NorESM1) global circulation model, using Representative Concertation Pathway (RCP) 2.6, and 8.5 for 2006-2100 (Bentsen et al., 2013). We chose the NorESM1 model as chosen by several authors in Greenland for cryosphere evolution modelling (Colgan et al., 2016; Hofer et al., 2020) thanks to his good performance in the region (Fettweis et al., 2011). The RCP 2.6 is the NoreESM1 outcomes for scenarios of declining emissions since 2020 (optimistic scenario, dataset e), while the RCP 8.5 is simulated with unregulated emissions increasing at a rate compatible to the present-day industrial development (pessimistic scenario, dataset f).





Air temperature and solar radiation are downscaled at any location in our study area using a topographical approach. The downscaling was dependent on three parameters: elevation, potential incoming solar radiation (PISR) and snow cover probability. Elevation is used to downscale air temperature by applying a constant lapse rate of 0.47°C/100 m, measured on the study area by the MAGST monitoring network by detrending the data for slope aspect and snow cover. The elevation data are obtained by the Arctic digital elevation model (DEM) at 10m resolution (Porter, 2018). PISR is used to downscale the solar radiation forcing by using the ratio between Potential Incoming Solar Radiation at the logger location and the PISR at the ERA5 reference cell. The PISR map is evaluated using SAGA by applying the PISR module to the DEM (Conrad et al., 2015).

Snow influence is evaluated not by directly downscaling a forcing parameter, but rather the GST timeseries resulting from the GST model (see next section). This is done by applying a constant offset: $GSTs_i(z_i, PISR_i) = GST_i(z_i, PISR_i) + dTs * SnowP_i$; where $GST_i$ is the GST depending on the local elevation z and PISR, used to downscale air temperature and solar radiation. dTs is the thermal offset due to snow cover, which we evaluated by comparing the mean annual GST of sensors that were/were not snow covered during the entire winter 2020-2021. SnowPi is the local probability of snow cover presence/absence, varying from zero (absence of snow cover) to 1 (presence of long lasting snow cover). We created the SnowP map by training a neural network classifier with a categorical variable describing presence/absence of snow at specific locations, and topographical data describing slope angle and curvature (planar, longitudinal and profile - obtained with the morphometric features module in SAGA, Wood (2009)) at those locations. This dataset is created by interpretation of landscape pictures taken during winter in the study area and assigning snow/no snow areas on a GIS. The classifier provides a probability of snow cover for a given set of curvature and slope, creating a map snow cover probability SnowP (figure 2).

### 3.2.2 Statistical modeling of GST

In this step, we model the relationship between forcing and GST data using a data-driven approach. Previous studies used an offset-based approach based on the evaluation of a constant thermal offset between air temperature and GST (Magnin et al., 2017a; Etzemüller et al., 2021). In our study, we use a conceptually identical approach, based on the following hypothesis: the GST can be predicted by an empirical model trained using available forcing variables that dominate GST distribution on steep bedrock (air temperature, solar radiation, wind characteristics, precipitation and humidity for dataset d, and air temperature and solar radiation for datasets c, e and f). To do so, we aggregate each GST measurement to the forcing data that occurred during that acquisition time step, downscaled at the logger location. This creates a database of Nx1 targets and NxM predictors, where N is the number of available GST data and M is the number of climatic predictors used. We then split this database into training and validation sets, following a pseudo-randomized cross validation approach, as we randomly exclude entire loggers time series from training. This allows observe the model's performance in predicting GST at locations not used in the training process. We use a multinomial linear model, trained using the Matlab function fitlm. Since all datasets cover the period 2020-2021, i.e. when we have our GST measurements, this process is done to cover the whole period 1850 - 2100, creating four independent models, each trained on a specific dataset.



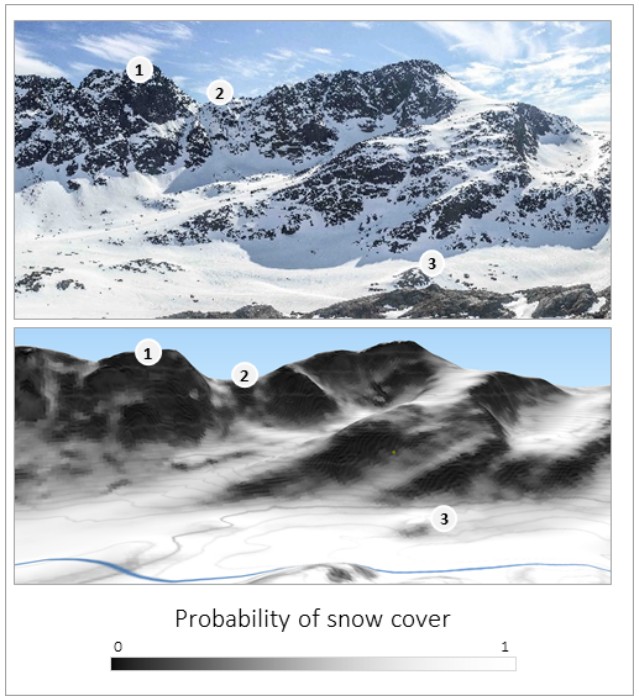

**Figure 2.** Comparison of the SnowP map (3D visualization) with a field picture of the North face of Nasaasaaq taken in May 2021 from the summit of Oqummiannguaq. The SnowP map correctly identifies snow free steep terrain (1), steep chutes that are mostly snow covered (2) and gentle terrain snow free because of wind erosion (3).

### 3.2.3  Numerical modeling of heat transfer

Heat transfer is simulated in 1D conditions for model calibration and large scale mapping, while 2D simulations are restricted to areas of interest for more detailed analysis. The heat transfer process is modelled using the "heat transfer in porous media" module in COMSOL, which assumes valid the local thermal equilibrium hypothesis. The model accounts for three materials: solid matrix, fluid and solid with phase change – all materials parameters are summarized in table 3. The fluid phase is the default COMSOL "water" material, to which we assigned a phase change to ice at 273.15 K and transition interval to ice of 5K, according to Noetzli and Gruber (2009). The porous matrix material is assigned as the default material "granite" K = 2.9$[W(m*K)^{-1}]$ and Cp = 850$[J(kg*K)^{-1}]$. For the 1D model we entered a custom function to describe the matrix density, which was evaluated from the cores extracted form SIS2021-01, providing an empirical function of depth increasing from 2600 kgm$^{-3}$ in the subsurface to 3000 kgm$^{-3}$ at 20 m depth. We attributed the average density of 3000 kgm$^{-3}$ to the 2D models.

The numerical simulation consists of four successive studies: a stationary study for initial conditions (mean conditions for 1850-1860, forcing dataset c), a transient study 1850-1979 (dataset c), a transient study 1979-2022 (forcing dataset d) and a transient study 2022-2100, depending on the scenario chosen (RCP 2.6 and RCP 8.5 - forcing datasets e,f). For each time period (1850-1979; 1979-2022 and 2022-2100), the corresponding GST model and forcing time series are stored in COMSOL



as analytical functions using local elevation, PISR, SnowP and time as parameters that can be modified by the user to reproduce ground temperatures in different settings. Therefore, by entering custom local topographical conditions, the COMSOL model will produce downscaled GST time series and use them as upper boundary conditions for the heat transfer module. As lower

boundary condition, we used the constant geothermal heat flux, which we also evaluated from SIS2021-01 data. Our data indicate a temperature gradient of $0.015°\mathrm{Cm}^{-1}$, which, considering a thermal conductivity of 2.9 $\mathrm{W(m*K)}^{-1}$, gives a constant geothermal heat flux of 0.045 $\mathrm{Wm}^{-2}$.

### 3.2.4 Model calibration

The numerical model was calibrated for two parameters: matrix porosity and initial conditions in 1850. The calibration was

225 carried by simulating conditions in SIS2021-01, which was modelled using a 1D geometry of a 100m column. Both parameters are calibrated carrying the simulation from 1850 to 2022 and comparing model results to data available from SIS2021-01, i.e. from August 2021 to April 2022. We evaluated the GST at the location of SIS2021-01 using the downscaled climatic variables over the period 1850-2022. For downscaling, we evaluated elevation, PISR and snow cover probability on the respective raster maps (elevation = 77 masl, PISR = 790 $\mathrm{kWhm}^{-2}$, SnowP = 0.3). We tested different porosity values according to previous

studies findings, i.e. porosity ranging from 0.01 (Rico et al., 2021) to 0.05 (Magnin et al., 2017a). As initial conditions, we evaluated the temperature profile of the stationary solution of the 1D model forced by the average GST over the period 1850-1860. We then added a positive ground temperature offset as parameter to account the fact that temperatures in 1850 – 1860 (at the Little Age peak) were lower than the previous period, and deep ground temperatures were likely higher than what modelled by our stationary model . The optimization of the two parameters targeted the best fit between measured and modelled deep

ground temperatures (below 20m depth), as well as active layer thickness.

### 3.2.5 Mapping ground temperatures

The ground temperature map was computed by evaluating the calibrated 1D model for each set of topographical condition in our study area. This process is handled by Matlab's Livelink, which runs the routine through each gridcell in the DEM, PISR and SnowP maps, and passes the local topographical parameters to the 1D COMSOL model. The COMSOL studies are run

from Matlab command and the export is stored in a text file which contains the evolution of temperature in the column over time for each study. Matlab then imports the output at each loops and stores the Mean Ground Temperature at 20m depth in the period 2012-2022 (MGT20) and assigns it to the corresponding raster cell, creating the MGT20 map.

### 3.2.6 2D Models

The 2D model computed for the ERT location and the Nasaasaaq summit, as they present complex topography. Both areas

have interests as we would like to compare the ERT data to our model, as well as observing permafrost evolution at Nasaasaaq - the tallest mountain in the study area. For each location we imported the elevation profile z in COMSOL and converted to a solid using the parametric function option. Also PISR and SnowP are imported as interpolation functions whose coordinates



are consistent to the reference system of the solid representing the terrain. In this way, we can provide the GST model as an interpolation function over the spatial variable (x) and temporal variable (t): $GST(x,t) = f(z(x), PISR(x), t) + dT * SnowP(x)$, being f the linear GST models for each forcing dataset. As lower boundary condition, we used the geothermal heat flux of 0.045 Wm$^{-2}$ evaluated form SIS2021-01, while we implemented zero-flux conditions on the lateral boundaries.

## 4 Results

### 4.1 Sensors calibration

he sensors calibration revealed that DS1922-L have an average absolute temperature offset of 0.20°C, reaching a maximum of 0.71°C. The standard deviation of the measurement is 0.02°C. M5-Rock offsets are on average 0.04°C, reaching a maximum of 0.10°C, while the standard deviation of the measurement is below 0.01°C. Sensors' offsets are applied as correction to each corresponding logger measurements.

### 4.2 Measured ground temperatures

### 4.3 Boreholes

Both boreholes show deep ground temperatures close to 0°C. SIS2019-02 was measured three times since its installations and presents an active layer of about 10m, and a minimum of temperature of +0.3°C at 30m depth, reaching +1.0°C at 100 m. SIS2021-01 has too short record to define precisely active layer depth, which seems to be around 10 m deep for summer 2021. Below 10m depth, ground temperatures are lower than SIS2019-02, reaching -0.2°C at 30 m and +0.3°C at 100 m. This indicates the presence of permafrost, which reaches 70 m depth.

#### 4.3.1 Ground surface temperatures

All loggers here reported run for one full year, from fall 2020 to fall 2021. Among the 27 loggers, fifteen presented snow free data, seven present thick snow cover and six present intermediate characteristics (Fig 3). Concerning snow free loggers, the data show mean annual ground surface temperature (MAGST) varying from +3.5 °C for south facing rockwall at sea level to +1.2 C at 460 masl. Aspect causes a MAGST offset of 2.2 °C from north faces to south faces at similar elevations. When snow covers the loggers, these quantifications do not hold. On average, based on our data, the presence of snow cover causes an offset on the MAGST of +1.58 °C ± 0.41 °C when other conditions do not change (R$^2$ = 0.81). This value - +1.58 °C- was assigned to dTs for GST downscaling.

### 4.4 Geophysical survey

Electrical conductivity tomograms acquired show a vertical distribution of the conductivities with rather low conductivity values ($< 10^{-3.5}$ Sm$^{-1}$) below the N and S face and higher values inside the mountain ($> 10^{-4.4}$ Sm$^{-1}$ - Fig 4). The petrophysical



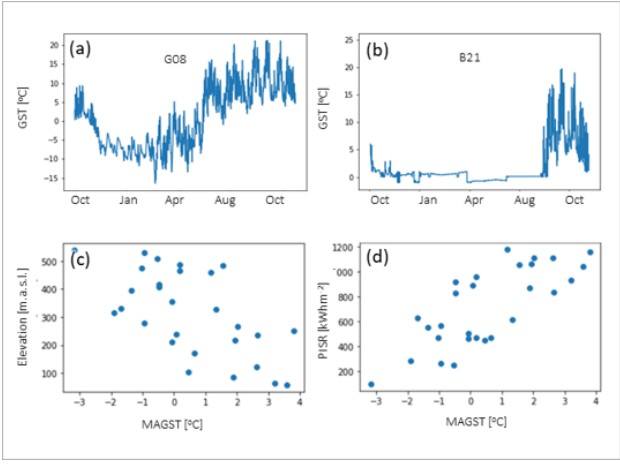

**Figure 3.** Summary of GST recorded by the loggers during 2020-2021. On top, examples of GST time series are for a snow free logger (a) and snow covered logger (b). On bottom, the MAGST in relation to topographical predictors Elevation (c) and Potential Incoming Solar Radiation (d).

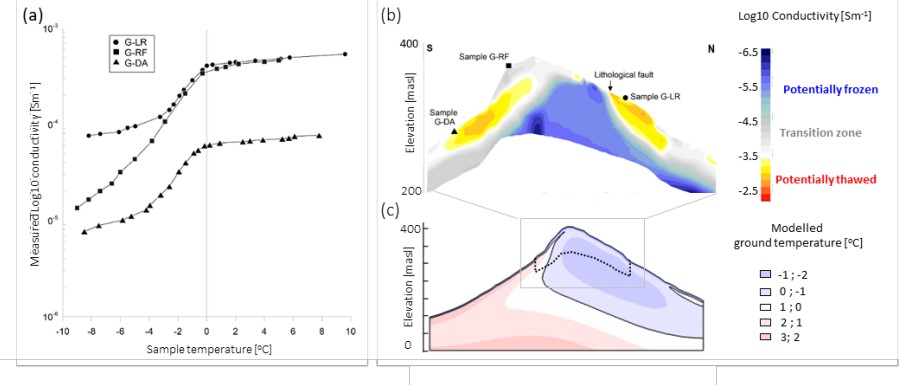

**Figure 4.** Comparison between ERT and 2D model. In panel (a), the inverted conductivity model obtained by ERT on the field. On panel a, the results of the petrophysical analysis, showing the conductivity range for the transition between frozen and unfrozen state. On panel (b) the combination of the ERT inversion and petrophysical analysis, showing in blue colors expected frozen areas, while on yellow colors the expected unfrozen areas. On panel (c), the resulting 2D model of the ridge where the ERT line was conducted.

analysis suggests a transition zone from frozen to thawed conditions between $10^{-4.4}$ and $10^{-3.5}$ $Sm^{-1}$. This suggests that permafrost presence is restricted inside the mountain and possibly closed the surface in the North face and lower part of the South face. The electrical conductivity anomaly on the north face occurs near a large lithological fault that has an impact on the distribution of permafrost and ground characteristics.





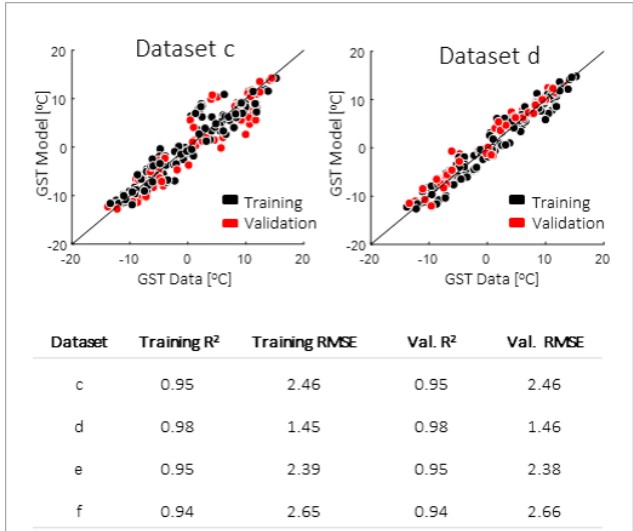

**Figure 5.** GST models summary. On top, examples of model fit for the dataset c (DMI 1850 – 2021) and d (ERA5 1979 – 2022). On bottom, table with training and validation performances for the four different GST models.

## 4.5 Modeling

### 4.5.1 GST Model

GST modeling is done at monthly time steps by aggregating weather forcing and on snow-free loggers data collected on bedrock (13 loggers available), averaged over monthly periods. The training results of the GST models are summarized by period in Figure 4. Considering the dataset d, i.e. the dataset providing most climatic variables we achieve similar performance in both training and validation sets, indicating good generalization power of the model even for high number of predictors. The prediction performance is best for dataset d ($R^2$ = 0.98 and RMSE < 1.6 °C), while it is lower for the other datasets, reaching RMSE 2.46 – 2.66 °C.

### 4.5.2 Heat transfer model

The optimal porosity was achieved for 0.03, while the optimal initial offset was evaluated at +1.8 °C with respect to the average temperature on the period 1850-1860. Using these values, the model reaches a good agreement of the seasonal frost and heat penetration depths at SIS2021-01 for the period August 2021 – April 2022 (figure 5). The difference between model and data is consistently below 0.1 °C for depth below 10 m, while in the active layer the model has disagreement up to 2 °C with the measured data. When tested and compared to SIS2019-02 (z = 55m ; PISR = 690 kWhm$^{-2}$, SnowP = 1), the model produces similar results, indicating errors up to few degrees in the active layer (10 m depth), while for depths below 20m the errors are consistently below 0.15 °C.





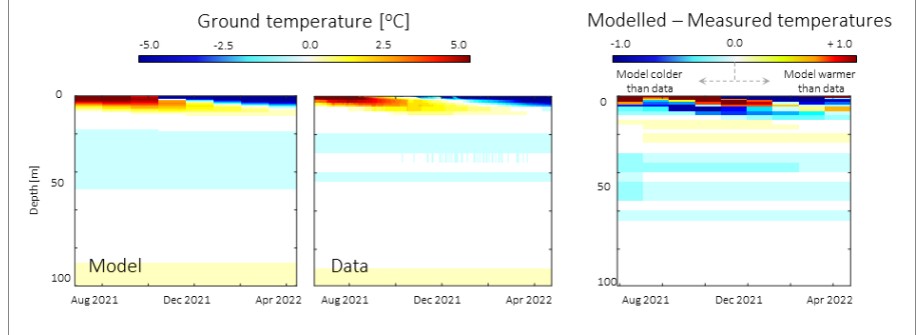

**Figure 6.** Model compared to data at SIS2021. On left, temperature plot for the available measured period of both data and model results. On right, difference between model and measured temperatures. Cold colors indicate that the model is colder than the data, while warm colors indicate vice versa.

### 4.6 Permafrost mapping

The GST model is run to produce boundary conditions for each cell in the DEM, which are used to run the heat transfer module. We obtain a database of monthly-averaged ground temperatures from 1850 to 2022 up to 100 m depth. This database can be used to produce several maps describing ground thermal conditions in the area, here we propose mean ground temperature
(2012-2022) at 20 meters depth (MGT20 – figure 6a, 6b) and summarize it in a polar plot (figure 6c). At sea level, north facing snow free slopes can reach negative MGT20. Negative MGT20 can be found on south slopes starting at 200 m.a.s.l.. Snow cover plays an important role, as snow covered areas on steep south slopes (i.e. chutes), can have positive MGT20 up 450 masl, elevation at which permafrost is continuous. The colder MGT20, which occur on the north faces of the Nasaasaaq peak - 763 m.a.s.l. -, reaching -4.0 °C.

#### 4.6.1 Comparison 2D model - ERT

The 2D model simulation on ERT1, indicates, as of October 2020, the presence of sporadic permafrost on the Nattoralinnguaq summit. The south face is permafrost free, with ground temperatures above zero at 20-40 m depth. The north face on the other hand is permafrozen, reaching temperatures below -1 °C. By comparing the model results with the ERT data (figure 7), we can observe a qualitative agreement between the two datasets, as they both indicate the presence of sporadic permafrost on
the summit. Both datasets indicate a mostly unfrozen south face, and a colder north face. However, the ERT data indicate a large unfrozen section at the extremity of the north face, which is likely due the lithological fault observed on the field at this location.

#### 4.6.2 Permafrost evolution in future scenarios: RCP 2.6 and RCP 8.5

Future scenarios simulations are conducted both at the landscape scale and at SIS2021-01 location (Figure 8a and 8b). The
simulations conducted at SIS2021-01 show that, regardless the scenario used, permafrost conditions will disappear by the end



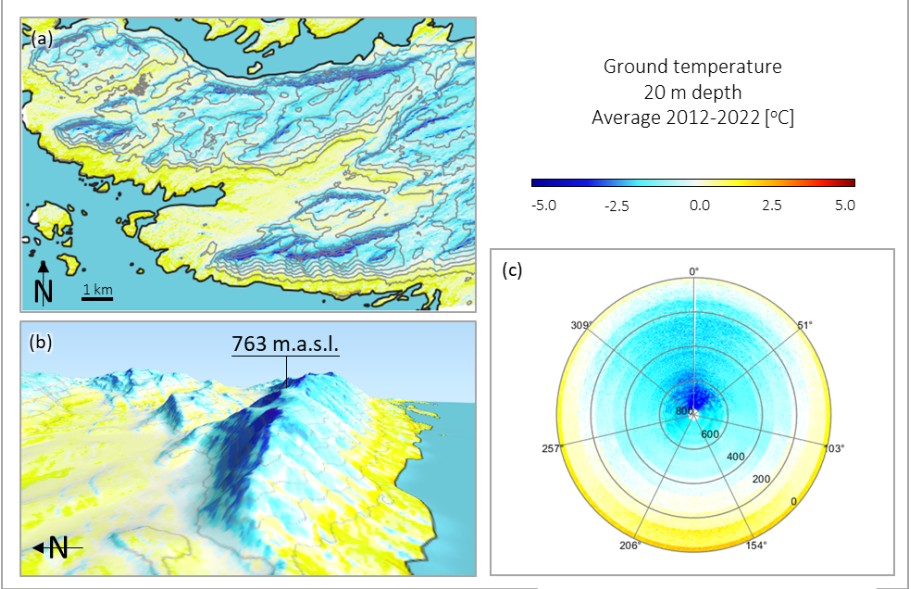

**Figure 7.** Summary of modelled ground temperature distribution in the study site. The parameters used to describe ground temperature here is the average ground temperature at 20 meters depth in the past ten years. On panel a, study site map colored by MGT20. On panel b, 3D view of the Nasaasaaq range from East. On panel c, polar plot of the distribution of MGT20 based on slope aspect (0° is North) and elevation (outer radius is sea level, increasing to 800 masl at the center.

of the XXI century. For scenario 2.6 permafrost seems in phase transition by 2100, being at 0.05 – 0.1 °C between 50 and 70 meters depth. For scenario 8.5 ground temperatures are consistently above 0.3 °C, indicating total permafrost loss. In 2100, ground temperatures at 20-50m depth are about 1 to 2.5 °C higher for the RCP 8.5 compared to RCP 2.6, indicating that, due to thermal inertia of the ground, surface heat is not yet fully propagated at depth by 2100 in this scenario.

At the landscape level, any future scenario causes a significant reduction in the extents of frozen grounds by 2100 (Figure 8c). For the RCP2.6, in 2090-2100 is simulated a slight increase in elevation of the MGT20 isotherm by about 150 m. This causes a widespread loss of permafrost grounds, from 81 km$^2$ (57% of the study area) to 53 km$^2$ (37%). For the scenario RCP8.5 the impact on permafrost is more severe, as permafrozen ground disappears from most of the study area, except for the highest summits and covering 4km$^2$ (3% of the study area) in 2090-2100. The MGT20 isotherm elevation increases above

the 700 masl on south faces, while on north faces, we observe a retreat of the 0 °C isotherm MGT20 up to 500 masl.

A similar result is obtained when evaluating the expected ground temperature evolution in complex terrain (figure 9). Nattoralinnguaq and Nasaasaaq present today sporadic and continuous permafrost respectively. Considering the optimistic scenario (RCP 2.6), the model suggests a significant increase of ground temperatures and a loss of permafrost grounds compared to present day's estimations, corresponding to a slight increase in elevation of the permafrost margins. Scenario RCP 8.5 de-

lineates a situation where almost all permafrost is relict, i.e. below the reach of seasonal frost, except for the north face of the Nasaasaaq summit

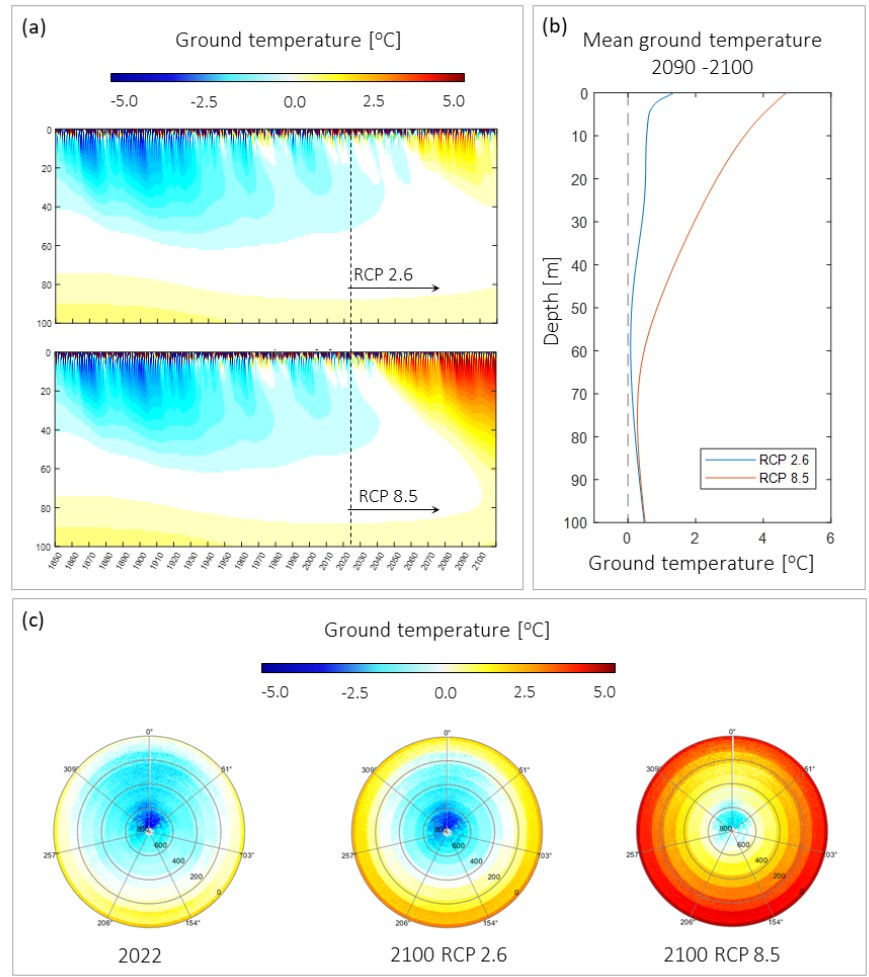

**Figure 8.** Summary of future scenarios from 1D models. Panel a, ground temperature evolution at SIS2021-01. On panel b, comparison of mean ground temperature profiles (2090-2100) between optimistic and pessimistic scenarios. On panel c, landscape distribution of MGT20 as polar plots (elevation as radius, angle as slope aspect – 0° points North) in 2022 and the two scenarios.

## 5 Discussion

### 5.0.1 Model uncertainties and evaluation

Despite evaluating GST using an empirical approach has been already used by several studies achieving good results (Magnin
et al., 2017a; Rico et al., 2021; Etzemüller et al., 2021), it involves strong assumptions. Correlating GST with aspect and elevation only, as proxies of solar radiation and air temperature, disregards other processes as near surface air advection and



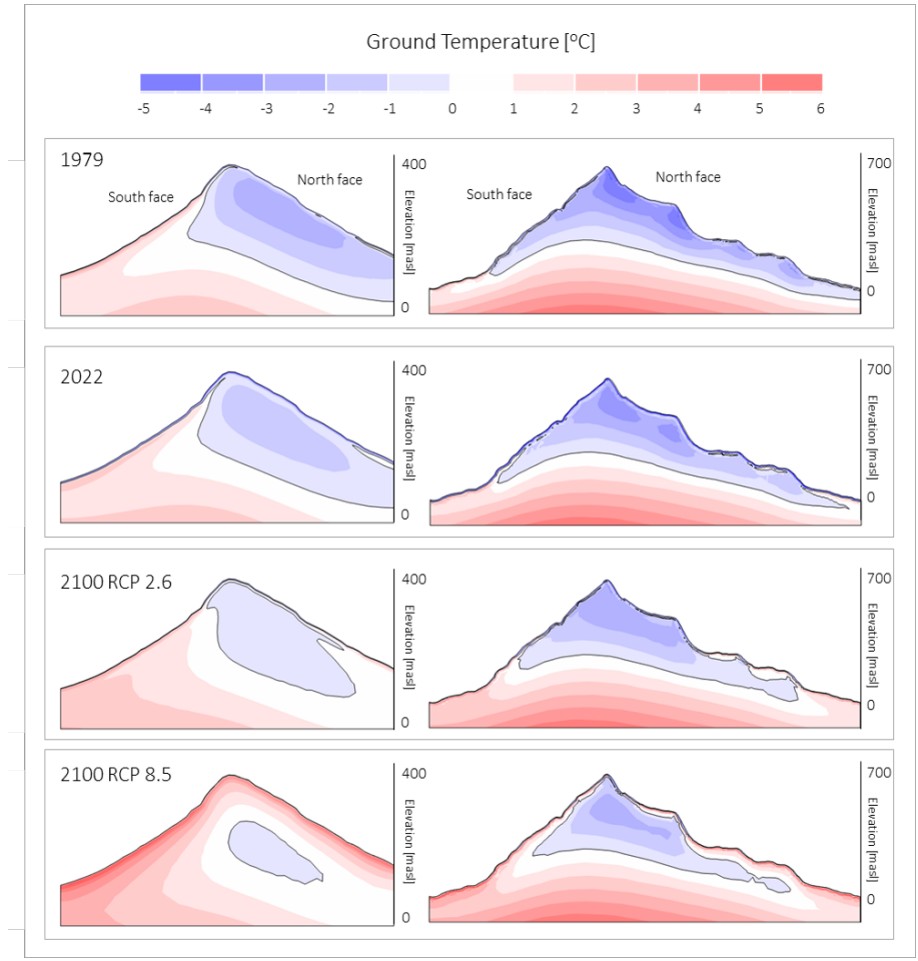

**Figure 9.** Modelled evolution of ground temperatures on Nattoralinnguaq (left column) and Nasaasaaq (right column). The images are issued form 2D models.

longwave radiation. Although our method is slightly more complex as it involves other weather parameters for the period 1979-2022, the basic assumption of linear relation between GST and topographical settings still holds.

To better contextualize our result, we can compare our model to Schmidt et al. (2020) which represents the state of the art
of numerical modeling of the physical processes affecting rockwall temperature in the arctic. Their approach is based on the SEB module of Cryogrid 3 modified to account for vertical terrain, including vertical moisture transport affected by latent heat flux and skiview factor adapted to steep terrain. By comparing model runs and field data, Schmidt et al. (2020) obtained $R^2$ = 0.97 and RMSE = 1.2°C on monthly GST data, which is comparable to our model performance on the validation dataset c ($R^2$ = 0.99 and RMSE = 1.6°C). Although this comparison is encouraging, we suggest that the time period covered by our data
is still too short (only one year) to fully understand the predictive performance of our model. Maintaining the operational the



GST monitoring network and updating the model as time passes will be crucial to define with more confidence this source of uncertainty.

Another source of uncertainty is due to snow accumulation. Snow is known to cause severe disturbance to GST, which can significantly affect active layer thickness even when accumulating in isolated patches (Magnin et al., 2017b). Although our

estimated offset is comparable to previous findings in Greenland (Rasmussen et al., 2018), models are extremely sensitive to snow characteristics (Etzemüller et al., 2021), causing estimation of permafrost extents to greatly vary depending on the modeling assumptions (Czekirda et al., 2019). Additionally, the offset method disregards snow physics as melting of the snowpack, layers metamorphism, and coupled interactions at the ground-snow interface. This method is incapable to explain how snow can sometimes cool the ground depending on the timing of its onset and its persistence during early heatwaves

(Lerjen et al., 2003).

Typically, snow free areas are outlined by applying a filter snow/no snow area using a slope threshold (Magnin et al., 2019). Our snow probability map proposes a description of snow accumulation in steep chutes and snow absence on flat wind-exposed outcrops. Overall, this approach, provides a good idea on how and where ground temperatures are influenced by snow cover at the large scale. Although we do not have enough data to validate this approach, it is encouraging that the SnowP map provides

an acceptable boundary condition to model deep ground temperature in the snow-affected borehole SIS2019-02 and snow-free borehole SIS2021-01.

To provide a more detailed quantification of such processes, various numerical algorithms as CryoGrid, Snowpack and Crocus, are used to describe large scale – hectometric resolution permafrost distribution (Gisnås et al., 2014). At this resolution and scale, it is possible to obtain relevant information on snow characteristics using remote sensing data. Nevertheless, when

dealing with the metric resolution required to successfully model steep bedrock permafrost, it becomes extremely challenging to achieve a good knowledge of spatial characteristics of the snow, given the spatial variability of snow redistribution by wind. This issue has discouraged most authors to use a numerical method to approach the problem. Nevertheless, some studies ventured into a detailed modeling of such process at a smaller scale – decimeter resolution – for selected slopes. In particular, Haberkorn et al. (2016) proposed the combination of repeated Terrestrial Laser Scanning and Alpine3D to finely model (0.2 m

resolution) snow deposition on steep terrain and its influence on the rockwall surface temperature. Although such approach is not directly applicable at our scale, this framework seems to be the right approach to provide detailed knowledge of a selected slope of interest. In this sense, the offset method used in this study is relevant to provide a general frame of the influence of snow on ground temperatures at the large scale, while a more detailed approach is recommended when dealing with site-specific situations.

## 5.1 Heat transfer modelling

Our heat transfer module is representative for isotropic bedrock as it does not consider anisotropic thermal conductivity caused by rock porosity. Therefore, sediments, vegetation patches, debris and individual large fractures are not correctly modelled and will need a dedicated mapping and modelling effort to compute a complete permafrost map of the area. Concerning bedrock, factures can either warm the bedrock through water advection, while air circulation can cool the bedrock (Hasler et al., 2011).





Ice cemented fractures are major drivers in permafrost aggradation and degradation, and their behavior is not representable by a simple conductive approach (Hasler et al., 2012). Fractures delay permafrost degradation when bedrock is warming, but as soon as ice begins to thaw, they create thawing preferential corridors (Magnin et al., 2020). Furthermore, ice cemented fractures are major drivers in bedrock permafrost stability (Hasler et al., 2012) as their thermal behavior create strong anisotropies in the hydraulic head and fluid pressure (Magnin et al., 2020). Therefore understanding their behavior is not only relevant for

correctly model ground thermodynamics, but also slope stability.

  Another major limitation is due to the use of a 1D model for the mapping module, which disregards lateral influences. Although the mountain terrain near Sisimiut is mostly gentle, we expect our map to be imprecise in the proximity of sharp slope breaks and ridges. 2D models on the other hand provide a much stronger approximation in complex terrain, as they provide similar results for 3D transient simulations (Noetzle et al., 2007). In this sense, representative 2D transects seem

adequate to describe and predict permafrost evolution patterns in complex terrain, unloading computational efforts. Overall, our approach is adapted to provide a large-scale and computationally efficient evaluation of permafrost characteristics using a numerical approach. However, when more detail is required in a specific site, more sophisticated approaches exist. In particular, the new approach described by (Magnin et al., 2020) seems to overcome most of the aforementioned issues. Their approach describes thermal process in fractured bedrock using a quantitative numerical approach in FeFlow. Deep fractures are directly

represented in the model mesh and their geometry can be evaluated by lithological interpretation of dip angle of main surface discontinuities (Mamot et al., 2020) or by Seismic Refraction Tomography (Phillips et al., 2016). These approaches are advised for selected study sites with high relevance, justifying the effort of collecting the data required to run the – CPU intensive – numerical model.

## 5.2 Global performance

To have an idea of the overall performance of our model, we can compare our results to Magnin et al. (2017a) which use a comparable approach. Magnin et al. (2017a), reaches an average difference between modelled and measured temperatures at 10m depth of 0.01°C, indicating a significantly better performance than our model. Since most of the disagreement between our model and measured data occurs above 10m depth, we believe that, this is likely imputable to our climatic database. While Magnin et al. (2017a) disposed of near-in situ long term weather stations, our data which seems to be not precise enough to

explain short-time variability in ground temperatures. Also, our boreholes are on flat terrain, which are influenced by lateral variations in snow accumulation and surface characteristics, while Magnin et al. (2017a) dispose of boreholes on vertical bedrock. These different surfaces are expected to create lateral influences on the borehole 1D column, causing deviations from the ideal isotropic conditions simulated by our model. Finally, our model is based only on one year of data coverage, which is likely a too short time period for achieving a full understanding of the processes involved and a proper calibration of the GST

models.

  Given this, our results indicate that the model describes within 0.1 C deep ground temperatures (below 10m meters) for SIS2021-01 and similar values are achieved on the validation borehole SIS2019-02. Since SIS2019-02 was not used for numerical model calibration, we have good confidence in estimating an expected uncertainty of 0.1-0.2 °C when estimating deep





temperatures. This performance is satisfactory as it suggest that the model fulfills the study's aim to describe general bedrock
permafrost conditions in the area with acceptable confidence. This conclusion is also supported by the comparison between
our 2D model and the ERT inversion jointed with petrophysical analysis. Considering that the unfrozen area on the north face
indicated by the ERT is likely due to a lithological fault observed on the field, indicating that the petrophysical analysis and
isotropic assumption is not valid in this area, there is an overall qualitative agreement between ERT and numerical simulation.

### 5.3 Present-day bedrock permafrost characteristics compared to other regions

Keeping in mind the limitations we discussed above, we can use our results to discuss the general characteristics of bedrock
permafrost in the Sisimiut area and their future evolution. Ground surface temperatures in rockwalls seems comparable to
conditions described in Northern Norway (69 – 71 °N), where permafrost can be found at sea level as sporadic on north
facing slopes (Magnin et al., 2019). In Sisimiut, the solar radiation creates an offset of 2.4°C from North to South facing
slopes, causing a rise of about 400 m of elevation in the permafrost 0°C isotherm between this two aspects. This offset is
known to be dependent on latitude, varying from 8°C m in the European Alps (45-46 °N, Magnin et al. (2015a)) to 1.5 °C in
Northern Norway (69-71 °N, Magnin et al. (2019), Figure 10). In coastal climates, previous studies suggested that steep bedrock
permafrost could be influenced by other factors than pure solar radiation, as cloudiness and icing, creating an abnormally
low offset in New Zealand (Allen et al., 2009). In this context, the North-South offset we measured in the Sisimiut area is
consistent with the latitudinal trend obtainable by previous studies, suggesting that, despite the fact that the Sisimiut mountain
area is coastal, pure solar radiation is dominant on landscape-scale permafrost characteristics, after elevation-dependent air
temperature variations.

### 5.4 Future evolution of bedrock permafrost in the area and implications

Our model suggests that as of 2020 the deep ground temperatures are in disequilibrium with the current climate. This is high-
lighted by the fact that, even in scenario RCP2.6, which causes a relatively mild increase in air temperatures and MGT20, the
permafrozen bedrock area will decrease by about 35% by 2090-2100. This corresponds to the disappearance of permafrost in
most low elevation south facing slopes and plateaus, as well as an increase of the active layer thickness for most of north facing
slopes at low elevation. This situation becomes more and more critical with scenario RCP8.5, where only 5% of permafrost
ground existent in 2022 will survive to the end of the XXI century. As highlighted by the 2D simulations, this does not involve
a dramatic reduction of the deep frozen bodies, as they will persist in form of relict permafrost well after the end of the XXI
even in the pessimistic scenario. These results are comparable to the French Alps where mountain permafrost is expected to
retreat only on the highest summits of the Mont Blanc massif, while relict permafrozen bedrock can persist at lower elevations
(Magnin et al., 2017a).

Our findings imply that in the near future permafrost degradation will affect most of the rockwalls in the Sisimiut area,
creating the preliminary conditions for a possible increase in rockfall activity of both small and large magnitude (Krautblatter
et al., 2013). Although the correlation between permafrost degradation and rockfall activity is accepted within the scientific
community (Ravanel and Deline, 2011; Patton et al., 2019), the process chain linking the two phenomena is very complex. In

general, permafrost degradation causes a progressive deepening of the frozen body, increasing water circulation in cracks and weakening of the mechanical properties of the ice-bedrock matrix (Davies et al., 2001; Krautblatter et al., 2013). This does not mean that we automatically expect an increase of rockfall activity in the area, as lithological predisposition to failure, as factures dip versus slope, is the major control on stability. In addition, water infiltration patterns play a role in stability, as rainwater may infiltrate the bedrock while snow meltwater cannot percolate in case of ice basal layer (Phillips et al., 2016). Based on our results, we suggest that, for those areas already known for existing rockfall activity and highlighted by our model as expecting severe permafrost degradation, future efforts should be made to better assess rock slope stability.

## 6   Conclusions

This study presents a first quantification of bedrock permafrost on mountain terrain in Sisimiut, West Greenland, using a heat transfer module forced by simplified weather data. The modeling approach produces results that are consistent with available data from deep boreholes and geophysical investigations. Based on our results, permafrozen bedrock can be found at sea level on north facing/snow free slopes, while on south facing slopes the lower margins are at about 400 m.a.s.l.. This indicates that, considering the local topography, most of the mountain terrain hosts temperate permafrost. Forcing our model with future climatic projections shows different degrees of permafrost degradation based on the scenario used. For scenario RCP8.5, i.e. with no mitigation on carbon emissions, our model predicts a reduction of 95% of the active permafrost area, meaning that permafrozen ground will persist only as relict condition at depth greater than the reach of the seasonal frost. This condition suggest a future strong disequilibrium between ground thermal conditions and climatic forcing. Future efforts in the area should focus on investigating slope stability characteristics, and their relation to permafrozen conditions. Once (and if) problematic slopes are identified, site specific models integrating high resolution snow distribution and crack mapping will provide a more detailed understanding of slope thermodynamics, overcoming the main uncertainties of our model. Further efforts should also apply at larger scale, in order to characterize mountain permafrost in the whole region. In this sense, our modeling approach based on few weather parameters, downscalable with a simple topographical approach, provides a good trade-off between results quality and uncertainty.

*Data availability.* Ground temperature data are available at: Marcer, Marco (2022): Dataset - Bedrock Permafrost in Greenland. Technical University of Denmark. Dataset. https://doi.org/10.11583/DTU.21215591

*Author contributions.* MM designed the study, conducted fieldwork and modeling. PAD conducted geophysical fieldwork and data processing. ST participated in geophysical fieldwork. SRN organised deep boreholes drillings. AR advised on geophysical data processing. TIN supervised the study, field logistics and data interpretation. All authors contributed to the manuscript.



*Competing interests.* No competing interests are present

*Acknowledgements.* This study is part of project TEMPRA and Siku Aajiutsoq, funded by the Greenland Research Council. The study is also part of the Nunataryuk project, which is funded under the European Union's Horizon 2020 Research and Innovation Programme under Grant Agreement 773421. The deep boreholes were established as part of the Greenland Integrated Observing System (GIOS) funded by the Danish National Fund for Research Infrastructure (NUFI) under the Ministry for higher Education and Science. This work was realised in
cooperation with EDYTEM and Styx 4D. We acknowledge Jessy Lossel for his work on the rock samples.



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
