# Peer review of "Modeling present and future rock wall permafrost distribution in the Sisimiut mountain area, West Greenland"

_The Cryosphere, 2022_

## Referee Comment (RC1)

[referee-annotated manuscript omitted]

---

## Referee Comment (RC2)

**Characteristics and evolution of bedrock permafrost in the Sisimiut mountain area, West Greenland**

Marco Marcer1,2, Pierre-Allain Duvillard3,4, Soňa Tomaškovičová1, Steffen Ringsø Nielsen2,5, André Revil4, and Thomas Ingeman-Nielsen1

1DTU Sustain,Bygningstorvet, Bygning 115, 2800 Kgs. Lyngby, Denmark
2Arctic DTU, Siimuup Aqqutaa 32, B-1280, 3911 Sisimiut, Greenland

[revised manuscript text omitted]

---

## Author Response (AR1)

Author's response to reviewers on

**"Characteristics and evolution of bedrock permafrost in the Sisimiut mountain area, West Greenland"**

by Marcer et al., The Cryosphere Discuss.,

https://tc.copernicus.org/preprints/tc-2022-189/

**Contents**

**Referee #1**

Anonymous Referee #1
Referee comment on "Characteristics and evolution of bedrock permafrost in the Sisimiut mountain area, West Greenland" by Marco Marcer et al., The Cryosphere Discuss., https://doi.org/10.5194/tc-2022-189-RC1, 2022

The manuscript "Characteristics and evolution of bedrock permafrost in the Sisimiut mountain area, West Greenland" presents an efficient approach for the modeling of bedrock permafrost in Greenland. Aiming at the prediction of the bedrock permafrost evolution the study considers two different regional carbon pathway scenarios to model the permafrost distribution at the end of the 21st century. Accordingly, this manuscript addresses a topic that has so far been underrepresented in the existing literature, and thus is particularly relevant for readers of The Cryosphere. The abstract of the manuscript is well-written arousing the interest of the reader by providing a concise yet complete overview of the study. Unfortunately, in its current version the manuscript itself fails to meet the expectations raised by the abstract. While the introduction is also of good quality (objective/aim of the study are nicely described) the following sections do not adequately present the otherwise great outcome of this study.

We thank the reviewer for this detailed feedback that will help us improve the text. We are also glad that the reviewer values the topic we chose for our study, and most of the concerns regard the text rather than the essence of the study. In this document, the reviewer can find the answers to the main points, while we reply to the notes in the annex directly in the supplementary file.

**General comments**

In particular, my main points of concern are:

**Structure** The manuscript lacks a clear structure, which also affects the adequate separation of the (content in the) different sections. In particular, the authors could consider merging the Results and Discussion sections, which would allow for a more concise presentation of this really interesting study.

We agree that current structure of the manuscript does not meet the requirements for scientific publication. As direct response to this comment, we would like to keep a Results and Discussion split, in favor of a traditional writing style. We thank the reviewer for the great effort in commenting the text, pointing out where structure could be improved. In particular, the major changes:

- We moved the section introducing snow modeling to the introduction P2L45-P2L54.
- We restructured the methods' sections, and added a clear "Snow cover modelling" section P9L215
- We removed the calibration results from the results section and cited these results as loggers properties in the methods P4L107, P4L111
- We made the discussion significantly more concise. We avoid all the discussion on slope stability, which is not relevant for this study.

**Figures** In general, the figures are of good quality and nicely prepared, i.e., by just looking at the figures the potential of the study is evident. However, not all figures are correctly referenced in the text. Moreover, the figure contents are not properly described/discussed in the text with some sub figures not being addressed at all.

We agree with the comment; we apologize for the incorrect referencing of figure in the text and we thank the reviewer for pointing out the error. Figures are now properly referenced and addressed in the text. We also have added a new figure (Fig. 10, P21), to support the discussion of section 5.2. Also, following the instructions from reviewer #2, we made major changes to:

- Fig. 3: data are now color coded depending on snow cover type, P13
- Fig. 4: data and model from SIS2019-02 are now presented in the figure, P14

**Tables** In general, the structure of the tables is fine; yet, in the text, the authors refer to a Table 3 that is not included in the manuscript.

Thank you for this remarks; this is a bad blunder. Table 3 belongs to an older version of the document, while in this version we write the material's properties are given in the text instead. Also notice, following the instructions form reviewer #2, we improved Table 1 (P6) and Table 2 (P8).

**Captions** The text of figure and table captions should provide more information so that the reader can easily understand the presented content. In the current version of the manuscript, the captions lack a consistent structure and information content.

This is also a solid point, and we agree with the reviewer. We have improved the captions of all figures.

**Numbers and Units** The authors did not implement the guidelines regarding the correct formatting of numbers and units.

Yes, we overlooked these guidelines, and we will solve the issue upon revision. We thank the reviewer for highlighting with detail in the text where this should be corrected, this helped us to revise the text. See the point by point corrections for details.

**General** I suggest that the authors consult and implement the manuscript preparation guidelines provided by The Cryosphere to ensure the manuscript meets the formatting requirements as well as general quality standards (especially with respect to sentence/paragraph structure and formulations). Obviously, in the current version, the manuscript does not fulfill standards expected for scientific publications. However, due to the relevance of the presented study for permafrost research in Greenland and taking into account the good quality of the (final) products of the study I suggest that the manuscript should be reconsidered for publication in The Cryosphere after major revisions.

We would like to thank the reviewer for this work; we appreciate the effort to help us improving the text.

Finally, we propose a major revision at L137, concerning the petrophysical analysis. In order to clarify the evolution of temperatures below -10°C, we re-saturated the samples and measured their electrical conductivity as a function of temperature in a thermostat bath. This protocol has been used in previous studies such as Coperey et al. 2019 (https://doi.org/10.1029/2018JB017015). We chose to redo these measurements because the measurements in the first version were done randomly between -8 and +10°C. The temperature was not controlled and stagnated during the measurement, which led to an error. For these reasons, we chose to perform these measurements with a thermostat bath in order to control the temperature of the bath as well as the sample. We also extended the temperature range from -15 to +15°C and double checked that the sample temperature was at equilibrium before each measurement. The results of the measurements are more precise and very similar as they are within the same range of values.

**Line-by-line comments**

In the supplementary file, I provide detailed comments and suggestions that might help the authors during the revision of the manuscript. The annotations use the following code:

**Highlighted (yellow)** Should be addressed/considered during the revision.
**Highlighted (red)** Needs to be addressed/considered during the revision.
**Strike-through (red)** Remove.
**Underline (red)** Indicates repeated words in single sentences.
Thank you for this very detailed work. We respond to these comments with page and line number of the original comment, reviewer comment, authors' answer and, when relevant, page and line position of the new text.

[P1 L10] frozen/unfrozen
Now "frozen/unfrozen", P1L10

[P1 L12] until the end of the 21st century.
Now "until the end of the 21st century", P1L12

[P2 L27] Maybe add reference(s) here, e.g., PERMOS.
Added Pellet and Noetzli (2020); Isaksen et al (2022), P2L23

[P2 L42] as well as
Now "as well as", P2L36

[P2 L41] and more complex 3D
Now "more complex 2D", P2L37

[P2 L46] surface
Now "ground surface",P2L42

[P2 L47] Rephrase to improve sentence structure and readability.
Changed to:
" This approach has the advantage of reaching good performances while requiring only basic climatic input, i.e. air temperature and solar radiation.", P2L43-44

[P2 L55] Rephrase
Changed to:
"The statistical model is then used to compute time series of GST at any location in the landscape and for the period 1850-2022.", P3L60-61

[P2 L55] -/ ... use the COMSOL heat ...
-/ "Use" and "using" in one sentence; suggest to rephrase
Second "using" removed, P3L64

[P3 L57] Whitespace between number and unit missing. Found throughout the document - needs to be addressed.

Yes, this is has been corrected through the text

[P3 L58] based on

Accepted, P3L66 -67

[P3 L59] Wording could be improved.

Sentence changed to:
"This methodology develops a bidimensional transect of ground freezing conditions at a given survey date, which is compared to our 2D numerical simulations". P3L67-69.

[P3 L63] Check the guildelines on how to refer to figures (and tables) in the text:
https://www.the-cryosphere.net/submission.html#figurestables

Figures and tables are now referred following the guidelines.

[P3 L70] Reference?

This is based on the data from Cappelen et al 2021. This part has been modified and reference made explicit, P3L81-86

[P3 L75] In the area,

Sentence modified to:
"This climate locates Sisimiut in the sporadic permafrost zone (Obu et al.,2019; Biskaborn et al., 2019), and morphologically active rock glaciers are present in the area, reaching sea level elevation", P3L87

[P3 L75] Rephrase to improve wording.

Sentence modified to:
" This climate locates Sisimiut in the sporadic permafrost zone (Obu et al., 2019; Biskaborn et al., 2019), and morphologically active rock glaciers are present in the area, reaching sea level elevation", P3L87

[P3 L77] Is this sentence necessary?

Sentence removed

[P4 L86] Abbreviation ERT has not been introduced yet.

Now referred as "geophysical profile", P4L88

[P4 L90] ?

Removed

[P4 L91] Suggest to combine with section 3.1.1.

Now combined with 3.1.1 in section 3.1, P4L92

[P4 L93] Separate sentence?

We split the sentence in two, P5L102

[P5 L96] is a common trade-off

Accepted, P5L106

[P5 L99] Is this an assumption? Or can you provide data/references to sustain this?
This is based on Gubler et al 2011. Sentence changed to:
"According to Gubler et al. (2011), about 5-10% of the deployed loggers can be expected to fail and protecting the logger with plastic film (as we did) helps to reduce failures", P5L109-110

[P5 L103] Use either iButton or DS1922L (or both) - be consistent.
Yes, we use iButtons consistently through the document now

[P5 L105] Suggest to merge with section 3.1.1.
Yes, see answer [P4 L91]

[P6 L121] m.a.s.l. Ensure consistent usage within the document.
Yes, m.a.s.l. is used consistently through the document now

[P6 L122] (approximately 100 m high)
accepted, P6L137

[P6 L124] Which infrastructure was relevant for the decision (besides the walking path that is mentioned in the second part of the sentence)?
This summit is located near a road, simplifying the logistics. The sentence has been changed to:
"This specific mountain was chosen for its accessibility, as the road leading to the airport passes just nearby a short path that leads to a popular viewpoint to the summit."  P6139-140

[P6 L125] conducted in early
accepted,P6L144

[P6 L125] 100 m long
accepted,P6L145

[P6 L126] 500 m long
accepted,P6L145

[P6 L126] deployed with 5 m spacing
accepted,P6L145

[P6 L128] improve the galvanic contact/reduce the contact resistances
accepted, P6L147

[P6 L129] Hard to follow - consider rephrasing. I suggest to provide a brief description of the electrical resistivity tomography method. Such description is particularly relevant as you mention "(in-phase) conductivity measurements [...] obtain[ed] at a frequency of 1 Hz" (line 145f).
We have added a brief introduction to the methodology:
" Electrical resistivity tomography (ERT) yields only qualitative information on the thermal state of materials because electrical conductivity depends on many parameters including water content, salinity, cation

exchange capacity, and temperature. The advantages of these geophysical methods are their low cost and the fact that they provide 2D or 3D tomograms/images of the subsurface." P6L141-144

[P6 L131] ... was extracted from a DEM based on electrode positions measured with a handheld GPS device.
 accepted,P6L149

[P6 L136] What was the convergence criterion?
 The difference between two successive data misfit error is smaller than a prescribed value.

[P6 L133] Provide further information about the filtering.
 We have modified the sentence to:
 "We cleaned 4% of the data 150 point acquired before the inversion (549 points acquired, 528 inverted) by filtering out the outliers from the pseudo section." P7L151

[P6 L138] two rock samples
 accepted,P7L155

[P6 L139] ?
 Sentence changed to:
"In addition to the field measurements, we performed a laboratory electrical conductivity experiment on two rock samples collected in the field from the rockwalls on the south and north face." P7L155-156

[P6 L141] were dried for 24 hours
 accepted, P7L159

[P6 L140] were
 accepted, P7L158

[P6 L142] are
 accepted, P7L158

[P6 L143] Check if this is the correct format.
 Format corrected, P7L161

[P6 L139] The three granite cubic core samples considered for laboratory analyses ...
 accepted, P7L156

[P6 L146] Consider rephrasing to make it clearer and enhance readability.
 Section changed to:
"We moved the freezing point temperature TF = 0 °C based on direct observations on instrumented boreholes for G-RF. The measurements with TF = -3 °C reflect the fact that the measurements were made only in the downward direction of the temperatures and not in the upward direction. These analyses define the relation between resistivity collected in the field and freeze-thaw conditions of the ERT transect." P7L166-169

[P6 L146] What does this frequency refer to? Was never introduced.
(See comment regarding a description of the ERT method).
 The complex conductivity in the laboratory is observed by making measurements at different frequencies of the electrical field to observe the polarization of the ice between the grains for example. Only the data at 1 Hz is of interest to us here as it is representative of the field measurement conditions. Induced polarization can be done in the frequency or time domains. Usually we do frequency domain measurement in the lab and time-domain measurements in the field. We can go from one domain to the other through a Fourier transform analysis.

[P7 L152] evaluates
 accepted, P7L172

[P7 L154] refers to
 accepted, P7L174

[P7 L160] provides
 Sentence modified to:
"The regression is then used to generate air temperature for the period prior to 1958", P7L182-183

[P7 L161] based on
 See answer [P6 L160]

[P7 L161] yielding the air temperature values of dataset c.
 See answer [P6 L160]

[P7 L163] from/through
 accepted, P8L185

[P7 L165] Earth
 accepted, P8L187

[P7 L166] Rephrase to improve sentence structure and readability.
 Sentence modified to:
"The NorESM1 model is also chosen by several authors in Greenland for cryosphere evolution modelling due to its good performance in the region (Colgan et al., 2016; Hofer et al., 2020) thanks to his good performance in the region (Fettweis et al., 2011)." P8L189-191

[P8 L172] depends
 accepted, P8L194

[P8 L174] Never introduced
 Now introduced at P8L196

[P8 L177] Never introduced.
 Now introduced at P8L200

[P8 L178] from
accepted, P8L198

[P8 L179] -/ What is i?
-/ Does * denote a convolution or just a multiplication?
-/ In the text, write variables/terms of the equation also in math style/mode.
We deleted this formula., as it is not providing useful information and generates confusion instead.

[P8 L187] in
accepted, P9L224

[P8 L200] Not clear - provide further information.
This information is trivial and confusing, as it repeats the previous section. It has been removed

[P8 L200] by using
changed to "trained with", P9L214

[P9 L206] which assumes the local thermal equilibrium hypothesis to be applicable/valid.
accepted, P10L232

[P9 L207] There is no Table 3 in the manuscript.
Thank you for the remarks. Table 3 belonged to an older version of the text. We now present the material properties directly in the text instead of using a table

[P9 L211] from
accepted, P10239

[P9 L209] Check correct formatting.
Formatting corrected, P10L236

[P10 L220] from the
accepted, P11L250

[P10 L224] The structure of this description is hard to follow - consider rephrasing.
Changerd to:
"The numerical model is calibrated for two parameters: matrix porosity and initial conditions in 1850. The calibration is carried out by simulating conditions in SIS2021-01, from 1850 to 2022 using a 1D geometry of a 100 m column. The simulation results are then compared to the field data acquired during the period August 2021 to April 2022. This is repeated for different combinations of matrix porosity and initial conditions, aiming to minimize the difference between data and model results." P11L254-256

[P10 L233] Little Ice Age?
Accepted, P11L263

[P10 L240] through
Accepted, P11L269

[P10 L244] profile
Accepted, P12L279

[P10 L245] are of interest
The entire sentence has been changed to provide a stronger argument for the 2D models location choice: "For this reason, we compute 2D model for two location of special interest: the ERT profile and the Nasaasaaq summit. The first location is chosen to compare the ERT data to our model, while the second location allows us to observe permafrost distribution and evolution in the tallest mountain in the study area." P12L178-281

[P10 L246] Unclear.
-/ What is the elevation profile z?
-/ Which solid?
Provide further information and a more precise description of the workflow.
The sentence has been modified to:
"For each location we set-up a north-south transect in the QGIS software, and used it to sample the elevation profile from the DEM. The elevation profiles are then imported into COMSOL as 2D geometry using the parametric function option. " P12L281-283. In COMSOL, 2D geometries are called "solid". This information is not necessary and is now deleted.

[P11 L249] See comment regarding the other equation above.
Equation changed accordingly, P12L285

[P11 L250] where
Changed to "where", P12L285

[P11 L251] from borehole
Changed to "from the borehole", P12L286

[P11 L244] I consider this a report of the sensor calibration and not results of this study.
We now present this briefly in the methods while describing the sensors, P5L106, and P5L111

[P11 L259] Likely 4.2.1
The whole section structure has been modified and this part belongs now to 4.1, P12L289

[P11 L265] Likely 4.2.2
This is now integrated in 4.1, P12L289

[P11 L260] In SIS2019-02, temperature data were
Sentence modified to:
"In SIS2019-02, the depth of zero annual amplitude is approximately 20 m. Below this depth, temperature data indicate a minimum of temperature of +0.3 °C, reached at 30 m depth, and a temperature of +1.0 °C at 100 m.", P12L301-303

[P11 L261] , which indicate

See answer P11L260

[P11 L261] whitespaces between number and unit missing
 See answer P11L260

[P11 L262] The length of the time series does not allow for a precise delineation of the ALT?
 We now do not talk about active layer, but rather of depth of zero annual amplitude: "Since temperatures are positive below the depth of zero annual amplitude, the measurements at SIS2019-02 indicate absence of permafrost.", P12L303-304

[P11 L266] Different number of loggers mentioned above.
 Now corrected, P5L101

[P11 L267] Figure 3 does not show this.
 Now corrected to "Fifteen loggers present snow free data (Fig. 3a), seven present thick snow cover and six present intermediate characteristics (Fig. 3b).", P12L291-292

[P11 L267] For
 Sentence changed to:
"To show the effect of elevation, we compare two snow-free loggers installed on south facing rockwalls, one at sea level (MAGST = +3.5 °C) and at 460 m.a.s.l. (MAGST = +1.2 °C).", P12L297

[P11 L271] Use ( and ) instead of -
 Sentence modified to:
"We used this value of +1.58 °C, comparable to previous findings in Greenland (Rasmussen et al., 2018), as constant offset when modeling snow cover." P14L322

[P11 L274] Also lateral variations observable.
 Sentence changed to
"As shown in Fig.5b, the electrical conductivity tomograms acquired show a vertical and also lateral variations distribution of the conductivities", P13L308-309

[P11 L274] As shown in Fig. 4b, the electrical ...
 See answer P11L274

[P11 L260] Why are the data not presented (figure or table)?
 Data are now presented in figure 4b, P14L323

[P12 L276] How did you perform the petrophysical analysis?
 We have added a brief introduction to the methodology: ""Before performing the laboratory measurements, the samples were dried for 24 hours then saturated under vacuum with degassed water from melted snow taken in the field. The samples were left several weeks in the solution to reach chemical equilibrium before performing the laboratory measurements. The water conductivity at 25 °C and at equilibrium was 0,0118 S m-1 for G-DA and 0.0142 S m-1 for G-RF and G-LR.  The sample holder was placed in a heat-resistant insulating bag immersed in a thermostat bath (KISS K6 from Huber; bath volume: 4.5 l). The temperature of this bath was controlled with internal sensor the temperature of the sample was

control with external sensor with a precision of 0.1 °C. Glycol was used as heat carrying fluid and the conductivity measurements were carried out with the impedancemeter.", P7L159-164

[P12 L275] ?
 Figure 5 now shows the color bar with scale in resistivity and also in conductivity.

[P12 L276] -3.5
 Agreed, P13L310

[P12 L277] close to
 Agreed, P13L310

[P12 L277] Is this an assumption, an interpretation or an a-priori known fact?
 This is an interpretation. The sentence has been moved to the discussion:
"This anomaly occurs near a large lithological fault, visible in the field. Overall, the observations indicate that ground characteristics at this location are not isotropic and a direct comparison between model and geophysics is not meaningful.", P20L401-403

[P12 L277] and the lower
 This was a mistake, it is now changed to "the upper", P13L311

[P13 L284] Figure 5!
Almost all references to figures are wrong.
 Figures references have been updated through the text

[P13 L284] For
 Accepted, P13L317

[P13 L292] deviates
 Accepted, P14L327

[P13 L292] from
 Accepted, P14L327

[P13 L289] 0.03 is the value for which parameter/variable?
 For porosity. Sentence now changed to "The optimal porosity was achieved for a value of 0.03, ", P14L323

[P14 L296] What are cold/warm colors? Please refer to the actual values (or colors).
 Sentence removed, not relevant

[P14 L298] down
 Sentence removed as already the concept is already explained in the methodology section, P11L269

[P14 L299] different?
 This information is trivial, and omitted in the new version of the text

[P14 L300] Why is such plot relevant/needed?
 This plot gives a straight-forward summary of ground temperature at different elevations and aspects. It is used in the text to give description of permafrost distribution in terms of elevation and aspect, P15L335-P16L338

[P14 L304] This sentence seems to be incomplete. What is the actual message?
 Sentence modified to: "The colder MGT20 occurs on the north face of the Nasaasaaq peak (763 m.a.s.l. ), reaching -4.0 °C.", P16L337

[P14 L308] Figure 7 does not compare modeling and ERT - maybe the authors meant to refer to Fig. 4b and c?
 Figure is now properly referenced, P16L344

[P14 L314] This is presented in Figure 8c?
 Accepted, P17L349

[P15 L316] 21st
 Accepted, P17L351

[P15 L316] RCP 2.6
--> Needs to be consistent throughout the document. Check other occurences.
 This is now corrected through the text

[P15 L324] Formatting needs to be consistent in the entire manuscript.
 Formatting now consistent: 2090-2100

[P16 L336] such as
 accepted, P17L373

[P17 L337] ?
 Sentence changed to "The images are computed using the 2D model of ground temperature evolution.", P19L386

[P17 L342] skyview?
 accepted, P17L378

[P18 L346] Conclusions/Outlook?
 Sentence deleted, as not providing relevant information

[P18 L348] Consider rephrasing
 Paragraph deleted; this issue is now explained in the introduction, P2L45-54

[P18 L348] substantially influence
 See answer P18L348

[P18 L352] Introduction/Methods?

See answer P18L348

[P18 L356] ?
Rephrase to make it clear.
See answer P18L348

[P18 L358] information ...
Paragrpah rephrased, "On the other hand, our model is more reliable when describing ground temperatures below the depth of zero annual amplitude. Here, the model has a maximum error of 0.15 °C when compared to SIS2019-02, which is used only as validation dataset. SIS2019-02 has also different snow conditions than SIS2021-01, suggesting that our SnowP map and offset provide an acceptable boundary condition to model long term effect of recurrent snow cover induced by topographical patterns.", P19L395-P20L398

[P18 L360] How/where is this demonstrated?
See answer P18L360

[P18 L362] Read like part of an introduction. Consider rephrasing it as discussion otherwise move to the introduction or methods section.
The concepts described in this paragrpah have been integrated in the introduction, P2L45-51, and in the conclusion, P21L445

[P18 L378] In particular,
Paragraph integrated in the conclusion, P21L445

[P19 L380] Introdcution?
This paragraph as been removed, a focusing too much on slope stability, which is not the aim of this study.

[P19 L390] reducing
Paragraph integrated in the conclusion, P21L445

[P19 L402] Have you conducted a statistical analysis? If not, I would not use the term "significant(ly)".
accepted, P19L389

[P19 L404] ?
Sentence changed to:
"While Magnin et al. (2017a) had near-in situ long term weather station data, [..]", P19L391

[P20 L415] How is the "acceptable confidence" defined?
This is now omitted; we rather provide the measured uncertainty of our model compared to the validation data, P19L394-396

[P20 L415] Merge sentences and make the description more concise.
This concept has been developed more in detail:
"The general agreement between model and geophysical data also indicates that the model is suitable for describing permafrost extents in a wide range of elevations, slopes and aspects. Most of the disagreement

between model and data is due to the electrical conductivity anomaly on the north face of the geophysical profile. This anomaly occurs near a large lithological fault, visible in the field. Overall, the observations indicate that ground characteristics at this location are not isotropic and a direct comparison between model and geophysics is not meaningful.", P20L398-402

[P20 L438] 21st
 accepted, P20L424

[P20 L439] 21st
 accepted, P20L425

[P21 L448] Unclear -> rephrase.
 This has been rephrased and moved to the conclusion: "Although the correlation between permafrost degradation and rockfall activity is accepted within the scientific community (Ravanel and Deline, 2011; Patton et al., 2019), the process chain linking the two phenomena is very complex.", P21L441-443

[P21 L452] Conclusions
 Accepted, rephrased to be integrated in the conclusion:
"Therefore, future efforts in the area should focus on investigating slope stability characteristics, and their relation to permafrost distribution and degradation.", P21L443

[P21 L468] selected
Rephrased as: "In this sense, our modeling approach based on weather parameters readily available for the whole region, downscalable with a simple topographical approach, provides a good first assessment for mountain permafrost zonation", P21L446

[P21 L460] depending on the considered scenario
 accepted, P21L437

[P23 L511] Etzelmüller
 corrected, P26L499

[P23 L514] Already published
 corrected, P26L502

**Referee #2**

Anonymous Referee #2
Referee comment on "Characteristics and evolution of bedrock permafrost in the Sisimiut
mountain area, West Greenland" by Marco Marcer et al., The Cryosphere Discuss.,
https://doi.org/10.5194/tc-2022-189-RC1, 2022

In the manuscript "Characteristics and evolution of bedrock permafrost in the Sisimiut mountain area, West Greenland", a minimalist approach is used to model the spatial distribution and future evolution of bedrock permafrost in the region around Sisimiut, Greenland. Soil temperature measurements from the hydrological year 2020/21 and air temperature data from climate stations in the area are used as input data. The model results are verified with borehole data and a geophysical measurement (ERT). Accordingly, the manuscript addresses a current topic relevant to The Cryosphere. In general, the presented approach is extremely interesting, since a model of small-scale permafrost distribution could be achieved with relatively little data. Nevertheless, no fundamentally new concepts are presented. However, the factor that few publications exist on the distribution and future evolution of permafrost in Greenland makes the manuscript relevant. Unfortunately, there are a number of points of critique that outweigh the many positive aspects of the manuscript. This includes formal, structural as well as methodological and content-related aspects, which I will address below in general, as well as in specific comments within the manuscript. We would like to thank the reviewer for this feedback. We could see that there has been a lot of effort to help us improve the text with reasoned, fair and challenging points. In addition, we appreciate the work done to improve the language in the text, thanks to several comments in the annex. In general, we can see that most of the comments regard the text structure and the way we present and discuss our results. We are glad to see that the reviewer agree with the fundamental aspects of the study and the methodologies. Here we respond to the reviewer's comments, to which we mostly agree, and that we will use as a guide to propose a new version of the text. Due to the variety of comment typology, we organized our answers in the following way:

- Response to the general comments found in the document "Comment on tc-2022-189". Some answers concern main issues that are often present in the annotated annex. These answers are referred in square brackets as, e.g. [M1], [M2].. The square brackets numbers are sometimes used in the point-by-point comments when relevant.
- Line by Line answers to the annotations in the annex file. This is presented with page and line number of the original comment, reviewer comment, authors' answer and, when relevant, page and line position of the new text.

**General comments**

**Formal Aspects:**
- Overall, the manuscript is well written, but typos and incomplete sentences are frequently encountered. Also, a number of sentences are long and somewhat difficult to follow.
- We agree with this comment. We would like to thank the reviewer (also R#1 detailed this issue) for the detailed noting on the text that helped us to improve readability.

- Mathematical formulas, symbols, and units are not used consistently throughout the manuscript and according to The Cryosphere's specifications.
- Yes, the text was missing consistency on this aspect. We corrected the text according to TC's specifications.

- Cross-references to figures and tables are largely incorrect. Also, references in the text could be more precisely placed and all points in the figures (a, b, c..) could be addressed.
- Yes, there was a lot of confusion in the text and we apologize about that. Thanks to reviewer for marking in the text all these mistakes. We have improved referencing through the text by precisely addressing figures and subfigures.

- Furthermore, the figure captions and table headings could be more comprehensive in order to simplify the understanding of the results without having to read through the text. The font size of the axis labels in the graphics is inconsistent and sometimes too small.
- We have implemented the reviewer's suggestions and produced new figures with updated captions.

**Structural Aspects**

- [M1] Generally speaking, the manuscript could be structured more stringently. It should be examined whether one could separate more clearly between methodological background, results and the discussion or merge chapters. For example, climate data, temperature measurements, and geophysics provide a data basis for the modeling, are presented as results, but in contrast to the modeling results are not presented in great depth. Here one could check how far the manuscript can be restructured to address the results sufficiently without letting the main point of the manuscript fade into the background.
- [M1] We agree that the manuscript needed re-structuring. In particular, the discussion section contains several paragraphs that belong to the introduction. This causes some confusion through the text as the reader does not receive the necessary background to understand some of the methodology used and actual discussion, e.g. snow modeling with thermal offset. In some instances, we agree that some results should not be presented in the results section, as they are not the focus of the manuscript, but rather rapidly mentioned in the methods section, e.g. calibration results. In this sense, we are going to reshape the manuscript to improve readability, and would allow us to describe the results more in depth without loosing the main point of the manuscript fade into details. In general, we used this comment to make a number of modifications through the manuscript. The major changes are the following:
    - We removed from the introduction the sentences describing slope stability issues.
    - We added to the introduction a paragraph describing snow cover modeling in permafrost research, and our strategy. This can be found at P2L45-54. In this sense, we discussion relative to snow cover modeling, as suggested in the annotated text by the reviewer.
    - The large portions of the discussion concerning slope stability have been compressed into few sentences and moved to the conclusion. This can be found at P21L442-447

- In some cases, the naming of the chapters does not perfectly match the contents. For example, the chapter "Ground Surface Temperature" is primarily about the calibration of the temperature sensors.

- Yes, good observation. We have adapted the chapter naming to match the actual content. Also, following the ideas from [M1] and the reviewer's comment in the annotated text, we have condensed some subchapters into larger chapters, causing a substantial reorganization of the text's structure. We now have, in the methods and results:
    - Ground temperature monitoring
    - Geophysical data
    - Modeling

**Comments on the content**

- In my view, the title of the manuscript does not optimally reflect its content. Perhaps a title can be found that focuses on the modeling approach as well as the spatial aspects of permafrost distribution, rather than the characterization of permafrost?
- Interesting observation; we agree with the reviewer and the title should better reflect the actual paper's content. We suggest "Modeling and mapping bedrock temperature in the Sisimiut mountain area, West Greenland".

- [M2] As noted above, I think some data could be presented much more comprehensively. This is particularly true for the borehole temperature data. For example, it is not really clear from the information presented whether permafrost - or at least perennial subsurface ice - is present in both boreholes.
- [M2] We agree on this point. To this regards, we have updated figure 6, which now appears in the text as figure 4, P14. This figure provides clear visualization of boreholes data. In addition, the text is failing to refer to this figure while presenting the boreholes results. In this sense, we have added proper referencing to the figure, to help the reader visualize the data we present in the results.

[Figure]

- It would also be interesting to show to what extent the climate data generated from stations 300 km to the south and 250 km to the north are truly representative of the Sisimiut region. This aspect does not appear again in the results section.
- Although this would be an interesting study, it is not covered by the paper's aim. What we aim to evaluate, among other issues, is how the available data are suitable to model permafrost temperatures. Our measure of performance is comparing model to data concerning permafrost only.

- [M3] The use of terms is partly not quite clear. Since the measurement series in the boreholes is only one year, it should be checked whether permafrost can be assumed with certainty, or one should speak of "frozen ground" in places. In my opinion it could be checked whether "Permanently Frozen" can be used instead of "Permafrozen".
- [M3] Yes, interesting point. It is true that we do not have more than two years of measurements, so we cannot define the ground as "permafrost". However, and this concerns SIS2021-01 and ERT profile, we observe negative temperatures at depths that are below the reach of the summer heat. Since these areas remain frozen under seasonal fluctuation, we talk about permafrost. We agree that this point is not clear through the text, and we have improved the text in this sense; now we do not refer to the "active layer", but to the "depth of zero annual amplitude". We agree that "Permanently frozen" is better than "Permafrozen", and have changed this through the text.

**Fundamental aspects**

[M4] The authors conclude that their "modeling approach based on few weather parameters, downscalable with a simple topographical approach, provides a good trade-off between results quality and uncertainty". Even though I have to agree that the results look very promising and are graphically presented in an excellent way, I miss some basic information to evaluate the quality of the results. This concerns the representativity of the boreholes (Lines 404-405) as well as the input climate data (Lines 402-404), the length of the measurement period (Lines 344- 345) as well as the influence of the snow cover (Lines 348-361). Points that are addressed very critically by the authors themselves in the course of their discussion. However, justifications, why the results can be regarded as representative nevertheless, come somewhat briefly.

[M4] Yes, this is a very good point. The discussion on the uncertainties is more substantial than the discussion on the validation, and the reader is left in doubt about the actual value of the study. We believe that a thorough description of uncertainties is necessary to develop further studies in the region, but we also believe that our results are valuable despite these uncertainties. The main point that we have now developed in the text are the following:
- Our heat transfer model reproduce deep ground temperatures, i.e. below the depth of zero annual amplitude, within 0.15 C for both boreholes (P14L331). This is key result as, thanks to ground thermal inertia, deep ground temperatures are influenced by climatic trends rather than short-term variability. In this sense, our heat transfer model has good performance in predicting deep ground temperatures despite the short measurement period. This suggest that the GST model has good coupling with the available weather data.
- These boreholes have similar elevation/aspect, but substantially different snow conditions. Using our snow modelling approach, we manage to model deep ground temperatures accurately. Although we miss the seasonal variability induced by the snow-ground complex interactions, this method is suitable for the goal of modeling deep ground temperatures in varying snow conditions, when snow conditions are determined by topographical patterns (P19L397).

- The ERT data are acquired to test the model at varying elevation and aspects, and overcome the limited representativity of the two boreholes in mountain terrain. Our model is able to reproduce the observed temperature patterns. This result is very important because it allows us to trust the model in complex topographical settings, which is fundamental for mountain permafrost mapping (P20L398).

[M5] Rather, it is written absolutely reasonable that "(...) the time period covered by our data is still too short (only one year) to fully understand the predictive performance of our model. Maintaining the operational (of) the GST monitoring network and updating the model as time passes will be crucial to define with more confidence this source of uncertainty". The question arises whether it would not be within the scope of possibilities either to optimize and extend the data basis, or to check and justify more clearly that the currently available results are nevertheless relevant and representative, and are superior in their significance to simpler models.

[M5] We would like to point out that this sentence describes the performance of the GST model, not the modeling performance in its whole (GST + heat transfer). The meaning of this sentence is that we achieve excellent fit between data and model, but we are cautious to suggest that this empirical approach is as good as the CryoGrid SEB, since we do not have long-term data. The methods and results restructuring now clearly separates GST and heat transfer models. We have also rephrased the discussion accordingly, and hopefully this concept is clearer now.

We would like to point out that our model (GST + heat transfer) offers a substantial improvement to previous studies in Greenland, by reducing the spatial resolution from km to m scale, by calibrating and validating on dedicated data, and by including topography in downscaling. We have now specified the weaknesses of available permafrost models in the region, and how they are not suitable for characterizing mountain permafrost (P2L25-28).

In my opinion, the current state of the manuscript does not meet the requirements of a scientific publication as well as the quality standard of The Cryosphere. As already noted, I think that the work has very good approaches and an absolute relevance. For this reason, I would like to suggest that the manuscript should be reconsidered for publication in The Cryosphere after a major revision. Due to the large number of comments and suggestions, these are included in the supplemental document. Comments marked in yellow correspond to basic comments that should be considered in the revision. Green markings refer to phrases and sentences that should be rephrased to facilitate understanding.

We thank the reviewer for the feedback and we agree with the comment. We would also like to thank the reviewer for the high quality work that was done to improve our manuscript, with both general remarks as well as detailed comments in the annex. The revision has been provided with competence and detail.

[M6] Finally, we propose a major revision at L137, concerning the petrophysical analysis. In order to clarify the evolution of temperatures below -10°C, we re-saturated the samples and measured their electrical conductivity as a function of temperature in a thermostat bath. This protocol has been used in previous studies such as Coperey et al. 2019 (https://doi.org/10.1029/2018JB017015). We chose to redo these measurements because the measurements in the first version were done randomly between -8 and +10°C. The temperature was not controlled and stagnated during the measurement, which led to an error. For these reasons, we chose to perform these measurements with a thermostat bath in order to control the temperature of the bath as well as the sample. We also extended the temperature range from -15 to +15°C and double checked that the sample temperature was at equilibrium before each measurement. The results of the measurements are more precise and very similar as they are within the same range of values.

**Line-by-line comments**

**[P1 L15] The definition could be more precise and supported with references.**
Changed to:
" The term "permafrost" defines ground presenting temperatures that remain below 0 °C for at least two consecutive years. In cold mountain regions, complex topography influences shading, snow distribution and ground type, causing a highly variable distribution of ground temperatures and permafrost (Etzelmüller, 2013)",P1L15-17

**[P1 L17] Haeberli is always a good source, but this statement could be supported with more recent literature. Possibly, among other sources, this one: https://doi.org/10.1016/j.geomorph.2019.04.029**
Sentence changed and newer references added:
"Several field studies describe a significant correlation between warming climate, mountain permafrost degradation and increased slope instability, observed as rockfall frequency (Ravanel and Deline, 2011; Gallach et al., 2020), large rockslide occurrence (Patton et al., 2019; Guerin et al., 2020; Frauenfelder et al., 2018; Walter et al., 2020), high elevation infrastructure destabilization (Duvillard et al., 2019) and debris permafrost creep rate increase (Marcer et al., 2021).", P1L17-21

**[P1 L18] Please provide references**
See answer P1L17

**[P1 L6] ground surface temperature data**
accepted, P1L5-6

**[P1 L6] more specific?**
for the hydrological year 2020/21?
Accepted, sentence changed : "We first acquire ground surface temperature data for the hydrological year 2020/21 to model bedrock surface temperatures time series from weather forcing on the period 1850 - 2022.", P1L5-7

**[P1 L14] In terms of content, I don't think the introduction is very balanced. Slope stability is very present, but is not further addressed in the rest of the manuscript. On the other hand, the state of research is not fully presented. The influence of snowpack, which is frequently addressed later, does not appear here, and other points raised in the discussion would fit better in the introduction.**
This is a valid point and we thank the reviewer for the remark (see also [M1]). We agree that slope stability takes too much of the introduction, while other issues are not presented at all – as influence of snowpack and modeling state of the art. This issue also drags into most of the discussion, as pointed out by the reviewer later in the text. Therefore, according to the reviewer's notes, we have restructured the Introduction accordingly. The paragraph regarding slope stability has been deleted. We instead present a more detailed state of the art of available permafrost models in Greenland - P2L24-31. We also provide a paragraph describing snow influence and modeling strategy:
"Another challenge in modeling mountain permafrost is due to the influence of snow cover. Snow is known to cause severe disturbance to ground surface temperatures, which can significantly affect active layer thickness even when accumulating in isolated patches (Magnin et al., 2017b). Models are sensitive to snow

characteristics, causing estimation of permafrost extents to greatly vary depending on the modeling assumptions (Czekirda et al., 2019). Although some numerical models are able to describe snow physics at hectometric resolution (Gisnås et al., 2014), it becomes extremely challenging to achieve a good knowledge of spatial characteristics of the snow in complex terrain, given the spatial variability of weather forcing, as wind and shading. To overcome this issue when modeling mountain permafrost, snow is often accounted with a topographical approach, based on filtering snow covered areas using a slope threshold (Magnin et al., 2019) to exclude them form the model or to apply specific offsets (Boeckli et al., 2012). Overall, this method allows for a first order quantification of ground temperatures in complex terrain when detailed snow data are not available.", P2L44-53

[P1 L14] Also, the objective could be formulated more clearly. The objective of quantifying permafrost conditions based on modeling in the Sisimut region somewhat exceeds the content of the manuscript, as no reference is made to the entire Greenland.
 We also agree that the object of the study is not properly defined, as our results are limited to a small area in Greenland, and by no means are relevant at the national scale. The objective is now formulated more clearly as follows:

"The aim of this study is to move a first step towards a high resolution regional characterization of mountain permafrost in Greenland. To do so, we focus on the Sisimiut area, (68° N on the west coast), where we have a relatively large amount of data.", P1L55-P2L57

[P1 L0] The title, in my opinion, could be a little more closely tailored to the contents of the manuscript. Perhaps the modeling aspect as well as the spatial permafrost distribution could be brought to the focus a bit.
 The title is now changed to:
"Modeling present and future bedrock permafrost distribution in the Sisimiut mountain area,West Greenland", P1L0

[P2 L26] references? if with reference to the preceding sources, please rephrase
 It is now rephrased as following:
"Therefore, understanding the spatial distribution of mountain permafrost and its future evolution is a key step in understanding these hazards, and several countries started comprehensive programs to monitor this phenomena as a basis for risk assessment (Pellet and Noetzli, 2020; Isaksen et al., 2022).", P1L21-P2L23

[P2 L29] please provide references
 There are no references because, as the sentence explains, there is no study dealing with this specific issue. References relating to available models for lowland permafrost now presented as following:
"Available models are based on numerical simulations at kilometer scale (Daanen et al., 2011), are not calibrated with in-situ data (Gruber, 2012), or valid for sedimentary terrain only (Obu et al., 2019).", P2L25-26

[P2 L45] ground surface?
 accepted, P2L42

[P2 L50] on?
 accepted, P2L56

[P2 L49] Could you be a little more specific about the objective? The focus is clearly on the Sisimiut region. A permafrost quantification related to Greenland is not given. And regional approaches to permafrost quantification already exist in Greenland, so "first" seems a bit imprecise.

Notice that available models at regional scale do not provide quantification of mountain permafrost, due to their low resolution are lack of representativity in mountain terrain/bedrock. This now highlighted as following:

"Available models are based on numerical simulations at kilometer scale (Daanen et al., 2011), are not calibrated with in-situ data(Gruber, 2012), or valid for sedimentary terrain only (Obu et al., 2019).", P2L25-26

The sentence concerned by this specific comment, has been changed to the following:

"The aim of this study is to move a first step towards a high resolution regional characterization of mountain permafrost in Greenland. To do so, we focus on the Sisimiut area, (68° N on the west coast), where we have a relatively large amount of data.", P1L55-P2L57

[P2 L49] Furthermore, you write in chapter 3.2.6 that the analysis of permafrost evolution would be a goal. Does that also apply here?

This goal is now explicit as following:

"Finally, we model future evolution of permafrost distribution in the area using scenarios RCP 2.6 and RCP 8.5, observing a relevant permafrost loss.", P3L69-70

[P2 L50] perhaps rephrase? In general, it would be nice if accurate weather data were available, but if not available there is a need for alternatives I suppose. Furthermore, instead of writing "a large number" just write 28 loggers. I am not sure if 28 is really a large number.

Sentence changed as following:

"In fall 2020 we installed 28 surface temperature loggers in the area measuring Ground Surface Temperature (GST), covering the local range of elevations and aspects.",P3L57-58

[P2 L53] difficult sentence. "allowing creating" please rephrase

Sentence changed as following:

"The statistical model is then used to compute time series of GST at any location in the landscape and for the period 1850-2022.",P3L60-61

[P2 L41] Noetzli! please correct throughout the document and in the bibliography.

accepted, P2L37, P26L587

[P2 L31] what is the difference between this and your study? You also criticize later that the borehole data are not representative?

Partly of this point can be referred to [M4]. In addition to that, notice that previous studies do not have data describing variability due to local topography nor bedrock conditions. This is because they do not focus on mountain permafrost, but rather on large scale sedimentary permafrost patterns. In our study, we have the GST network covering the local range of elevations and aspects, boreholes drilled in bedrock, and high elevation geophysical data. This allow us to increase the model resolution from 1 km (Obu et al, 2019) to 10 m, which describes the spatial variability of mountain permafrost and is more suitable for the purpose of mapping permafrost in a small area. We made this concept more clear through several modifications. In particular at:

- "Available models are based on numerical simulations at kilometer scale (Daanen et al., 2011), are not calibrated with in-situ data (Gruber, 2012), or valid for sedimentary terrain only (Obu et al., 2019).", P2L25-27
- "A major challenge when modeling mountain permafrost in this region is due to data availability, as ground temperature data are limited to few low-land sedimentary boreholes that are not representative for higher elevation and complex terrain (Obu et al., 2019). A common strategy to overcome this issue is based on the approach developed in Switzerland in the early 2000's (Gruber et al., 2004) relying on a network of permanent surface temperature loggers", P2L32-35
- "The aim of this study is to move a first step towards a high resolution regional characterization of mountain permafrost in Greenland. To do so, we focus on the Sisimiut area, (68° N on the west coast), where we have a relatively large amount of data. In fall 2020 we installed 28 surface temperature loggers in the area measuring Ground Surface Temperature (GST), covering the local range of elevations and aspects.", P2L55-58
- "We test our model for 1D simulation, which we compare to temperature data obtained by two 100 m deep boreholes drilled in bedrock at low elevation in 2019 and 2021. To obtain field data on ground temperature at high elevation, we used the approach proposed by Duvillard et al. (2020) based on geophysical surveys and calibration of resistivity - temperature dependencies in laboratory experiments.", P3L64-67

[P3 L62] repetition of words, please rephrase
Sentence changed as following:
"Our study site is located in the mountains surrounding Sisimiut, a city on the coastline of the widest non-glaciated area in West Greenland, about 200 km from the Greenland Ice sheet (see Fig.1).", P3L73-74

[P3 L68] additional references should be added for these points. The evidence is not optimal like this. Temperature means are given for the period 2000-2020, precipitation means for 1961-1990, without reference. No sources are provided for temperature trends either. The information that they have not changed precipitation needs to be substantiated. Furthermore, it would be interesting to know if precipitation has remained constant as an annual average, or if there may have been seasonal variations in precipitation.
Following the comment, the sentences have been rephrased as following:
"Climatically, Sisimiut is located in the low arctic oceanic area, and weather data are recorded at the airport weather station (Cappelen et al., 2021; Cappelen and Jensen, 2021). The warmest month is July (6.3 °C on average), while the coldest is March (-14.0 °C). Mean annual air temperature increased from -3.5 °C in 1961-1981 to -1.8 °C in 2000-2020. Mean annual precipitation decreased from 509 mm in 1961-1981 to 422 mm in 1984-2004, year in which the rain gauge was decommissioned. Decrease in precipitation concerns both solid (mean monthly precipitation in January-April decreased from 28 mm in 1961-1981 to 25 mm in 1984-2004) and liquid (mean monthly precipitation in June-September decreased from 58 mm in 1961-1981 to 49 mm in 1984-2004) precipitation.", P3L81-87

[P3 L77] In my opinion, you could either increase the information content here or delete the sentence
Sentence deleted

[P3 L76] Can you mark the block glaciers in the map / figure 1? Are there any references to the activity of the block glaciers?
Rock glaciers are now marked in the figure. No, these rock glaciers have not been investigated so far, so there no references. The activity is judged by us using the morphological characteristics of the front.

[P3 L75] sentence structure, please rephrase
Sentence rephrased as following:
"This climate locates Sisimiut in the sporadic permafrost zone (Obu et al.,2019; Biskaborn et al., 2019), and morphologically active rock glaciers are present in the area, reaching sea level elevation", P3L87-88

[P3 L80] The content of the chapter refers primarily to the calibration of the loggers and therefore does not fit the title. Maybe change the general structure of the subsections in chapter 3? Maybe instead of the subdivision into Field data and Modeling, Temperature Data, Geophyiscs and Modeling?
The strucutre of chapter 3 has been changed according to the comment

[P3 L65] Table 1 lists that 11 loggers were installed in the soil. Can you provide any information about the substrate other than the bedrock? Is there block cover, soil, and what are the conditions? I think especially with reference to heat fluxes it would be important to know the ground cover.
This is now specified as following:
"In soil conditions, loggers where placed in 50 mm holes manually dug in gravel.", P5L118-119

[P3 L56] Please provide a reference for COMSOL (COMSOL Multiphysics Simulation Software?) and rephrase the sentence.
Sentence rephrased as following:
"In this study, we use COMSOL Multiphysics® heat
transfer module, connected to Matlab through LiveLink (COMSOL Inc., 2015).", P3L63-64

[P4 L86] abbreviation has not yet been introduced
We now adress it here as "geophysical profile", P4L88

[P4 L86] ground surface temperature sensors?
Referred now as "GST sensors", P4L88

[P4 L86] is shown is used for a-c. Maybe rephrase
Rephrased as following:
"Map of the entire study area (a), with location of deep boreholes SIS2019-02 and SIS2021-01 and rockglaciers fronts, identified by green stars. Detail of the Nattoralinnguaq area, where most of the GST sensors are installed (b). South face of Nattoralinnguaq and Miguttunguup Qulaa (picture taken from Sisimiut in October 2020) with GST loggers and geophysical profile locations (c).", P4L88

[P4 L90] ?
Now deleted

[P4 L91] Maybe call this chapter GST Monitoring?
Now called "Ground temperature monitoring", to include borehole in the same section as proposed in P3L80.

[P4 L97] maybe split sentence?
Sentence splitted and rephrased as following:

"We established a GST monitoring network consisting of 28 individual monitoring locations, covering as evenly as possible the range of aspects, elevations and slopes at the study site. Data were acquired for one year, from fall 2020 to fall 2021.",P5L101-102

[P4 L94] Please support this statement with additional or more appropriate sources. Gubler et al. (2011) deal with iButtons in detail, but in Gruber et al. (2004) I could not find any reference to iButtons nor to M-Log 5. Thus, references to M-Log5 are completely missing.

Sentence rephrased and references added:

"Both iButtons and Geoprecisions are widely used in permafrost studies and the community has previous experience in their strength and weaknesses (Gruber et al., 2004; Gubler et al., 2011; Magnin et al., 2015a, 2019; Hipp et al., 2014; Schmidt et al., 2021; Duvillard et al., 2020).", P5L103-105

[P5 L96] The information on the data loggers is difficult to follow. There are several points here.
- The Geoprecision loggers could be named more clearly (M-Log5 (W?) Rock?).
- Is it correct that TNode sensors are installed? They are listed on the Geoprecision page for the ThermistoreStrings, while PT1000 sensors are listed for the M-Log 5.
- The use of temperature resolution and uncertainty (chapter 3.1.1) is not quite clear to me. Here it would be very useful to list the accuracy as well. This is clearly different between M-Log and iButtons. In this respect, I cannot quite understand why the uncertainty can be better than 0.02 °C when the accuracy of the iButtons is +/- 0.5 °C according to the manufacturer.

Thanks for the comment; we agree to modify the table accordingly. To answer the specific questions:
- We will name the loggers correctly, according to manufacturer: M-Log5W-Rock and M-Log5W-String
- Good point. TNode sensors are installed only in the M-Log5W-String. PT1000 are installed in M-Log5W-Rock
- We add the resolution and accuracy as declared by the manufacturer.

The table has been modified according to the comment, P6L133

[P5 L102] Can you provide information on how the loggers were installed? Was drilling done? How were the iButtons installed in the soil at a depth of 5 cm, how in the Bedrock?

This information is now added as following:

"These loggers were placed by drilling a 10 x 300 mm hole and sealing the sensor using frost resistant resin. For The iButtons (19 loggers in total), were installed in other more accessible conditions, such as flat bedrock (6), soil (11) and easy-access rockwalls (2) (See Fig. 1). In bedrock conditions, loggers were placed in 22 x 100 mm holes, sealed with a mixture of sand and frost resistant sealant. In soil conditions, loggers where placed in 50 mm holes manually dug in gravel.", P5L115-119

[P5 L110] Information about the HOBO logger, regarding accuray and resolution is missing. In my opinion, it would improve the overview if the technical information about all loggers or temperature sensors were prepared together and illustrated in one table. Then a chapter for the installation and positioning of the loggers could be written.

See main answer [M4]. The table has been modified according to the comment, P6L133

[P6 L120] Only one ERT transect was measured? Rename chapter with reference to ERT?

Chapter renamed as "Geophysical data", P6L134

[P6 L123] what infrastructure elements other than the path were relevant? Were there possibly also scientifically relevant reasons for the site selection?

This is now specified as following:

"This summit presents typical characteristics of the mountains in the Palasip Qaqqa– Sammisoq ridge: a steep and rocky south face approximately 100 m high with a debris slope underneath, and a more gentle north face characterized by small vegetation patches and some short steeper sections. This specific mountain was chosen for its accessibility, as the road leading to the airport passes just nearby a short path that leads to a popular viewpoint to the summit.", P6L137-145

[P6 L122] These informations would be interesting in the study site description

This is a specific summit within a mountain range. We therefore provide in the study site description the description for the typical mountain characteristics:

"The mountains of the region typically have pyramid-shaped summits and steep rockwalls generating debris slopes underneath. Mountains are dominated by bedrock, although vegetation patches are common at up to 400 m.a.s.l.", P3L78-80

While here we state that this specific summit "[...] presents typical characteristics of the mountains in the Palasip Qaqqa– Sammisoq ridge: a steep and rocky south face approximately 100 m high with a debris slope underneath, and a more gentle north face characterized by small vegetation patches and some short steeper sections.", P6L137-140

[P6 L129] Please rephrase, sentence is difficult to understand. Especially as there is no background on ERT the 4-point setup of the Wenner Array is difficult to understand. Furthermore you could add further Information like the Voltage you used

We simplified the sentence:

"TheWenner configuration was used because of its best signal-to-noise ratio in complex environments (Dahlin and Zhou, 2004; Kneisel, 2006).", P7L148-149.

The use of the Wenner protocol in complex environments is very common, and we suggest to not enter the details of this methodology as we are presenting it from the user's perspective only. We used a voltage between 0 and 600 volt. We suggest not adding this information as it is not useful

[P6 L133] pleas provide background on the filtering

Sentence rephrased as following:

"We cleaned 4% of the data point acquired before the inversion (549 points acquired, 528 inverted) by filtering out the outliers from the pseudo section.", P7L150-152

[P6 L136] Third iteration sound quite early to me. Please provide information on the convergence criterion

The data here are of good quality and the convergence criterion was taken from having the difference between the data misfit value below a prescribed threshold.

[P6 L134] sentence structure - based on? What do you mean with the pseudo section?

A pseudo-section is a depth representation of the measurements (apparent resistivity using Ohm's law). For each quadrupole, we can define a pseudo-depth from the distance between the electrodes. The data are shown on a cross-section called a pseudo-section. Sentence rephrased as shown in P5L133

[P6 L131] sentence hard to follow, please rephrase.

Sentence rephrased as following:
"Topography was extracted from a 2 m resolution digital elevation model (DEM, Porter (2018)) based on electrode positions measured with a handheld GPS device.", P7L149-151

[P6 L132] What is the spatial resolution of the DEM?
2 meter, now added to the text, P7L150

[P6 L138] `two rock samples
accepted, P7L155

[P6 L140] were
please rephrase sentence
Sentence rephrased as following:
"The three granite cubic core samples considered for laboratory analyses (sample G-RF, G-LR and G-DA) are characterized by a porosity of $\Phi$ = 0.032 for G-RF, $\Phi$ = 0.015 for G-LR and $\Phi$ = 0.023 for G-DA." P7L156-158

[P6 L141] were
accepted, P7L158

[P6 L141] for 24 hours
At what Temperature?
Sentence rephrased as following:
"Before performing the laboratory measurements, the samples were dried for 24 hours at 60 °C, then saturated under vacuum with degassed water from melted snow taken in the field.",P7L158-159

[P6 L142] were
accepted, P7L159

[P6 L143] repetition of words (previous sentence)
Sentence rephrased as following:
"The samples were then left several weeks in the solution to reach chemical equilibrium", P7159-160

[P6 L145] unclear information. Measurements were conducted under room temperature, after samples were removed from the freezer?
See [M6]

[P6 L146] please provide background on frequency
This is the frequency of the electrical field used to do frequency-domain induced polarization.

[P6 L146] sentence hard to follow, maybe rephrase.
Sentence rephrased as following:
"We moved the freezing point temperature TF = 0 °C based on direct observations on instrumented boreholes for G-RF. The measurements with TF = -3 °C reflect the fact that the measurements were made only in the downward direction of the temperatures and not in the upward direction", P6L166-169

[P6 L148] This sounds like the aim of the Lab Analysis, maybe move it to the beginning of the chapter?
 Accordingly, we moved the sentence to P7L156-157

[P7 L152] please rephrase
 Sentence rephrased as following:
"The methodology evaluates Ground Surface Temperature (GST) time series with an empirical approach, which are then used as upper boundary conditions for a heat transfer numerical model.", P7L172

[P7 L167] its
 accepted, P8L189

[P7 L163] from
 accepted, P8L185

[P7 L149] the information in the text and in the table on the data basis is somewhat confusing. Perhaps you could make this a little clearer and optimize the table.
- It is not immediately clear in the table that dataset a already consists of data from two weather stations.
- It is not obvious without reading in the text, why for datasets a, b and c temperature data for the period until 2021 are available, but only until 1979 are used
- it is understandable that reanalysis data are used to include non-measured parameters. But, why are the temperature data measured in Sisimiut not used anyway?
- if I understand it correctly, the merged data set c includes the solar radiation from d? If so, this information should be visible in the table.
 We improved the table accordingly to the comment, P8L179.
 In particular:
   • We changed the datasets names, giving priority to the datasets directly used in the modeling.
   • We highlight that dataset a is generated from datasets f and c. Datasets f are used only to generate air temperature for dataset a, therefore they are described separately from the datasets directly used in the modeling.
   • We use as much as ERA5 data as available, since they available in the whole region. This helps us to understand the model performance with generic data available everywhere in Greenland

[P7 L152] published in 2022
 accepted, P7L172

[P7 L160] 2021?
 accepted, P8L182

[P7 L163] can you provide clear references for the Copernicus database and the CryoGrid SEB?
 References added: Hersbach et al, 2020; Westermann et al, 2016, P8L186-187

[P7 L165] Earth
 accepted, P8L187

[P8 L175] Three times PISR in one sentence. Could you try to rephrase?

Sentence rephrased as following:
"Solar radiation forcing is downscaled using the ratio between the PISR at each logger location and the PISR at the ERA5 reference grid", P8L198-199

[P8 L177] abbreviation has not yet been introduced. could you insert the reference to Conrad et al. (2015) immediately after SAGA to make the context clearer?
The reference Conrad et al. (2015) refers to the PISR module, not to the software SAGA. The sentence has been rephrased as following:
"The PISR map is evaluated using the software System for Automated Geoscientific Analyses (SAGA) and the module PISR (Conrad et al., 2015).", P8L199-200

[P8 L180] Partial repeat to sentence in line 175. Can this be reworded and summarized more specifically?
Formula has been removed as it does not bring substantially new information. This paragraph has been included in a dedicated section "Snow cover modeling", P9L215.

[P8 L187] Could you provide additional information? How many Pictures, how many Winters? What Year and Month? Several Pictures per Winter?
Furthermore, it should be "by interpreting" and "in a GIS"
Sentence rephrased as following:
"This dataset is created by interpretation of 5 landscape pictures of the study area taken at the peak of the snow accumulation season (late April) in winters 2021 and 2022. We manually assigned snow/no snow areas in a GIS and combined this dataset to terrain parameters to train a binary classifier.", P9L222-224

[P8 L183] Considering the fact that snow thickness has significant influence on the ground thermal regime, is there also information on the thickness of the snow cover?
No, we do not have that information, as we found very difficult to model snow depth distribution at the landscape scale. This is due to the combination of strong winds, relatively shallow snow cover and rugged terrain. For this reason we use only probability of presence/absence, under the hypothesis that this is a proxy for snow depth too. This hypothesis finds support by the fact that snow depth tends to be consistent across seasons in the arctic, due to the dominance of strong winds in spatial distribution patterns. This concept is now presented as following:
"The classifier provides a probability of snow cover for a given set of curvature and slope, which can be extrapolated at the landscape scale creating a map of snow cover probability SnowP (Fig.2). The map identifies drift traps where snow is most likely to accumulate. The validity of this method is based on the observation that snow drift patterns in the arctic are generally stable over time due to relatively dry and windy weather (Parr et al., 2020)." ,P9L225-228

[P8 L192] 2022
accepted, P9L205

[P8 L198] logger time series?
Sentence rephrased as following:
"We then split this database into training and validation sets, following a pseudo-randomized cross validation approach, as we randomly exclude entire GST time series from training.", P9L212-214

[P8 L199] allows the observation of

Sentence removed as it does not provide substantially new information

[P8 L200] THis is difficult to understand. Could you rephrase and provide additional information?
This sentence is misleading and does not provide substantially new information. We removed. The relevant information is now given by:
"While GST data from the period 2020-2021 are used as dependent variable, the climatic datasets overlapping on the same periods are used as predictors. Datasets a, c and d (see Table 2) are fitted to the GST data using air temperature and solar radiation as predictors. For dataset b, we also use cloud cover, dew point temperature, total precipitation, wind speed and direction as additional predictors.", P9L208-211

[P9 L202] snow drift?
accepted, P10L232

[P9 L202] maybe include a) and b) or top and bottom for clarification?
accepted, figure updated, P9L232

[P9 L202]  1 and 2 partly cover interesting parts of the picture. Could you move the dots a little?
accepted, figure updated, P9L232

[P9 L206] please rephrase
Sentence rephrased as following:
"The heat transfer process is modelled using the "heat transfer in porous media" module in COMSOL, which assumes the local thermal equilibrium hypothesis to be valid", P9L231-232

[P9 L212] what do you mean with subsurface? Could you be more specific?
Sentence rephrased as following:
"For the 1D model we entered a custom function to describe the matrix density, which was evaluated from the cores extracted from SIS2021-01, providing an empirical function of depth increasing from 2600 kgm−3 at the surface to 3000 kgm−3 at 20 m depth, and being constant until 100 m depth", P10L238-240

[P9 L209] In chapter 2 you write that predominantly amphibolitic gneiss occurs in the study area. In 3.1.5 you also give values for porosity. Can you prove that the values assumed for granite are really applicable for the occurring rock?
We tested the sensitivity of our model to these parameters, by imposing different values typically found in igneous rocks (see Cermak et al,  1982, doi:10.1007/10201894_62). The results indicate that the model is not very sensitive to these parameters, as we obtained ground temperature differences below 0.01 C below the depth of annual amplitude. The most sensitive parameter is porosity - as suggested by previous studies, e.g. Noetzli and Gruber, 2009. For this reason we calibrate porosity and leave other parameters as COMSOL default.  This is now summarized in the following sentence:
"quick sensitive analysis shows that the model computes ground temperature differences smaller than 0.01 °C when varying these parameters within the typical ranges of different crystalline rocks.", P10L236-238

[P9 L212] Why do you use different parameters for 1D and 2D Models? If values increase from 2600 to 3000, why do you apply 3000 as the average?

The argument here is the same as above at line 209; the model is not sensitive to this parameter. Since for the borehole we have a precise quantification of density, we use it. However, we are not sure how this is representative for the rest of the study area. This is now summarized in the following sentence:
"We attributed the constant density of 3000 kgm−3 to the 2D models, as it not known if the near-surface values measured at SIS2021-01 are representative for the entire study area.", P10L240-241

[P10 L225] ...carried out by...?
 accepted, P11L253

[P10 L225] sentence is hard to understand; please consider rephrasing.
 Sentence rephrased as following:
"The calibration is carried out by simulating conditions in SIS2021-01, from 1850 to 2022 using a 1D geometry of a 100 m column. The simulation results are then compared to the field data acquired during the period August 2021 to April 2022.", P11L253-255

[P10 L229] Why did you test different porosity values instead of using the porosity you presented in chapter 3.1.5?
  The values presented in 3.1.5 are within the range we tested (0.01 to 0.05). However, we did not use the porosity values in 3.1.5 because they relate to surface samples and are extremely variable (0.012 to 0.046). The aim of this calibration is to find optimal porosity valid for the overall borehole depth.

[P10 L234] than modelled
 accepted, P11L264

[P10 L245] please rephrase
 Sentence rephrased as following:
"For this reason, we compute 2D model for two location of special interest: the ERT profile and the Nasaasaaq summit.", P12L278-279

[P10 L245] observing permafrost evolution implies long-term measurement series, whereas model data are the data basis here. Please consider rephrasing
 Sentence rephrased as following:
"The first location is chosen to compare the ERT data to our model, while the second location allows us to model and understand permafrost distribution and evolution in the tallest mountain in the study area.", P 12L279-281

[P10 L246] This sentence is hard to understand. Please rephrase and include information on what is meant by "profile z" and by "a solid"
 Sentence rephrased as following - considering that "profile z", and "solid" are not necessary information - :
"For each location we set-up a north-south transect in the QGIS software, and used it to sample the elevation profile from the DEM. The elevation profiles are then imported into COMSOL as 2D geometry using the parametric function option.", P10L246-247

[P11 L248] as above, what is meant by "the solid"?

Solid is the generic COMSOL name for 2D-3D geometries. We avoid using this term now, and use 2D geometry instead.

[P11 L249] Please note that formulas and symbols in the text are used consistently and according to TC's specifications.
Formula updated following the TC's specifications, P12L285

[P11 L251] from
accepted, P12L286

[P11 L254] The
Sentence deleted and calibration results included in the Methods,  P5L111, P5L106

[P11 L253] I would suggest that the calibration results be placed within the methodology in Section 3.1.1, as it is not a central result of the study, but only the calibration of the instrumentation.
Calibration results included now in the Methods,  P5L111, P5L106

[P11 L254] Please use terms consistently; iButtons or DS1922-L or a combination of the terms.
iButtons used now consistently through the text

[P11 L254] have on average an absolute temperature offset?
Sentence deleted as information is not considered relevant. We provide instead the maximum measured offsets.

[P11 L260] low  or maybe only: Temperatures in both boreholes ... close to 0°C
We have rephrased the whole paragraph:
"Boreholes temperatures are shown in Fig. 4. In SIS2019-02, the depth of zero annual amplitude is approximately 20 m. Below this depth, temperature data indicate a minimum of temperature of +0.3 °C, reached at 30 m depth, and a temperature of +1.0 °C at 100 m. Since temperatures are positive below the depth of zero annual amplitude, the measurements at SIS2019-02 indicate absence of permafrost. SIS2021-01 shows consistently negative temperatures between 20 and 70 m depth, reaching a minimum of -0.2 °C at 30 m depth. The depth of zero annual amplitude is approximately 10 m.b.g.s.. Since we measure negative temperatures below this depth, the data from SIS2021-01 indicate the presence of permafrost.", P12L301-306

[P11 L260] Information on measurement time points and measurement frequency should be mentioned in the methods chapter
Information now moved to the methods:
"Borehole SIS2019-02 does not have a permanent sensor installed, and it was logged manually three times since it was drilled.", P5L126-127

[P11 L261] Please show Data for both boreholes!
Data now included in Figure 4b, P14

[P11 L261] Based on these statements, it is impossible to assess whether this is truly an active layer, and whether permafrost is present at all in SIS2019-02.

Good point about the incorrect terminology, we cannot talk about active layer here. This is changed to "depth of zero annual amplitude", Also, see [M3]

[P11 L262] To be precise, data for a period of at least 2 years would have to be available in order to be able to speak of permafrost and an active layer at all. Please provide data!
See answer [M3]. Borehole data are now presented in figure 4b, P14.

[P11 L263] What ist the minimum temperature at both boreholes?
The minimum temperature is in the text; +0.3 C for SIS2019-02 and -0.2 C for SIS2021-01. We now specifiy that these are minimum temperatures measured below the depth of zero annual amplitude - see answer P11L260

[P11 L264] Please provide data!
See answer P11L261

[P11 L259] 4.2.1? and Borehole Temperatures - just to be consistent with the following chapter?
Section merged with 4.3.1 in the new section 4.1, P12L289

[P11 L259] This chapter needs to be revised. Please present data and review the use of the terms Active Layer and Permafrost.
See answer [M6]. We now use the terms "frozen ground" and "depth of zero annual amplitude"

[P11 LNaN] 4.2.2?
See answer P11L259

[P11 L266] All loggers used recorded data for a full year, from fall 2020 to fall 2021.
Sentence rephrased as following:
"GST data are measured during one full year, as loggers were installed in September-October 2020, and retrieved one year later", P12L290

[P11 L266] Could you provide more precise dates than just from fall to fall?
Loggers were installed during September 2021 at across a period of 2 weeks. For each logger we waited exactly one year to retrieve the data. We consider reporting precise dates here too consuming. Precise dates can be found in the annex data.

[P11 L266] 28? As mentioned above?
Numbers corrected, and now reported as following:
"Fifteen loggers present now-free GST data (Fig. 3a), seven present thick snow cover and six present intermediate characteristics (Fig. 3b).", P12L291

[P11 L266] what are snow free data? Please rephrase.
GST data from a logger that is snow-free. Now rephrased as in answer P11L266

[P11 L267] the reference to figure 3 does not match the contents of the sentence. But in the following sentences a reference to figure 3 (a-d) would be helpful.

Figure 3 updated accordingly, P13. References to subfigures a-d are now included in the text, P12L290-300

[P11 L268] for the south facing rockwall?
Sentence rephrased as following:
"To show the effect of elevation, we compare two snow-free loggers installed on south facing rockwalls, one at sea level (MAGST = +3.5 °C) and at 460 m.a.s.l. (MAGST = +1.2 °C).", P12L297-298

[P11 L272] For downscaling the value of +1.58 °C was assigned to...
The sentence was rephrased as following:
"We used this value of +1.58 °C, comparable to previous findings in Greenland (Rasmussen et al., 2018), as constant offset when modeling snow cover.", P14L323

[P11 L270] delete
accepted, P13L320

[P11 L271] what conditions did change and how did they affect data? Are these effects taken into account later on?
This is an average effect, as now clarified in the methods:
"We first evaluate the average temperature offset due to snow cover by comparing the mean annual GST of sensors that were/were not snow covered during the entire winter 2020-2021.", P9L216-218
Topographical conditions are later taken into account using the SnowP method:
"This offset then is multiplied to the local probability of snow cover presence/absence, that we call SnowP, varying from zero (absence of snow cover) to 1 (presence of long lasting snow cover).", P9L218-219

[P11 L265] This chapter is primarily concerned with the correction of MAGST data as a function of snow cover. The presentation of the actual surface temperatures comes somewhat short. Furthermore, a distinction is made with respect to the snow cover thickness, but not with respect to the snow cover duration. An assumption would be that in areas with thick snow cover, it also takes longer to reach the aperture, so that a more differentiated approach to MAGST could be useful? If the temperature data are to be used exclusively as input for the model and not for an independent interpretation, the question would still be whether the paper should be fundamentally restructured.

Very good comment that points out how we need a proper introduction of the snow modeling issue (see [M1]). In particular:
- We have separated methods for the evaluation of snow cover influence on the GST time series from the GST model - see P9L215
- Yes, we do not differentiate snow cover duration under the assumption that snow cover thickness is a proxy for duration. We may use a sophisticated approach for evaluating the double effect of duration and depth, but we would not be able to extrapolate these properties at the landscape scale (and for longer time periods). The offset method is used as a proxy for both processes, providing a first order approximation of the effect of snow cover on the GST. We now specify that "The validity of this method is based on the observation that snow drift patterns in the arctic are generally stable over time due to relatively dry and windy weather (Parr et al., 2020).",P9L227-228. We evaluate the quality of this method by comparing data model at our target: temperatures below the depth of annual amplitude from boreholes with different snow conditions : "Here, the

model has a maximum error of 0.15 °C when compared to SIS2019-02, which is used only as validation dataset. SIS2019-02 has also different snow conditions than SIS2021-01, suggesting that our SnowP map and offset provide an acceptable boundary condition to model long term effect of recurrent snow cover induced by topographical patterns"., P19L396-P20L397

- We are not sure what is meant by "independent interpretation"; we assume this means "model validation". If this is the case, we highlight how data are used for both training and validation. In particular, we use cross validation on the GST modeling, and borehole/ERT data for validation of deep ground temperatures. This is now more clear throug the discussion (e.g., P17L380, P19L396, P20L399)

[P11 L275] reference to chapter 3.1.5 (methods)?
We need to reject this, otherwise for consistency we would have to reference every method section while presenting the relative results.

[P11 L273] This chapter could also be revised in some places and the data presented a little more comprehensively. Among others, figure 4c talks about potentially thawed and transition zone. Are these seasonal effects or long-term permafrost degradation? The question whether on this basis one should speak of permafrost or rather only of "frozen ground" would have to be answered.
We added the following sentence:
"This indicates that permafrost presence is restricted inside the mountain and close to the surface in the north face and the upper part of the south face.", P13L311-312.
See answer [M4] about seasonal effects on permafrost.

[P12 L276] skalierung der y-Achse könnte. scaling of the axes could be identical. What does the data look like for a logger with intermediate snow conditions? What ist the elevation and exposition?
See answer [M9]

[P12 L276] could you color code the data points to relate to the snow conditions?
Figure 3 updated accordingly, P13

[P12 L276] please increase font size of the axis lables a little bit.
Figure 3 updated accordingly, P13

[P12 L276] please increase the font size of the axis labels
Figure 3 updated accordingly, P13

[P12 L276] the caption is difficult to follow. In particular, it is not clear to me whether Figure 4 a really refers to the measured field data or not the laboratory analyses. Please try to rephrase the text and make the connections a bit clearer.
Caption updated as following:
"Comparison between ERT and 2D thermal model . a) Petrophysical analysis, showing in-phase electrical conductivity data versus temperature for the three samples collected along the geophysical profile; b) profile of electrical conductivity/resistivity tomography (in Sm−1 and kΩm) measured on the field: c) resulting 2D numerical model of the ridge where the ERT line was conducted", P15L332

[P12 L278] can you clarify this statement? Is this an interpretation of the data? How is this lithological fault characterized and how does it affect permafrost and subsurface conditions? Here, however, the question arises whether this should be the content of the results chapter.

This sentence has been moved to the discussion and developed as following:
"Most of the disagreement between model and data is due to the electrical conductivity anomaly on the north face of the geophysical profile. This anomaly occurs near a large lithological fault, visible in the field. Overall, the observations indicate that ground characteristics at this location are not isotropic and a direct comparison between model and geophysics is not meaningful.", P20L400-403

[P13 L281] dieses Kapitel besteht zu einem großen Teil aus methodischen Hintergründen, die so auch nicht vollständig verständlich im zugehörigen Methoden-Kapitel erklärt wurden. Warum gehen nur Logger ohne Schneebedeckung ein?

The reason for this choice is explained in the method section:
"Here, we use snow-free GST data, as snow cover effects will be modelled in the next step. Previous studies used an offset-based approach based on the evaluation of a constant thermal offset between air temperature and snow free GST (Magnin et al., 2017a; Etzelmüller et al., 2022).", P9L202-205
In particular, snow cover loggers are not significant developing this modeling approach as their data variability is mostly influenced by the snow cover property rather than the terrain predictors. This is common practice while using this methodology. For ex Magnin et al 2019 : "snow accumulation over the logger would make the recorded RST unsuitable for statistical modelling because its thermal effect is not linear and depends on a variety of parameters that are highly fluctuating in space and time in high-elevated and high-relief environments, such as the snow thickness, duration and time period of the accumulation, the sun exposure of the affected rock face, and the snowpack thermal properties"

[P13 L284] 5?
Now figure 6, P13L318

[P13 L288] The argumentation is a bit difficult to follow. It would be useful to make a clear reference to the methodology (3.2.4?) right at the beginning, so that it is clear what 0.03 and "optimal porosity" should mean. If possible, this chapter should be restructured and phrased more clearly.

We now introduce the results with the following sentence:
"The model was calibrated by optimizing two parameters: matrix porosity and initial offset value.", P14L324.

[P13 L293] please rephrase and make clearer statements. "a few degrees" should be made more precise. The use of the term "active layer" should also be reconsidered in this passage.

Sentence rephrased as following:
"When tested and compared to SIS2019-02 (z = 55 m ; PISR = 690 kWhm−2, SnowP = 1), the model produces similar results (Fig.4b), indicating errors up to 2 °C above the depth of zero annual amplitude (20 m depth), while the errors are consistently smaller than 0.15 °C below this depth.", P14L329-331

[P14 L295] Would it be possible to use different color scales for the subsurface temperatures and temperature differences? I think in general this sentence can be deleted.

We'd rather have one color scale as the white color placed at 0 C, has a good visual effect in both ground temperatures (easy to identify cold zones) and in model evaluation (easy to identify good model performance). We will delete this sentence

[P14 L296] The presentation of the results is a bit short. The first 4 lines describe methodological aspects. In the second part, the basics of snow are given and already interpreted.

We extended the methodological description and avoided repetition here. Here we extended the results presentation including a description of the influence of snow cover:

"Snow cover plays an important role, as snow covered areas can increase of 250 m the elevation of the MGT20 0 °C isotherm. This effect is prominent on mountain flanks characterized by sequences of ridges and 340 chutes (Fig.7b), as the chutes are warmer than the ridges due to their predisposition to accumulate snow.", P16L338-340

[P14 L299] present?

Sentence now removed - see answer P14L296

[P14 L300] Figure 7

accepted, P15L334, P15L335, P16L340

[P14L301] here results and interpretation are mixed. Furthermore, statements contradict each other. First of all, one should not necessarily speak of continuous permafrost in the mountains, since topography plays a significant role in permafrost distribution.

Permafrost can be continuous in mountains. This starts at an elevation when permafrost is found also in the most unfavorable topographical conditions – e.g. south facing slopes, unfavorable snow accumulation. However, we agree that this may cause confusion since we are in arctic area, and will avoid this term. We now instead describe elevation limits of MGT20 relative to aspect and snow conditions. (P15L336)

[P14L301] Second, what is meant by "positive MGT20 (...) at which permafrost is continuous"? If permafrost exists, temperatures cannot be positive, can they?

Yes, this sentence is very confusing. Thank you for pointing it out. It is now changed to:

"Snow cover plays an important role, as snow covered areas can increase of 250 m the elevation of the MGT20 0 °C isotherm.", P16L338-339

[P14L301]  Furthermore, the importance of snow cover is again emphasized here. Maybe it would be useful to discuss the snow cover and its effects on permafrost in the introduction and to give references?

We have now dedicated proper space to the snow cover issue in the introduction, P2L45-54

[P14 L301] The evidence here is not quite clear, in areas with thick snow cover permafrost does not exist? The relationship between snowpack and permafrost is highly dependent on thickness, timing of snowing in and melting out. This should be made clearer (but not in the results section).

No, our data indicate that, on average, snow has a warming effect on the ground. When we apply this effect to our model, permafrost is found at higher elevation when snow is present when other conditions (elevation and aspect) are equal. This is clarified by:

"On average, the presence of snow cover causes an offset on the MAGST of +1.58 °C ± 0.41 °C when other conditions do not change (R2 = 0.81).", P13L320-P14L321

Yes, we agree that the relationship between snowpack and ground temperatures is complex, but we do not aim to model these interactions. As specified previously, we use a simple offset method based on empirical data to evaluate the average effects of snow cover on deep ground temperatures. This is now clarified already in the introduction, P2L45-54

[P14 L303] Sentence does not seem complete to me
 Sentence rephrased as following:
"The colder MGT20 occurs on the north faces of the Nasaasaaq peak (763 m.a.s.l. ), reaching -4.0 °C.", P16L337-338

[P14 L305] this headline is difficult to understand. Please consider rephrasing
Furthermore, the presentation of the results is short and there is extensive interpretation.
 Headline rephrased as following:
"Comparison between 2D model and ERT profile", P16L341
We now clarify that we are describing model results rather than interpreting ERT data:
"The model indicates, as of October 2020, the presence of negative temperatures below the depth of annual amplitude on the Nattoralinnguaq summit, suggesting the presence of permafrost.", P16L347-348
We removed interpretation from the section and moved it to the discussion:
"Most of the disagreement between model and data is due to the electrical conductivity anomaly on the north face of the geophysical profile. This anomaly occurs near a large lithological fault, visible in the field. Overall, the observations indicate that ground characteristics at this location are not isotropic and a direct comparison between model and geophysics is not meaningful.", P20L400-403

[P14 L306] Please review the use of the term "permafrost" in this context. Possibly simply speak of frozen ground?
 Sentence rephrased as following:
"The model indicates, as of October 2020, the presence of negative temperatures below the depth of annual amplitude on the Nattoralinnguaq summit, suggesting the presence of permafrost.", P16L342-344
For more detail, see answer [M3]

[P14 L306] Is there an ERT2? Can you refer to data/images early on?
 Sentence rephrased as following:
"In Fig.5c is presented the 2D model simulation at the geophysical profile location.", P16L342

[P14 L308] this is speculative and interpreted. Maybe something like "the model indicates temperatures are positive down to depths of 20-40m"?
 This is a pure description of the model results. We further clarify this aspect now, see answer P14L305

[P14 L308] Simply frozen? Or is there any information that temperatures have been below 0 °C for more than 2 years?
 See figure 9: the model shows negative temperatures since 1979.

[P14 L308] What figure do you refer to here?
 Proper reference is Figure 5b. Now corrected, P16L345

[P14 L309] frozen ground?

Here we talk about permafrost a we are evaluating deep ground temperatures. See [M3]

[P14 L310] interpretation
Sentence rephrased as following:
"Both datasets indicate a mostly unfrozen south face, and a colder north face. However, the ERT data indicate a large unfrozen section at the extremity of the north face.", P16L347-348

[P15 L316] 21st?
accepted, P17L352

[P15 L316] in the previous sentence you write, that permafrost will disappear by the end of the century.
Sentence rephrased as following:
"For scenario RCP 2.6, the ground seems in phase transition by 2100, being at $0.05 - 0.1$ °C between 50 and 70 meters depth.", P17L352-353

[P15 L318] RCP 2.6
accepted, P17L354

[P15 L318] RCP 8.5
accepted, P17L354

[P15 L323] permanently frozen?
Please reconsider using "permanently frozen" instead of "permafrozen" throughout the text. To my knowledge, the term permafrozen is not clearly defined and is not commonly used.
accepted, P17L359

[P15 L327] please reconsider the use of sporadic and continuous for mountain permafrost.
We now use elevation/aspect to describe permafrost limits:
"For Nattoralinnguaq, the optimistic scenario (RCP 2.6) suggests an increase of ground temperatures of about 1 °C, causing permafrost retreat on the north face up to 200 m.a.s.l.. Scenario RCP 8.5 delineates a situation where permafrost is relict, i.e. below the reach of seasonal frost (Magnin et al., 2017a), at approximately 100 m depth on the north face. The model produce similar results for Nasaasaaq, as for scenario RCP 2.6 we observe permafrost retreat to 300 m.a.s.l. on the north face and to 500 m.a.s.l. on the south face. Scenario RCP 8.5 indicates that all permafrost on the mountain is relict, except for the summit's north face.", P17L364-369

[P15 L330] what do you mean by "relict permafrost" here?
Below the reach of seasonal frost. This term is used by Magnin et al, 2017. This is now specified in the text, P17L366

[P16 L331] Please increase the font size of the axis label in Figure 8a. If possible, use comparable font sizes within the figures.
Accepted, updated figure 8, P18

[P16 L334] difficult to understand, please consider rephrasing.
Sentence deleted, as this information is already conveyed through the methods

[P17 L339] Your are citing the discussion paper from 2020 instead of the final version of 2021!
accepted, P17L376, P26L599

[P17 L341] CryoGrid 3
accepted, P17L378

[P17L344] Here I agree completely. Even though the content of the Conclusions should be clarified, to what extent the results based on this short series of measurements are accurate.
See answer [M4]. We now make clear that:
"On the other hand, our model is more reliable when describing ground temperatures below the depth of zero annual amplitude. Here, the model has a maximum error of 0.15 °C when compared to SIS2019-02, which is used only as validation dataset. SIS2019-02 has also different snow conditions than SIS2021-01, suggesting that our SnowP map and offset provide an acceptable boundary condition to model long term effect of recurrent snow cover induced by topographical patterns. The general agreement between model and geophysical data also indicates that the model is suitable for describing permafrost extents in a wide range of elevations, slopes and aspects.", P19L395-P20L400

[P18 L346] Sentence seems to be incomplete. What do you mean by this? Please consider rephrasing.
This sentence has been deleted, as not carrying relevant information for the discussion.

[P18 L353] Please rephrase. Far as I can tell, snow does not actively cool the ground, but affects the thermal regime, which can result in a cooling. Also, instead of "early heatwave" heatwaves early in the year?
This discussion on snow effects is too complex for the scope of the study and we decied to remove it. We now present the issue of snow cover and state of the arte of modeling in the introduction, P2L45-54

[P18 L348] This information could also be part of the introduction. Please formulate in such a way that a discussion of the results is made apparent.
Agreed. See answer P18L353

[P18 L359] How can you tell the boundary conditions are acceptable? Could you present data on this?
The sentence has been rephrased as following:
"Here, the model has a maximum error of 0.15 °C when compared to SIS2019-02, which is used only as validation dataset. SIS2019-02 has also different snow conditions than SIS2021-01, suggesting that our SnowP map and offset provide an acceptable boundary condition to model long term effect of recurrent snow cover induced by topographical patterns.", P19L396-P20L398
The data of SIS2019-02 are now presented in figure 4b, P14

[P18 L362] This section also sounds like it belongs to the introduction rather than to the discussion.
This is now shortened and integrated in the introduction:
"Although some numerical models are able to describe snow physics at hectometric resolution (Gisnås et al., 2014), it becomes extremely challenging to achieve a good knowledge of spatial characteristics of the snow in complex terrain, given the spatial variability of weather forcing, as wind and shading.", P2L48-51

[P19 L400] who use? / Where a similar approach is used?

accepted, P28L385

[P19 L402] please provide statistics.
"Significantly" has been removed now

[P19 L402] (xy °C at 10 m depth) - please add information about the average differences of your model
This is already presented in the results, P14L327-331

[P19 L404] no data on the quality of the climate data has been provided up to this point.
It is not our aim to provide an evaluation of these data, which can be found in the respective publications. In this study we observe how these data perform within our database and methodology. Here we mean to say that using reanalysis data may have an effect to the result quality near surface, which is lower than previous studies using weather station data.

[P19 L405] this is very fundamental criticism of the entire experimental setup, which is also completely justified. What is the justification that the results are nevertheless relevant and reliable.
See [M4]

[P20 L414] Here I find it difficult to follow the argumentation. Above, you criticize both the duration of the measurement series and the positioning of the boreholes. The boreholes from the shallow foreland were used to model frozen subsurface in the mountains and then you state that the results are satisfactory because the modeled temperatures could be validated with those of the neighboring borehole. Please consider optimizing the evidence a bit. I think the approach in the next sentence to include the ERT results is purposeful here.
See [M4]. Sentence rephrased as following:
"All this considered, the results indicate that the modeling approach is suitable for evaluating of ground temperatures below the depth of zero annual amplitude in complex terrain.", P20L403-404

[P20 L416] As noted above, I think the comments on the ERT results and the petrophysical data are a bit too brief. To lead this discussion, a somewhat more comprehensive data interpretation would be interesting.
The sentence was rephrased as following:
"Most of the disagreement between model and data is due to the electrical conductivity anomaly on the north face of the geophysical profile. This anomaly occurs near a large lithological fault, visible in the field. Overall, the observations indicate that ground characteristics at this location are not isotropic and a direct comparison between model and geophysics is not meaningful.", P20L410-403

[P20 L420] is it about permafrost characteristics or permafrost distribution?
It is about permafrost distribution. The sentence was rephrased as following:
"We can use our results to compare the distribution of bedrock permafrost in the Sisimiut area to other mountain ranges, as presented in Fig.10.", P20L408

[P20 L421] contents of the next chapter
Sentence rephrased accordingly:
"We can use our results to compare the distribution of bedrock permafrost in the Sisimiut area to other mountain ranges, as presented in Fig.10.", P20L408-409

[P20 L423] could you provide reference to the associated chapter or other sources? In Chapter 4.3.1 you write: "Aspect causes a MAGST offset of 2.2 °C " and "When snow covers the loggers, these quanitfications do not hold" How are these statements as well as the discussion here related?

Yes thank you for noticing, this is confusing. The 2.2 C refers to actual measurements of two specific loggers, while the 2.4 C is an average value produced by the model. We rephrase to avoid confusion:
"In Sisimiut, the solar radiation creates an average offset of 2.4 °C from north to south facing slopes, causing a rise of about 400 m of elevation in the permafrost 0 °C isotherm between this two aspects.", P20L410-412

[P20 L432] The content of this chapter relates primarily to the relationship between solar irradiance and permafrost distribution, rather than permafrost properties. In addition, the influence of the snow cover could also be included in the discussion.

Chapter renamed accordingly "Local permafrost distribution compared to other regions", P20L407.
As specified in the introduction, previous studies mostly disregard snow influence by filtering steep terrain as valid for the model. Therefore, direct comparison here is difficult.

[P20 L438] (...) outlast the 21st century?

accepted, P20L425

[P21 L447] This section focuses heavily on the relationship between permafrost degradation and rockfall activity, a topic that is very present in the introduction. For this, the statements here are very vague and in part simply the state of research is presented. In order to remain consistent with the introduction, it would be nice to make more concrete statements here. Does permafrost degradation only affect rockfall activity? Does the potentially longer lasting low-lying, then relict permafrost affect mass movements?

See [M1]. This paragraph is now only briefly touched in the conclusion:
"Although the correlation between permafrost degradation and rockfall activity is accepted within the scientific community (Ravanel and Deline, 2011; Patton et al., 2019), the process chain linking the two phenomena is very complex. Therefore, future efforts in the area should focus on investigating slope stability characteristics, and their relation to permafrost distribution and degradation. Once (and if) problematic slopes are identified, site specific models integrating high resolution snow distribution (Haberkorn et al., 2016) and crack networks (Magnin et al., 2020) will provide a more detailed understanding of slope thermodynamics, overcoming the main uncertainties of our model.", P21L443-447

[P21 L461] until the end of the century?

accepted, P21L439-440

[P21 L462] ..that only relict permafrost will persist at a greater depth (quantification?)...

Sentence rephrased accordingly:
"For scenario RCP 8.5, i.e. with no mitigation on carbon emissions, our model predicts a reduction of 95% of the active permafrost area at the end of the century, meaning that permanently frozen ground will persist only as relict condition below 10-20 m depth, i.e. below the reach of the seasonal frost.", P21L438-441

[P21 L463] suggestst

accepted, P21L441

[P21 L464] permafrost distribution and degradation?
  accepted, P21L445

[P21 L467] In my opinion, the preceding text does not really lead to this conclusion. The majority of the conclusions consists of summary and outlook. The circle to mass movements as a consequence of permafrost degradatino is closed, but less based on the research results than on general knowledge. For me, a clearer derivation to this final conclusion would be desirable. Is there a rationale that your simple approach is really a good trade-off? Especially considering your own criticisms of the data basis, it would be interesting to derive the conclusion in a slightly more complex way and show that despite the criticisms, the approach has advantages over even simpler approaches, but comes close to the informativeness of more complex projects.

  Yes good point, this is linked to [M4]. The main conclusion here is that we are satisfied with the model performance due to its ability to reproduce ground characteristics below the depth of zero annual amplitude. This is useful for permafrost zonation in a first assessment. This is now specified by:
"In this sense, our modeling approach based on weather parameters readily available for the whole region, downscalable with a simple topographical approach, provides a good first assessment for mountain permafrost zonation.", P22L448-450

[P21 L471] The link does not work yet.
  The link is activated after paper acceptance

---

## Referee Report (RR1)

The manuscript "Modeling present and future bedrock permafrost distribution in the Sisimiut mountain area, West Greenland" is the thoroughly revised version of the initially submitted manuscript "Characteristics and evolution of bedrock permafrost in the Sisimiut mountain area, West Greenland". In general, the authors managed to address most of my concerns formulated during the initial revision of the manuscript.

However, based on a detailed review of the revised manuscript I encountered the following main points of concern:

- Structure
  Although largely improved the structure of the manuscript is still not easy to follow in certain parts where, for example, the figures (and/or panels in figures) are not referenced in ascending order.
  Moreover, I still believe that merging the Results and Discussion sections would be beneficial for the overall structure of the manuscript; thus, making it easier for the reader to follow the extensive data, methods and results presented in this manuscript. In the current version, the connection between the various subsections of the Results and Discussion sections is not always clear, which strengthens the impression that some descriptions were split just for the sake of having separate Results and Discussion sections.

- Figures
  Some figures require a thorough revision considering that it is not clear, e. g., how the presented data (sets) were selected, not all elements of the plots are described (missing or incomplete legend), or the figure captions are not sufficient.

- Geophysics
  The authors present both a laboratory and a field geophysical investigation, yet in the current version of the manuscript the geophysical results appear to be somewhat disconnected from and can only partially explain the results obtained from the other methods/approaches.

- Units
  In some cases, wrong or no information about the units of measured/observed values is provided, e. g., Table 1 or for porosity values in general.

- Miscellaneous
  Throughout the manuscript, descriptions lack required details regarding data acquisition/processing approaches. This concerns the temperature logging and modeling as well as the geophysical investigations. Moreover, not all variables in equations presented in the manuscript are properly introduced or described.

I am still convinced that this manuscript is a valuable contribution that should be published in The Cryosphere. However, the open issues listed above demonstrate that further revisions are required before the manuscript can be accepted for publication. Since my comments also concern the general structure of the manuscript as well as the preparation of the figures I suggest that the manuscript should be reconsidered for publication in The Cryosphere after major revisions.

Attached I provide an annotated version of the revised manuscript with detailed comments and suggestions that might help the authors during the revision of the manuscript. The annotations use the following code:
- Highlighted (yellow): Needs or at least should be addressed/considered during the revision
- Highlighted (red): Remove.

[revised manuscript text omitted]

---

## Referee Report (RR2)

In their revised manuscript "Modelling present and future bedrock permafrost distribution in the Sisimiut mountain area, West Greenland", originally "Characteristics and evolution of bedrock permafrost in the Sisimiut mountain area, West Greenland", the authors use a minimalist approach to modelling spatial permafrost distribution and its future evolution. In my opinion, the revision of the manuscript has massively improved its quality, although some points still remain open. Since the changes were very extensive, a few more questions arose during the re-review of the manuscript, which are addressed in the following.

**Formal Apects:**

The reviewers' points of criticism regarding formal aspects have been incorporated very well for the most part. Nevertheless, there are some sentences, especially in the newly written sections, whose wording should be revised. I have made some suggestions here, but would recommend that the manuscript be critically proofread once again.

The illustrations have also been significantly improved. A few suggestions, or points of criticism, are given below in the line-by-line comments.

**General Comments and Fundamental Aspects:**

In the first review, I raised a few points that I consider fundamental, which were taken up by the authors but were not clarified in a way that was completely comprehensible to me. This concerns the discussion conducted under [M4], where I would like to take up a few points again.

*[M4] The authors conclude that their "modeling approach based on few weather parameters, downscalable with a simple topographical approach, provides a good trade-off between results quality and uncertainty". Even though I have to agree that the results look very promising and are graphically presented in an excellent way, I miss some basic information to evaluate the quality of the results. This concerns the representativity of the boreholes (Lines 404-405) as well as the input climate data (Lines 402-404), the length of the measurement period (Lines 344- 345) as well as the influence of the snow cover (Lines 348-361). Points that are addressed very critically by the authors themselves in the course of their discussion. However, justifications, why the results can be regarded as representative nevertheless, come somewhat briefly.*

> *[M4] Yes, this is a very good point. The discussion on the uncertainties is more substantial than the discussion on the validation, and the reader is left in doubt about the actual value of the study. We believe that a thorough description of uncertainties is necessary to develop further studies in the region, but we also believe that our results are valuable despite these uncertainties. The main point that we have now developed in the text are the following:*

- *Our heat transfer model reproduce deep ground temperatures, i.e. below the depth of zero annual amplitude, within 0.15 C for both boreholes (P14L331). This is key result as, thanks to ground thermal inertia, deep ground temperatures are influenced by climatic trends rather than short- term variability. In this sense, our heat transfer model has good performance in predicting deep ground temperatures despite the short measurement period. This suggest that the GST model has good coupling with the available weather data.*

I don't want to appear cynical at this point, but I want to be deliberately provocative; does this mean that the heat-transfer model works in the depth ranges where heat transfer no longer takes place? That is, below the zero annual amplitude? And, isn't it the area above zero amplitude that is relevant for permafrost dynamics in the coming decades and for potential mass movements? Of course, climatic trends are relevant for long-term permafrost development and not short-term dynamics. Nevertheless, seasonal effects in particular are important for these dynamics. Since a central point of the paper is to test the uses of this minimalist approach, a much more comprehensive evaluation and discussion of these points (including the database on climate and weather data) would be very relevant to me.

(You can also find some comments on this point in the Line-by-Line Comments).

- *These boreholes have similar elevation/aspect, but substantially different snow conditions. Using our snow modelling approach, we manage to model deep ground temperatures accurately. Although we miss the seasonal variability induced by the snow-ground complex interactions, this method is suitable for the goal of modeling deep ground temperatures in varying snow conditions, when snow conditions are determined by topographical patterns (P19L397).*

I still cannot fully understand this line of reasoning. The seasonal variability driven by the snow cover is very central to the dynamics of the subsurface temperatures above the zero annual amplitude. If these heat fluxes cannot be correctly reproduced in the model, why can the model accurately model the temperatures below the zero annual amplitude?

- *The ERT data are acquired to test the model at varying elevation and aspects, and overcome the limited representativity of the two boreholes in mountain terrain. Our model is able to reproduce the observed temperature patterns. This result is very important because it allows us to trust the model in complex topographical settings, which is fundamental for mountain permafrost mapping (P20L398).*

I understand the importance of using ERT to achieve more spatial ground data. Especially in tough terrain like at your study site. Anyhow, I still think the results of ERT-Measurement should be discussed in more detail. I have addressed these points in the line-by-line comments.

In my opinion, one point that falls into the area of both fundamental and formal criticism is the discussion and conclusions. While chapter 5.1 clearly discusses the model uncertainties (in my opinion, this could be done in more detail), chapter 5.3 gives results on the future development of subsurface temperatures. There is no real discussion. Here I would wish that the results were discussed more clearly according to the objectives, which in my opinion could also be made more concrete. I have a similar feeling about the conclusions. Here there is a focus on the outlook regarding the relationship between permafrost degradation and mass movements. This is a topic that is not addressed in the paper, with the exception of the introduction. It is also not clear to what extent the temperatures below the zero annual amplitude that could be modelled are related to mass movements. Rather, I think it would be of interest to know where the weaknesses and possibilities of the minimalist approach are to be seen. This is an issue that I think is of great importance. And here a clear evaluation of the approaches carried out would take the scientific community much further.

**Line-by-Line Comments**

P1/L4 (…) towards the characterisation of bedrock permafrost (…)

As you changed the title with regard to the characterisation of permafrost, does is still make sense to focus on characterisation here?

P1/L12-13

Large parts of the abstract focus on background and methods. Only the last sentence is on results. Maybe you could give a little more information on results and maybe the fundamental conclusions?

P1/L20-21 (…) and debris permafrost creep rates increase

Maybe change to: (…) and an increase in permafrost creep rates?

P2/L24-26 Sentence

Suggestion: For Greenland, a precise quantification of mountain permafrost distribution is still not available. Models are based on numerical simulations at kilometer scale (Daanen et al., 2011), are not calibrated with in-situ data (Gruber, 2012), or are valid for sedimentary terrain only (Obu et al., 2019).

P2/L28-31 This knowledge gap challenges our understanding (…)

This sentence focuses on the same point as main parts of the first paragraph in the introduction. Maybe add the information and references dealing with hazard assessment and affected population to the first paragraph to avoid redundancies?

P2/L32 "A major challenge when modeling mountain permafrost in this region is due to data availability, as ground temperature data are limited to few low-land sedimentary boreholes that are not representative for higher elevation and complex terrain (Obu et al., 2019)."

Sentence is a little difficult. Suggestion: The fact that ground temperature data in Greenland are limited to a few low-land sedimentary boreholes that are not representative for high mountain bedrock permafrost in complex terrain, is a major challenge for modelling mountain permafrost in this region.

P2/L43-44 "This approach (…)"

Sentence; suggestion: This approach has the advantage that good results are obtained while only basic climate data are needed, i.e. air temperature and solar radiation

P2/L45-54 Paragraph on the influence of snow on permafrost

This paragraph focuses mostly on the spatial distribution of snow and its influence on the ground thermal regime. Anyway, the temporal aspect also has a great effect, with the timing of the onset of snow cover and snowmelt. Can this somehow be included in the model or would it be helpful to add some information here?

P2/L55 (…) aim of this study is to move a first step towards a high resolution regional characterization of mountain permafrost in Greenland

As already asked above, the characterization of permafrost aspect has been removed from the title, as it is not the main focus of the manuscript. I still think it would be nice to go a little bit more into detail regarding the aims of this study, especially with regard to the complex approaches you mention in the following sentences.

P2/L56-58 "To do so, we focus on the Sisimiut area, (68° N on the west coast), where we have a relatively large amount of data. In fall 2020 we installed 28 surface temperature loggers in the area measuring Ground Surface Temperature (GST), covering the local range of elevations and aspects."

To me this sounds a little bit confusing. Were the data available for use or did you install the loggers for the project? Maybe you could rephrase and make clear what data were available and what has been installed?

P3/L81 "Climatically, Sisimiut is located in the low arctic oceanic area, and weather data are recorded at the airport weather station (Cappelen et al., 2021; Cappelen and Jensen, 2021)"

Two sentences? I do not see why the "and" should connect the two parts of the sentence?

P3/L78-80 and P5/L117 "The mountains of the region typically have pyramid-shaped summits and steep rockwalls generating debris slopes underneath. Mountains are dominated by bedrock, although vegetation patches are common at up to 400 m.a.s.l." and "such as flat bedrock (6), soil (11) and easy-access rockwalls (2)"

In the chapter 3.1 you mention that iButtons were installed in soil. Anyhow, you did not mention the soil in the site description. As the thermal regime of soil most probably differs significantly from the

other substrates it would be nice to have some information on the soils, maybe thickness and distribution? Or do you consider it as insignificant?

P3/L89-90 "The recent climatic change on the other hand is believed to have caused significant glacial retreat in the coastal glaciers in the area, which lost about a fourth of their volume in the past three decades (Marcer et al., 2017)."

Maybe you could rephrase the sentence, as there is no "on the one hand" in the paragraph it does not make sense to introduce the other hand. Suggestion: "Within the last three decades coastal glaciers in this area experienced a loss of about a fourth of their volume"?

P4/L115 "These loggers were placed by drilling a 10 x 300 mm hole and sealing (…)"

I suppose you drilled holes for each logger, maybe you could rephrase?

P5/L114-119 Description of logger positions

Is there a reason, why holes in bedrock are 10x300 for geoprecision loggers and 22x100 for the iButtons? Also, why you used different sealing? And why in 50mm in gravel?

P5/L120 "Two 100 m deep boreholes, SIS2019-02 and SIS2021-01, are drilled in bedrock outcrops in relatively flat terrain at 50 and 70 m.a.s.l. (see Fig.1) and similar conditions of exposure to solar radiation, while snow conditions are different.

Maybe two sentences? Suggestion: Two 100 m deep boreholes, SIS2019-02 and SIS2021-01, are drilled in bedrock outcrops in relatively flat terrain at 50 and 70 m.a.s.l. (see Fig.1). While conditions are similar with regard to exposure (exposition?) and solar radiation, snow conditions are different.

P5/L127 "(…) and it was logged manually three times since it was drilled."

Could you be a little bit more specific here? Does this mean these three datasets are part of your study? Is there a general measurement interval for the borehole? Furthermore, in Figure 4 you show 4 datasets measured at that site. Could you give a little background on this?

P6/L136-137 Could you provide a cross-reference to Figure 1, showing the location of the ERT-profile?

P6 and 7/L135-154 Structure of the paragraphs

I think it was really helpful that you added some background information on ERT in this chapter. However, to me, the structure is a little confusing. You start with the setup of the measurement, followed by basics on ERT, back to the field-setup followed by data processing. Could you maybe rearrange this in a more straightforward way?

P6/L141 "(…) yields only qualitative information on the thermal state of materials (…)"

The wording here could be somewhat misleading and imply that one can directly derive information on thermal conditions from ERT. Even if only qualitatively. Here I would find it helpful to formulate the whole thing a little more precisely. Perhaps a clear introduction to the chapter would be helpful, in which it is made clear at the beginning that the information on the thermal status is derived from the ERT measurement in combination with the laboratory calibration.

P7/L157 "The three granite cubic core samples (…)

In chapter 2 you write, that the bedrock consists of amphibolitic gneiss. Is this information correct? If so, could you explain why you used granite for the calibration? And if there is granite and gneiss in the study area, could you take this into account in the site description?

P7/L158-220 In this paragraph values were changed compared to the first version of the manuscript. The changes are tracked, but – far as I can see – without explanation. Changes include porosity, conductivity and information on TF and downward/upward. Could you please explain the reasons for the changes and how they affect the data?

P7/L169 "(..) and freeze-thaw conditions of the ERT transect."

Freeze-thaw indicates a process, I think something like frozen/unfrozen might fit better in this context?

P8/L190 "(…) for cryosphere evolution modelling due to its good performance in the region (Colgan et al., 2016; Hofer et al., 2020) thanks to his good performance in the region (Fettweis et al., 2011)"

Repetition of "good performance". Please rephrase

P9/L205 "In our study, we use a conceptually identical approach, based on the following hypothesis (Magnin et al., 2019): the snow-free GST can be predicted by an empirical model trained using available forcing variables that dominate GST distribution on steep bedrock."

I do not fully understand the citation within this sentence. Does it mean the Hypothesis is to be found in Magnin et al. 2019? If so, maybe rephrase to: …based on the hypothesis given in Magnin et al. 2019, stating that…?

P9/L215-228 - Chapter Snow Cover Modeling

If I understand correctly, the chapter focuses more on the distribution of snow cover, rather than the snow cover in general? Maybe you could rename the chapter?

Furthermore, I must admit, that the chapter was a little hard for me to follow, as you tell the story backwards. You start with temperature offset you calculated that you multiply with SnowP, then explain what SnowP is and what it is based on, then how the SAGA dataset was created and so on. Even though I do not think this approach is wrong, maybe you could think about changing this paragraph and turn the line of argument around?

P10/L235-237 Here you argue, that varying the rock-type parameters results in temperature differences smaller than 0.1 °C. Could you add some information on the values you tested to get a little more context on the errors?

P10/L241 2D "…models, as it not known…"

here is an "is" missing, please change to "…model, as it is not known…"

P11/L250 Could you change 0.015°C to K?

P12/L279 "…we compute 2D model for two location…"

please change to: "…we compute 2D models for two locations"

P12/L281 Please provide reference for QGIS

P12/L281 "For each location we set-up a north-south transect in the QGIS software, and used it to sample the elevation profile from the DEM"

Suggestion: Elevation data along transects from north to south were extracted from the DEM using QGIS (Quantum GIS)

Chapter 4.1 P12/L290-306 In my opinion, the language of the largely newly written chapter 4.1 should be improved. A few suggestions or specific points:

> P12/L290 "GST data are measured during one full year, as loggers were installed in September-October 2020, and retrieved one year later."

> Maybe change to something like: GST data are measured between September 2020 and 2021? Did you retrieve the loggers or do they still measure?

> P12/L291-292 "Fifteen loggers present snow-free GST data (Fig. 3a), seven present thick snow cover and six present intermediate characteristics"

> Repetition of "prevent" maybe change to: Snow-free conditions prevailed at 15 logger sites, a thick snow cover was observed at 3 sites, 6 sites were continuously snow-free

> P12/293 "in general, snow cover onsets in early November and lasts until mid-June, although this depends on the specific logger location"

> "in general, there is a continuous snow cover from early November to mid-June, with topographical differences."

> Please also check the rest of the chapter.

P13/L308 "As shown in Fig.5b, the electrical conductivity tomograms  show  vertical and also lateral variations distribution of the conductivities with low conductivity values (< 10−3.5 Sm−1) below the north and south face and high values inside the mountain (> 10−4.4 Sm−1)."

Please rephrase. Difficult sentence and repetition of conductivity. Where is the difference between "values inside the mountain" and "below north and south face"? Also inside the mountain? Maybe you could provide a more precise description of the area you discuss?

P13/L310 "...from frozen to thawed conditions..."

Thawed implies, that the subsurface has been frozen. Maybe write "from frozen to unfrozen conditions"? as there is no data on the temporal evolution?

Chapter 4.2/Figure 5b It would be helpful if you could add an indication in Figure 5b on where North and South is. I have a question about the lithological fault shown in Figure 5a. Firstly, could you provide additional information on this in the text? Secondly, this lithological fault is located directly in the border area between relatively low and very high resistivities. I could imagine that in the vicinity of fault zones, the rock may be more weathered or fractured, which could potentially have an influence on the resistivity distribution. Partly you discuss this a little in Chapter 5.1 (L401-404) coming to the conclusion, that "a direct comparison between model and geophysics is not meaningful" I think this might need a little more background information.

Figure 4b – could you add a vertical line at 0°C as you did in the modelled-measured graph?? this would make it easier to see when the temperature falls below 0°C and where the zero amplitude is.

Figure 5 b/c – could you add distance values on the x-axis? Could you add a clear Elevation axis in Fig. 5b?

4.3.2 Heat Transfer Model & Figure 4 & Discussion

Could you provide a little more information on the results and also in the discussion of results? You clearly mention the errors of the model, but you do not go into detail. I think what is interesting is that temperatures in the uppermost 10-20 m are massively overestimated, showing positive temperatures while borehole temperatures are negative (it looks like in the shallow subsurface modelled temperatures are around 2 °C, while Data show values of -4 °C, is that correct?) In contrast, Temperatures at greater depths seem to be more underestimated around the errors of around 0.15°C.

In the discussion you mostly point out, that the problem might be the climate data. Could there also be other reasons? Could effects be also driven by parameters of the heat transfer model?

P15/L333 "The permafrost map is represented by the MGT20, i.e. the average temperature at 20 m depth (below the dpeth of zero annual amplitude) during the period 2012-2022 (Fig.7a and b)

Could you please rephrase this sentence? Is it correct, that you defined the MGT20 as permafrost?

Figure 7 "(outer radius is sea level, increasing to 800 masl at the center. "

")" is missing

Is there an exaggeration in the 3D model of Figure 7b? If so, could you add the info?

Chapter 4.4.1 Comparison between 2D model and ERT profile

As you compare and discuss results, would it make sense to make this part of the discussion chapter?

P16/L338 "Snow cover plays an important role, as snow covered areas can increase of 250 m the elevation of the MGT20 0 °C isotherm."

Sentence is difficult to understand. Could you please rephrase?

P16/L342 "In Fig.5c is presented the 2D model simulation at the geophysical profile location."

Suggestion: The 2D model simulation at the geophysical profile location is presented in Fig. 5c

P16/L346 "indicate the presence of sporadic permafrost on the summit"

Could you be a little more precise with locating the frozen ground? At least in the ERT it looks like the direct summit is unfrozen (or in the transition zone?)

P16/L347 "Both datasets indicate a mostly unfrozen south face, and a colder north face"

This information is already given in this paragraph (Lines 344-345)

P16/L347 "However, the ERT data indicate a large unfrozen section at the extremity of the north face."

As mentioned above, I think a little more background on the ERT-dataset would be helpful. Especially with view to the fault zone and potential effects on the resistivity distribution. One problem I see here is, that there are a few redundancies, as the ERT data are already presented in chapter 4.2. In contrast, there is no discussion on the ERT data in chapter 5.

P17/L356-363 General Comment on the use of "MGT20" and parts of the discussion, where you state, that the model is suitable for evaluating ground temperatures below the depth of zero annual amplitude (P20/L403)

This might be a relatively fundamental comment, but could you provide reasons, why you refer to MGT20 as a relevant depth with regard to ground temperatures? I can see, that the reliability of your

model is relatively weak in the area above. Anyhow, if you know, that there is an area, where the temperature does not change, is the critical aspect on the detection of this depth or to model the temperature itself?

With regard to MGT20 could you please add some information on how to understand this parameter? Does this value refer to the depth below the ground surface or vertically? Considering the topographic conditions, it has made it very difficult for me to understand what actually happens with a change in this MGT20 isotherm. Especially since both the introduction and the conclusions focus on the relationship between permafrost degradation and rockfalls, it would be interesting to explain the significance of this MGT20 in this context. Another commonly used measure, for example, is the temperature on top of permafrost (TTOP). Could this approach be more effective than MGT20? Could you maybe provide reference, why this value makes sense in this context?

P17/364 "(Fig.9).For"

Please add a space before For.

P18/Figure 8 (b) Could it be helpful to add the mean Temperatures of SIS2021-01 as recorded in 2021 as a reference?

P18/Figure 8 (c) Please check the readability of all numbers in the figure. The Degree-Values are hard to read

P18/L383 "A more reliable assessment…"

The section starts by stating that reliable data have been collected and ends by stating that the model is not suitable for short-term changes above zero-annual-amplitude. Then it continues with the statement that a higher reliability is achieved for depths below the zero-annual amplitude. Here it would be nice if this contradiction could be derived a bit more clearly.

P18/L384-385 "Although our model seems to provide good results (mean error of 0.14 °C from 0 to 100 m depth, 385 Fig.4) (…)"

I think the discussion could be a little more comprehensive at this point. Reference is made to figure and a mean error of 0.14 °C between 0 and 100 m depth. On the one hand, no information about this 0.14 °C is given in the text or in figure 4. On the other hand, does it make sense at this point to mention the mean error over the entire depth, if the errors between 0-20 m depth are significantly higher with more than 2 °C? In my opinion, these aspects could be considered in the discussion in a little more detail.

P19/L391 "(…) we believe that this performance difference is likely imputable to our climatic database"

As already asked in the first review, it would be interesting to include a little more discussion on the climatic database that has been used. In your reply to my question you wrote "Although this would be an interesting study, it is not covered by the paper's aim." Here I partly do not agree. I think, if a study is conducted, and it has been decided to use a certain database, one basic hypothesis should be, that the database is suitable to answer the research question. With regard to this, I think this aspect should be discussed a little more detailed.

P19/L397-398 "SIS2019-02 has also different snow conditions than SIS2021-01, suggesting that our SnowP map and offset provide an acceptable boundary condition to model long term effect of recurrent snow cover induced by topographical patterns"

I have trouble following the reasoning at this point. Can you please revise this section?

P20/L400-402 "Most of the disagreement between model and data is due to the electrical conductivity anomaly on the north face of the geophysical profile. This anomaly occurs near a large lithological fault, visible in the field."

With regard to Figure 5 I would like to ask if this observation is really correct. It looks like resistivity data indicate "Transition Zone" or "Potentially thawed" in areas where the model indicates negative temperatures. (As there are no distances or North/South included in the Figure, I refer to the area between peak and G-RF. Is this correct?

P20/L404-406 "All this considered, the results indicate that the modeling approach is suitable for evaluating of ground temperatures below the depth of zero annual amplitude in complex terrain. This is achieved with using forcing data available at the Greenland scale, making an significant step towards a comprehensive assessment of mountain permafrost in the region."

Here you provide a conclusion at the end of one discussion chapter. Anyway, I would suggest, to either delete or strongly rephrase this statement with regard to results and possibilities.

P20/L409 "temperatures in rockwalls seems comparable"

Think it should be "seem to be"

P20/Chapter 5.3 Discussion?

In my opinion, this chapter does not really correspond to a discussion. Rather, results are repeated. Also, at the end of the chapter, where Krautblatter et al. and Magnin et al. are cited, nothing is really discussed. In my opinion, this chapter is not really purposeful as long as the only result of the discussion is that climate change leads to permafrost degradation. Also, the reference to rockfall compared to Krautblatter et al. cannot really be derived from the discussion in this chapter.

P21/Figure 10 - I consider the information in this figure to be somewhat misleading. Basically, the comparison is very interesting, but it seems that the data is not really representative for the regions listed. It appears that meta-studies are cited, but as far as I could understand based on the literature mentioned, they are mostly case studies referring to specific locations. While the studies in Norway include one northern and one southern site, for the European Alps there is one study that refers to an area in the Mont Blanc massif. Is this correct? How representative are the studies listed for the regions? Can the graphic be optimized and made more concrete?

P21/L433 "quantification"

What is meant by quantification of bedrock permafrost? Far as I can see, a real quantification is not given within the manuscript?

P21/L434-435 "The modeling approach produces results that are consistent with available data from deep boreholes and geophysical investigations"

In my view, this conclusion is somewhat imprecisely worded. It sounds as if reliable results have been obtained, both in comparison to borehole data and to the geoelectrical measurement, which does not correspond to the statements in chapter 5.1.

P21/L435 One "Permafrozen" is left.

P21/L441-447 This section focuses on the relationship between permafrost degradation and rockfall activity. First, I would note that this is an outlook rather than a conclusion, as there is no data on these points within the manuscript. Secondly, I am not sure how relevant the results of the study are to establish a link between permafrost development and rockfall. Of particular importance to rockfall

activity are the near-surface processes. However, the manuscript primarily cites temperature development at 20 m depth. The connection between permafrost degradation and mass movements is definitely extremely relevant, but I do not see any points in the manuscript that allow conclusions that go beyond the numerous existing publications. It could be made clearer here what is conclusion and what is outlook.

P21/L446 "(…) and crack networks (…)"

This is a point I raised earlier; apparently there is a "lithological fault" in the area of the ERT array. This point is raised but not comprehensively discussed. Can you add a few points in the manuscript that makes a conclusion on this issue a bit clearer? Or that shows how the lithological fault mentioned in the manuscript affects the results?

P21 Conclusions in General

I think some of the main outcomes of the manuscript are missing in the conclusions. As discussed in chapter 5.1, there are some uncertainties regarding the model as well as the prediction of permafrost distribution. If I understood it correctly, one important aspect of the manuscript is, to check, if permafrost distribution and evolution can be predicted using a minimal amount of data. Therefore, maybe an important methodological aspect. These points are not part of the conclusion at all. As mentioned above, one main aspect of the conclusions is on rockfall. A topic, that can not be regarded as an outcome of the manuscript. I would suggest to optimize the conclusions – and to some parts the discussion – to get a more precise focus on these aspects.

P23/L464 Wrong DOI for Allen et al. 2009

---

## Referee Report (RR3)

[referee-annotated manuscript omitted]

---

## Author Response (AR2)

Author's response to reviewers on

**"Modeling present and future bedrock permafrost distribution in the Sisimiut mountain area, West Greenland"**

By Marcer et al., The Cryosphere Discuss.,

https://tc.copernicus.org/preprints/tc-2022-189/

**Contents**

**Note to the referees**

We would like to express our gratitude to both referees for their efforts in reviewing our manuscript and for their valuable feedback. We appreciate the time and effort to help us improve the quality of our study. Throughout the review process, we have considered every comment provided by the referees. As a result, we have made major modifications to our study, in order to improve its overall quality. In the following sections, the reviewers can find a summary of the major changes that will help to navigate through the revised manuscript.

**Structure**

Both reviewers noted issues with the text structure in both rounds of review. In response, we have embraced the suggestion to merge the results and discussion sections. Moreover, we have reorganized the entire text to ensure that sections in the methods align with those in the results and discussion, facilitating smoother navigation for readers. We have also revised and re-arranged all figures to address concerns about excessive jumps between distant figures during our discussion. Furthermore, we have provided a clear statement of our research questions in the introduction and ensured that the conclusions directly respond to these questions.

**Revised methods**

We recognized that our methodology introduces confusion due to the multitude of methods employed. Consequently, we embarked on substantial revisions of our methods embracing all suggestions from the reviewers. Eventually, we run all models from scratch with updated methodologies, which resulted in slightly different outcomes. The methods revised are the following:

1.  **Discarding the snow cover model** - After considering the points raised by the reviewers, we decided that this method generates confusion, without adding meaningful value to the study. **We have now limited our analysis to snow-free steep terrain, specifically rock walls**. This terrain still aligns with our primary interest in the broader context of mass movements. As a result, we now refer to Rock Surface Temperature (RST) instead of Ground Surface Temperature (GST). We continue to utilize deep borehole data from flat terrain, but cautiously. On the one hand, data from SIS2021-01, situated in wind-exposed settings that ensure snow-free conditions, can be used to calibrate our heat transfer model. On the other hand, SIS2019-02, which experiences snow cover during the winter due to drifts, is used to assess how well our model (calibrated for snow-free conditions) applies to snow-covered bedrock. We utilize the results to draw conclusions about the model's uncertainty in managing snow-covered bedrock. The uncertainty range derived from this analysis is integrated into the model outputs, particularly in 2D simulations, where snow accumulation may occur on some sections of the topography. This integration delineates a "transition zone" within the temperature range where the model's uncertainty prevents the clear distinction of permafrost presence or absence.

2.  We discard data from flat terrain, soil and iButtons. Apart from the comment above, it was expressed concern for using data from sensors installed with different techniques. Consequently, we have simplified our study by **exclusively utilizing geoprecision data** to ensure a consistent dataset.

3.  In our permafrost maps, we have **ceased using the MGT20** as doubts were expressed by both reviewers regarding its use. Instead, we rely on the classic Mean RST (MRST), which we average over 20-year time spans for 2002-2022 (current distribution of MRST) and 2080-2100 (projected distribution under the RCP scenarios). It is important to note that MRST values are warmer than MGT20 values, reflecting the ground conditions at depth, where permafrost may exist despite warmer surface temperatures due to ground inertia. Consequently, our results show higher elevations for the 0°C isotherms

4.  In response to R2's suggestion, we have **quantified the uncertainty associated with our weather data** using air temperature data from the AWS in Sisimiut (see new Figure 4). Additionally, we have chosen to use ERA5 data rather than AWS data for forcing the model. This decision allows us to better control the uncertainty on the RST model generated form regional-scale data.

5. We updated the downscaling procedure and now **use the TopoSCALE algorithm** instead of our previous method. This algorithm is the state of the art of weather downscaling in complex terrain and we have started recently to use it in our research group. This decision enhances the clarity of our manuscript, since we have one single RST model to apply to downscaled data, instead of a model for each dataset.

**ERT**

Geophysics has garnered varying opinions from the two reviewers and the editor, ranging from its usefulness to the suggestion that it should be discarded. In this revision round, we continue to support the use of geophysics for several reasons, including the fact that it provides essential data from mountain terrain. We address these concerns as follows:

1. We have clarified in the introduction why we chose to use Electrical Resistivity Tomography (ERT) and highlighted how this method has been employed in a similar manner by other studies addressing similar issues.

2. We improved our methodological description, thanks to the reviewer's suggestions

3. Thanks to merging results and discussion, we have now a more comprehensive description of ERT results and discussion on their value versus their limitation.

4. We have introduced a new figure (figure 7) that quantitatively compares ERT predictions of permafrost with 2D model outputs. This quantification allows us to describe the agreement between the two methods and derive meaningful conclusions from it.

**New RST data**

We now include RST data from the year 2021 – 2022. Now, our observational period for the RST model is two years (2020-2022). This point has been raised few times, especially during the first revision round. We agree that longer time series brings more solidity to our results.

In conclusion, we thank again the reviewers for their work in helping us refine our study. While we have considered and appreciated all style-related comments, some sections have been removed or significantly altered (e.g., regarding snow modeling) to streamline the paper. These comments are not specifically answered in the following sections, but rest assured that no comments have been disregarded.

**Referee #1**

We thank the reviewer for this detailed feedback that helped us to improve the text. All comments where useful to reshape the study and manuscript as described in the "Notes to the referees" section. In addition to the concerns addressed by that section, we add that:

- All general comments regarding structure, figures and units are accepted and integrated.
- All style-related comments are accepted.
- All qualitative descriptions are removed in favor of quantitative descriptions (e.g. comment L352, 356, 425)
- We have restructured the figures and subfigures to make sure that we cite them in ascending order. We now do not jump back to old figures through the text (e.g. comment L351, L421). Thank you for the suggestion; that helped massively in working on the text.

Hereafter, detailed answer to any comment that is not related to the previously described topics.

[R] L 154 - What is the convergence criterion? Maybe provide the value reached after 3 iterations.

[A] We now describe the convergence - L 144

[R] L 157 - So you extracted three core samples from the two rock samples you collected in the field?

[A] This is misleading; we have samples from the surface only. The sentence is now corrected – L 147

[R]L 163 - The internal sensor of what? What is the precision of this sensor (compared to the external one)?

[A] We have now provided an explanation of what these sensors are - L 154

[R] L 166 - Right now, the connection between these three sentences is not clear. I suggest to provide a more detailed description of the steps undertaken in your laboratory experiment. Maybe you could also briefly address how you consider secondary porosity or which assumption you use in this regard, respectively.

[A] We have now improved the description of the steps for the laboratory analysis. L145 to 158. We briefly address the issue on secondary porosity at L319. Thank you for raising the point.

[R] L 285 - I would not write this as an inline equation but instead as a numbered equation (even if it remains the only one in the manuscript). In this way, the different terms/variables in the equation can be properly introduced (e.g., dT is not addressed in the text). Moreover, I strongly suggest to highlight which variables in the equation are scalars or vectors/matrices, respectively.

[A] Since we do not model snow conditions anymore, we discard this equation. The boundary conditions are given by the RST model, shown in figure 5, P5.

[R] L 291 - Figure 3a shows only the time series obtained from a single logger

[A] We now show all loggers time series (new figure 2a, P12).

[R] Which loggers and why where they selected? Random selection? I suggest to incorporate/highlight this comparison in Figure 3.

[A] We have now dedicated figure 2b to this comparison P12.

[R] L 307 - This section somehow breaks the flow or the connections between the other (sub)sections, respectively. Actually, you could consider to present the ERT tomogram as part of M&M since it is not addressed in the Discussion anyway.

[A] Considering the text in the section "Notes to the referees", we have now expanded the discussion on the ERT and connected their result to the temperature data. See L311 to 316 and L324 to 326.

[R] L 311 - Be careful here, based on your petrophysical analysis in the laboratory and your ERT field measurements (single campaign) technically you can only discriminate between frozen and unfrozen conditions. For permafrost we need frozen conditions for at least two consecutive years. And I do not see how this information can be obtained from your data/analysis.

[A] We can discriminate between frozen and unfrozen conditions below the depth of the zero annual amplitude. We therefore expect permafrost existence here. This is now specified in the text (L307), as well as supported by the observations from the temperature data (L325)

[R] L 324 - How do you quantify that it is "optimal"?

[A] This is now described in M&M L220

[R] L 333 - Was this abbreviation introduced somewhere? MGT20 is a result of your GST modeling? If yes, then this should be made clear in the text. Otherwise I would consider that you obtained MGT20 from another data source and this whole sections would belong to M&M.

[A] MGT20 was introduced in M&M at line 272. R2 is also confused by this quantity. We just decided to not use it anymore and use the classic Mean Rock Surface Temperature instead – See the Note to the referees.

[R] L 338 - Please write your figure captions consistently throughout the manuscript. Here you write "on panel a" while in other figure captions you just write "(a)".

[A] We have now make the captions consistent through the manuscript. Thank you for the suggestion

[R] L 341 - This section appears to be disconnected from previous and following sections. Please reconsider the structure of the Results section or the manuscript, respectively.

[A] This is now part of the section "Model testing" P10, L233

[R]L 342 - 2D model simulation of which parameter based on which input data?

[A] This was described in the dedicated section: 2D models L 275. We have now moved this to the "Model testing" section. P10, L233

[R] L384 - In line 330f you wrote "[...] indicating errors up to 2 °C above the depth of zero annual amplitude (20 m depth), while the errors are consistently smaller than 0.15 °C below this depth."

[A] We now refer to figure 6c to describe errors, P17. We are now happy to describe errors across different depth ranges, thank you for this suggestion.

[R] L 388 - Are you referring to the results presented in Figure 5? The agreement between modeled temperatures and observed electrical resistivity needs to be quantified.

[A] Quantification now provided in new figure 7, P18, and L401 to 403

[R] L 403 - This could also be due to the data or processing, inversion (e.g., 2D vs 3D). Also if your geophysical results have such a substantial limitation with regard to the validation of your modeling approach why present them anyway? How is this supported by your geophysical investigations?

[A] The limitations here described are limited to a small area of the profile. The ERT profile is very valuable to our study. Considering the answer above, we have now put more emphasis on how we can interpret the ERT data and how they validate our understanding of permafrost in the area.

[R] L 411 - In Figure 9, I can see this for Nattoralinnguaq but less clearly for Nasaasaaq. How do you explain this?

[A] That's because the north face of Naasaasaq summit can accumulate more snow than the south face (see flat surfaces in old fig 9). We now use profile on the ridge instead of the summit, which is homogenously steep on both sides, and results, are clearer (see fig 10, P21).

[R] L 417 - So what is the dominant factor here? Solar radiation or elevation-dependent air temperature variations? From the current formulation this is not entirely clear.

[A] We have reformulated this concept L321 to 326.

[R] Actually, in the Discussion you state that is a substantial disagreement between modeled temperatures and the observed electrical response.

[A] The agreement between 2D model and ERT data is now in new figure 7, P18, and L401 to 403. An evaluation of the agreement, and why we consider it satisfactory, is at L412

[R] L 448 . To convey [...]

[A] Thank you very much for this. We like the text and included as closing statement in the conclusions, L481.

**Referee #2**

**General comments**

[R] In their revised manuscript "Modelling present and future bedrock permafrost distribution in the Sisimiut mountain area, West Greenland", originally "Characteristics and evolution of bedrock permafrost in the Sisimiut mountain area, West Greenland", the authors use a minimalist approach to modelling spatial permafrost distribution and its future evolution. In my opinion, the revision of the manuscript has massively improved its quality, although some points still remain open. Since the changes were very extensive, a few more questions arose during the re-review of the manuscript, which are addressed in the following. The reviewers' points of criticism regarding formal aspects have been incorporated very well for the most part. Nevertheless, there are some sentences, especially in the newly written sections, whose wording should be revised. I have made some suggestions here, but would recommend that the manuscript be critically proofread once again. The illustrations have also been significantly improved. A few suggestions, or points of criticism, are given below in the line-by-line comments.

[A] Many thanks to R2 for this work. We appreciate the effort and did our best to reshape the study according to the points raised. Please, see "Notes to the referee" for a summary. In short, all points raised are accepted and integrated, bringing to a major revision of the methods and the study.

[R] I don't want to appear cynical at this point, but I want to be deliberately provocative; does this mean that the heat-transfer model works in the depth ranges where heat transfer no longer takes place? That is, below the zero annual amplitude? And, isn't it the area above zero amplitude that is relevant for permafrost dynamics in the coming decades and for potential mass movements? Of course, climatic trends are relevant for long-term permafrost development and not short-term dynamics. Nevertheless, seasonal effects in particular are important for these dynamics. Since a central point of the paper is to test the uses of this minimalist approach, a much more comprehensive evaluation and discussion of these points (including the database on climate and weather data) would be very relevant to me.

[A] Thank you for your thought-provoking comments. Regarding your first question, this study is designed to understand the long-term evolution and zonation of permafrost in the area rather than the short-term dynamics. We acknowledge the importance of seasonal effects in permafrost dynamics, but our focus is on identifying the presence and potential future changes in permafrost distribution. Modeling shallow permafrost dynamics and potential mass movements is indeed a complex and challenging task, and it falls beyond the scope of our study. We've clarified this in the introduction by explicitly stating our research questions and objectives (L65). This is taken up again in the conclusions, where we clearly state that our model is not suitable for shallow dynamics, and other methodologies should be adopted instead (L481). In response to your suggestion for a more comprehensive evaluation of climate and weather data, we have now implemented a validation procedure, which quantifies the accuracy and reliability of our data (See dedicated sections 3.3.1 P7, and 4.3.1 P14, dedicated figure 4).

[R] I still cannot fully understand this line of reasoning. The seasonal variability driven by the snow cover is very central to the dynamics of the subsurface temperatures above the zero annual amplitude. If these heat fluxes cannot be correctly reproduced in the model, why can the model accurately model the temperatures below the zero annual amplitude?

[A] You've rightly pointed out the significance of seasonal variability driven by snow cover in subsurface temperature dynamics above the zero annual amplitude. In our approach, the model includes offsets to account for the overall average effect, which allowed it to approximate temperatures below the zero annual amplitude for long term permafrost dynamics. However, upon further reflection and considering your input, we have decided to revise our methodology and made the decision to limit our analysis to snow-free areas, specifically rock walls. We observe the effect of snow cover on the data from borehole SIS2019-02, and use these data to discuss how snow cover affect bedrock temperatures and generates uncertainty in our model. By doing so, we acknowledge the complexities associated with modeling the snow cover's impact on permafrost dynamics, and we have chosen to focus on areas where the model can provide more reliable and meaningful results.

[R] I understand the importance of using ERT to achieve more spatial ground data. Especially in tough terrain like at your study site. Anyhow, I still think the results of ERT-Measurement should be discussed in more detail. I have addressed these points in the line-by-line comments.

[A] Thank you for your feedback, and we appreciate your emphasis on the importance of discussing the ERT results in more detail. In response to your comments, we have made several improvements to the paper. First, we have updated the ERT description in relation to the observed temperature data (L322 to 326) and the heat transfer model (See L400 to 415). Additionally, we have merged the results and discussion sections to ensure that the information is not scattered across the text, allowing for a more coherent and comprehensive presentation. Furthermore, we have dedicated a new figure (Figure 7, P18) to comparing the ERT data to our model, quantifying the agreement and disagreement between the two models. This addition will provide readers with a clearer understanding of how the ERT measurements align with our modeling results.

[R] In my opinion, one point that falls into the area of both fundamental and formal criticism is the discussion and conclusions. While chapter 5.1 clearly discusses the model uncertainties (in my opinion, this could be done in more detail), chapter 5.3 gives results on the future development of subsurface temperatures. There is no real discussion. Here I would wish that the results were discussed more clearly according to the objectives, which in my opinion could also be made more concrete. I have a similar feeling about the conclusions. Here there is a focus on the outlook regarding the relationship between permafrost degradation and mass movements. This is a topic that is not addressed in the paper, with the exception of the introduction. It is also not clear to what extent the temperatures below the zero annual amplitude that could be modelled are related to mass movements. Rather, I think it would be of interest to know where the weaknesses and possibilities of the minimalist approach are to be seen. This is an issue that I think is of great importance. And here a clear evaluation of the approaches carried out would take the scientific community much further.

[A] We have taken your feedback into account and made the following improvements:

- Regarding model uncertainties, we have provided a more detailed description of these uncertainties in the paper. Specifically, we have included a better explanation of model errors concerning the available data from boreholes and ERT measurements. We define an uncertainty range that accounts for the model's root mean square error (RMSE) in training (See figure 6c, L364 to 367) and the uncertainty introduced by the possibility of snow cover (L384 to 389). This uncertainty is now integrated into the model outputs when describing future scenarios, especially in complex terrain using our 2D models (see L389 to 395). We have also incorporated this uncertainty into the discussion of permafrost evolution in future scenarios (see L431-440, figure 10 P21).

- In the conclusions section, we have revised our focus to emphasize the main outputs of our modeling approach (L465 to 480). We now clearly state the role of our model in describing present and future patterns of permafrost distribution in rock walls within the study area. We have now clarified that the model is not designed to describe processes directly lined to mass movements, and that other modeling approaches are required to make this further step (L482 to 485).

**Line-by-Line comments**

Due to the substantial changes to the text, we do not keep track of style-related comments, as we integrated them when consistent to the new structure. In particular, as explained in the "Note to the referees" section, we decided to discard our snow model, also thanks to the points raised. Therefore, we will not further discuss here this issue. Here, we provide response to content-related comments.

[R] In the chapter 3.1 you mention that iButtons were installed in soil. Anyhow, you did not mention the soil in the site description. As the thermal regime of soil most probably differs significantly from the other substrates it would be nice to have some information on the soils, maybe thickness and distribution? Or do you consider it as insignificant?

Is there a reason, why holes in bedrock are 10x300 for geoprecision loggers and 22x100 for the iButtons? Also, why you used different sealing? And why in 50mm in gravel?

[A] We acknowledge that the combination of loggers and ground conditions is confusing, and we've decided to homogenize the data sources and focus on a more consistent approach in our study. Therefore, for surface temperature data, we will only use data from the geoprecision loggers installed in rock walls in our study. This decision allows us to maintain a more consistent and coherent dataset for analysis. Also, this decision is in line with focus our study on rock walls only, as described above. Regarding the different hole sizes and sealing methods, this was due the practical issues of drilling 30 cm with 22 mm bit, which was not really possible with our driller.

[R] I think it was really helpful that you added some background information on ERT in this chapter. However, to me, the structure is a little confusing. You start with the setup of the measurement, followed by basics on ERT, back to the field-setup followed by data processing. Could you maybe rearrange this in a more straightforward way? […] The wording here could be somewhat misleading and imply that one can directly derive information on thermal conditions from ERT. Even if only qualitatively. Here I would find it helpful to formulate the whole thing a little more precisely. Perhaps a clear introduction to the chapter would be helpful, in which it is made clear at the beginning that the information on the thermal status is derived from the ERT measurement in combination with the laboratory calibration.

[A] Thank you for the comments. We have restructured the whole section accordingly (also considering the comments form R1 who shared similar concerns). We have now a structure consistent with the suggestions. We also focus on describing how the laboratory analysis is used to interpret the profile. We have also discarded some technical information that are not relevant to the analysis and rather refer to previous studies that described in deep detail the method we apply (L128)

[R] In this paragraph values were changed compared to the first version of the manuscript. The changes are tracked, but – far as I can see – without explanation. Changes include porosity, conductivity and information on TF and downward/upward. Could you please explain the reasons for the changes and how they affect the data?

[A] The explanation is given in previous rebuttal. In summary, we repeated the experiment with different time steps for temperature intervals. Time steps and temperature steps are now explained in the methods (L156 to 158).

[R] Here you argue, that varying the rock-type parameters results in temperature differences smaller than 0.1 °C. Could you add some information on the values you tested to get a little more context on the errors?

[A] We have now included the thermal properties in the calibration process and describe how porosity has more influence on the model outputs. We specify that we started the calibration procedure using an initial set of parameters and could not improve the model results by varying them (L360 to 363).

[R] It would be helpful if you could add an indication in Figure 5b on where North and South is. I have a question about the lithological fault shown in Figure 5a. Firstly, could you provide additional information on this in the text? Secondly, this lithological fault is located directly in the border area between relatively low and very high resistivities. I could imagine that in the vicinity of fault zones, the rock may be more weathered or fractured, which could potentially have an influence on the resistivity distribution. Partly you discuss this a little in Chapter 5.1 (L401-404) coming to the conclusion, that "a direct comparison between model and geophysics is not meaningful" I think this might need a little more background information.

[A] We have added a north arrow in the figure, thank you for noticing. We have added a more comprehensive discussion about this part of the tomogram, also including some considerations on the expected permafrost properties at this location versus the ERT. Hopefully, having merged results and discussion, the information about this issue are not scattered across the text anymore. (See L311 to 321)

[R] Could you provide a little more information on the results and also in the discussion of results? You clearly mention the errors of the model, but you do not go into detail. I think what is interesting is that temperatures in the uppermost 10-20 m are massively overestimated, showing positive temperatures while borehole temperatures are negative (it looks like in the shallow subsurface modelled temperatures are around 2 °C, while Data show values of -4 °C, is that correct?) In contrast, Temperatures at greater depths seem to be more underestimated around the errors of around 0.15°C. In the discussion you mostly point out, that the problem might be the climate data. Could there also be other reasons? Could effects be also driven by parameters of the heat transfer model?

[A] We have now included a discussion on why we observe large errors on the uppermost part of the heat transfer model. That is, probably because the model is based on conductivity while, at these depths, advective processes may take place and dominate heat transfer. We base this hypothesis from the discussion from Magnin et al (2017), who observe as well high deviances down to 6 m depth. We also indicate that the model overestimation in SIS2019-02 should be taken with care, as we have 4 measures and only from the same time in the year ( fall – early winter). We suggest that advective processes at this time and location tend to cool the borehole more than pure conduction would allow. (See L381 to 383)

[R] Could you be a little more precise with locating the frozen ground? At least in the ERT it looks like the direct summit is unfrozen (or in the transition zone?)

[A] A more precise description of the areas of the profile is now added in section 4.2, P13. To this purpose, also see new figure 7 P18.

[R] This might be a relatively fundamental comment, but could you provide reasons, why you refer to MGT20 as a relevant depth with regard to ground temperatures? I can see, that the reliability of your model is relatively weak in the area above. Anyhow, if you know, that there is an area, where the temperature does not change, is the critical aspect on the detection of this depth or to model the temperature itself? With regard to MGT20 could you please add some information on how to understand this parameter? Does this value refer to the depth below the ground surface or vertically? Considering the topographic conditions, it has made it very difficult for me to understand what actually happens with a change in this MGT20 isotherm. Especially since both the introduction and the conclusions focus on the relationship between permafrost degradation and rockfalls, it would be interesting to explain the significance of this MGT20 in this context. Another commonly used measure, for example, is the temperature on top of permafrost (TTOP). Could this approach be more effective than MGT20? Could you maybe provide reference, why this value makes sense in this context?

[A] We acknowledge that the MGT20 is not very popular among both reviewers. We therefore now use a more classical indicator: the Mean Rock Surface Temperature (MRST) across a long period (we use 20 years). MRST is conceptually similar to the TTOP parameter (TTOP accounts for different snow/ground classes, which we do not need to since we only have the class "snow free bedrock"). Other studies use the same approach, for example Boeckli at (2013) in the APIM map, or Gruber (2012) in the PZI map. The original reason we chose to use the MGT20 is that it gives a representation of deep ground conditions, as it extracted form transient simulations. However, deep ground conditions are already described by the 2D simulations, without all the inconvenient relative to topography and geometry you describe. Therefore, we discard the MGT20 in favor of the MRST without loosing value of the model output. Reviewers should notice that MRST are "warmer" than MGT20 as they represent surface conditions rather than deep temperatures – reason why the polar plots in figure 8, P19 are different than the previous version of the manuscript.

[R] I think the discussion could be a little more comprehensive at this point. Reference is made to figure and a mean error of 0.14 °C between 0 and 100 m depth. On the one hand, no information about this 0.14 °C is given in the text or in figure 4. On the other hand, does it make sense at this point to mention the mean error over the entire depth, if the errors between 0-20 m depth are significantly higher with more than 2 °C? In my opinion, these aspects could be considered in the discussion in a little more detail.

[A] To answer this comment we added figure 6c, P17. We now present model RMSE across the borehole depth. RMSE values are derived for different time aggregations, i.e. for the raw monthly model output, and as an average across the whole observational period of SIS2021-01. The discussion now refers to this figure when addressing the model errors (L364 to 367).

[R] As already asked in the first review, it would be interesting to include a little more discussion on the climatic database that has been used. In your reply to my question you wrote "Although this would be an interesting study, it is not covered by the paper's aim." Here I partly do not agree. I think, if a study is conducted, and it has been decided to use a certain database, one basic hypothesis should be, that the database is suitable to answer the research question. With regard to this, I think this aspect should be discussed a little more detailed.

[A] We embrace the comment and have now reshaped the study accordingly. We now provide a comparison between the different weather datasets and the air temperature measured by the Sisimiut weather station (see dedicated figure 4, and sections 3.3.1, 4.3.1). We have also decided to use AWS data as validation data only, i.e. we do not force the model with them. We decided to take this path as, in our future work, we will need to model permafrost distribution in areas where we do not have such long weather station data (AWS data are sparse in Greenland). We therefore prefer keeping this dataset for controlling what is the uncertainty generated by weather data available at the regional scale. This is now explained (L168 to 174)

[R] I consider the information in this figure to be somewhat misleading. Basically, the comparison is very interesting, but it seems that the data is not really representative for the regions listed. It appears that meta-studies are cited, but as far as I could understand based on the literature mentioned, they are mostly case studies referring to specific locations. While the studies in Norway include one northern and one southern site, for the European Alps there is one study that refers to an area in the Mont Blanc massif. Is this correct? How representative are the studies listed for the regions? Can the graphic be optimized and made more concrete?

[A] Based on the comment, we decided to discard this figure. Comparison with previous studies on this matter can still be found in the text (L289 to 296)

[R] I think some of the main outcomes of the manuscript are missing in the conclusions. As discussed in chapter 5.1, there are some uncertainties regarding the model as well as the prediction of permafrost distribution. If I understood it correctly, one important aspect of the manuscript is, to check, if permafrost distribution and evolution can be predicted using a minimal amount of data. Therefore, maybe an important methodological aspect. These points are not part of the conclusion at all. As mentioned above, one main aspect of the conclusions is on rockfall. A topic, that can not be regarded as an outcome of the manuscript. I would suggest to optimize the conclusions – and to some parts the discussion – to get a more precise focus on these aspects

[A] We agree with the comment. We have now revised the conclusion and listed the main outcomes of the study. As explained above, we make clear that slope stability is just the frame of the project and we do not aim to model processes directly connected. Therefore, this matter is only briefly presented as an outlook in the last sentences of the conclusion (L481 to 488)

---

## Author Response (AR3)

Author's response to reviewers on

**" Modeling present and future rock wall permafrost distribution in the Sisimiut mountain area, West Greenland"**

by Marcer et al., The Cryosphere Discuss.,

https://tc.copernicus.org/preprints/tc-2022-189/

**Contents**

**Editor**

Dear authors,

I am sorry for the long time required for my evaluation, but I was tracking all changes from the original version of the paper submitted to the last revision and this consumed more time than expected. The manuscript has considerably improved from the first version submitted, as also pointed out by the external reviewer. I agree with such evaluation and I decide to accept the manuscript for publication.

I am still requesting minor revisions, to give you the opportunity to include the suggestions provided by the external referee and those from my side indicated in the attached PDF. Please consider including those changes and use this opportunity to read the manuscript to check the English along the paper and improve formulations. This has also been recommended by the external reviewer.

Best regards,

Adrian Flores Orozco

Dear Editor,

Thank you for your thorough evaluation and valuable feedback. We appreciate the acceptance of our manuscript. We implement the suggested revisions and carefully review the manuscript for language improvements, as recommended. We are grateful for your time and consideration.

**Point-by-point reply to editor comments**

[6] 1.1.2020-31.12.2022?

The period depends on the logger, we change the text as *"..the period September 2020 - September 2022.."*

[15] Influence of snowpack has not been addressed

Influence of snow is discussed and assessed at 391

[42] meteorological

Agreed, Text changed accordingly

[47] resistivity

Agreed, Text changed accordingly

[56] ground surface temperature

Agreed, Text changed accordingly

[83] are you instigating many or just one? It seems only one …

There are several rock glaciers in the region. Only one is within the boundaries of the figure.

[87] rock glacier

Agreed, Text changed accordingly

[110] could you provide formation about the position of those boreholes, their distance to the study area?

Boreholes location is shown in Figure 1. In addition, we added: "*within the town's urban area*" L111.

[128] by Duvillard et a. (2020), consisting

Agreed, text changed accordingly

[138] could you describe what is the minimal and maximum levels? also briefly explain this configuration for the audience in TC?

Agreed, text updated to:

*"For the data collection, we use the Wenner configuration. This configuration corresponds to have the voltage electrodes M and N in between the current electrodes A and B with an equal spacing between the electrodes. This array is characterized by an excellent signal-to-noise ratio (Dahlin and Zhou, 2004; Kneisel, 2006)."* L139-141

[141] More details on the data processing, in particular the filtering of the data are needed.

Agreed, text updated to:

*"We cleaned 4% of the measures acquired before the inversion (549 measures acquired, 528 inverted) by filtering out the outliers and the data characterized by high standard deviations (higher than 10%) from the pseudo section and the apparent negative resistivity."* L143-145

[144] Why 10% as the threshold value? How many iterations were needed? Could you also explain what is the RMSE? - it is strange to have an acronym that is not introduced before, also how this parameter is actually computed.

We conducted 5 iterations but opted for the 3rd iteration, as the model converges with an RMS close to 10% by the third iteration. Iterations 4 and 5 tend to overly smooth the model. Text is updated to:

*"The inversion was stopped when the convergence criterion is reached. In this study, the convergence criterion is met when the change in the Root-Mean-Square Error (RMSE) between two iterations is below 10% (default criterion in RES2DINV). In the present case, convergence is reached at the third iteration.*

[156] provide information on the temperature steps defined and the time that each of these steps was taking

The temperature steps are visible at figure 3b, reference now added to the text (L162). Time spend at each step is already in text (L161).

[179] downscaled

Agreed, text changed accordingly

[215] from

Agreed, text changed accordingly

[246] allows
Agreed, text changed accordingly

[263] .
Agreed, text changed accordingly

[318] The role of the fracture needs to be better explained. Also i recommend to elaborate more on the possible factors affecting the resolution of the resistivity image.
Agreed, text updated to:

*"We suggest that the ERT data at this location are influenced non only by bedrock temperature but also by weathering (resulting from the formation of kaolinite, see Richard et al., 2010) and fracturing. We have assume that the rock is isotropic and that the laboratory measurements are representative of the scale investigated in the ERT tomogram (sensitivity close to the electrode spacing close to the ground surface)."*

We add the following reference:

*Richards K., A. Revil, A. Jardani, F. Henderson, M. Batzle, and A. Haas, Pattern of shallow ground water flow at Mount Princeton Hot Springs, Colorado, using geoelectrical methods, Journal of Volcanology and Geothermal Research, 198, 217-232, 2010.*

[320] improve English
Text changed to *"Overall, we consider…"* L327

[330] indicates an
Agreed, text changed accordingly

[365]Shown
Agreed, text changed accordingly

[390] Check English
Sentence is simplified to *"This indicates that, when there is snow cover, our model registers colder temperatures compared to the actual deep rock temperatures."* L398-399

[395]from
Agreed, text changed accordingly

[462]an
Agreed, text changed accordingly

**Referee #1**

I have carefully reviewed the revised manuscript titled "Modeling present and future rock wall permafrost distribution in the Sisimiut mountain area, West Greenland" submitted by Marcer et al. I would like to acknowledge that the authors have diligently addressed all the comments and concerns I raised during the review of the manuscript. The revisions made by the authors have substantially improved the overall quality and clarity of the manuscript. However, I would like to bring to your attention a few remaining minor issues that need correction before the manuscript can be considered for publication. Firstly, there are still some typographical errors and technical issues present in the manuscript that require attention. I recommend a thorough proofreading to ensure the text is free from any inadvertent mistakes. Additionally, I have observed an imprecise interpretation and description of the ERT imaging results, which needs to be addressed. I have attached a PDF document containing my comments for your reference and further consideration. I believe that once these minor issues are addressed, the manuscript will be ready for publication in The Cryosphere and am confident that the final corrections will further enhance the overall quality of the manuscript.

Thank you for your detailed review and constructive feedback on our manuscript. We appreciate your acknowledgment of the improvements made in response to your earlier comments. We address the remaining typographical errors, technical issues, and the imprecise interpretation of the ERT imaging results as per your suggestions – see attached marked-up manuscript and point-by-point reply to the comments.

Your insights have been fundamental in enhancing the quality and clarity of our work. Thank you for your continued support through this process.

**Point-by-point reply to R1 comments**

[105] Maybe merge these two sentences.
The second sentence is not bringing additional information and has been removed

[175] Duvillard et al. (2020)
Agreed, text changed accordingly

[176] We conducted the ERT measurements in ... ?
Agreed, text changed accordingly

[375] This reference might not be necessary as QGis is an established software?
Although we agree, this was requested by R2 in the previous round. We let the editor decide whether it is requested by the journal.

[384] These maps?
Agreed, text changed accordingly

[394] Remove?
Agreed, text changed accordingly

[429] MRST?
I think MARST was not introduced?

Agreed, text changed accordingly

[486] throughout?
Agreed, text changed accordingly

[490] Maybe rephrase to avoid this double usage of "consider"?
Agreed, text changed accordingly

[531] Maybe you could provide here a short statement why you prefer the regional evaluation?
Rephrased: *"While it is possible in principle to utilize weather station data to drive our model and enhance its performance, our preference is to evaluate the model uncertainties using data available for the whole Greenland. This provides an estimation of model performance consistent with the long-term goal to employ this approach for regional-scale use, i.e. in areas where weather station data may not be available."* L359-362

[546] shown in
Agreed, text changed accordingly

[547] consistently lower than? To avoid using "below" twice in this sentence.
Agreed, text changed accordingly

[579] a
Agreed, text changed accordingly

[583] Something is missing here. Maybe ", which is the reason ..."?
Text updated according to Editor comment *"This indicates that, when there is snow cover, our model registers colder temperatures compared to the actual deep rock temperatures. This temperature range describes our uncertainty range when predicting rock permafrost conditions in areas where snow may or may not accumulate, i.e. generic bedrock terrain."* L398-401

[600] "than indicated by the ERT imaging result" bc the ERT does not predict subsurface thermal condition but solves for the electrical properties in the subsurface, which are influenced by the thermal conditions.
Agreed, text changed accordingly

[601] The ERT does not estimate the permafrost extent. The interpretation of the electrical images can overestimate the permafrost extent.
Agreed, text changed accordingly

[635] the interpretation of the ERT imaging result.
Agreed, text changed accordingly

[648] to the
Agreed, text changed accordingly

[743] present
Agreed, text changed accordingly

[745] ERT investigations?
Agreed, text changed accordingly

[759] affect
Agreed, text changed accordingly

---

## Author Response (AR4)

Author's response to reviewers on

**" Modeling present and future rock wall permafrost distribution in the Sisimiut mountain area, West Greenland"**

by Marcer et al., The Cryosphere Discuss.,

https://tc.copernicus.org/preprints/tc-2022-189/

**Contents**

**Editor's comments**

Dear authors,

I read your submitted version of the manuscript. I find that the MS is getting in better shape. However, i still found some aspects in the reading that need to be addressed before finally accepting the manuscript for publication. As suggested before, I strongly recommend you to check the formulations and wording. I also recommend you to avoid the use of acronym before introducing them as the audience of TC is quite broad, and you are dealing with data sets from different disciplines. Please find enclosed a marked file with some suggestions. Please feel free to ignore those that you consider necessary, make your own editions on the marked lines and add further changes.

Best regards,

Adrian Flores Orozco

Dear Editor,

Thank you for your revision and feedback. We have updated the text accordingly. Also, as suggested, we did a number of minor edits to improve readability and wording, as highlighted in the track changes version of the manuscript.

Kindly

Marco Marcer

L1: is not rather the thwing or degradation of the rock wall permafrost? the permafrost itself is not responsible
Sentence changed to:

*"Degrading rock wall permafrost was found responsible for the increase of rock fall and landslide activity in several cold mountain regions across the globe."* L1

L4: we aim at at giving a first characterization of rock wall permafrsot towards a better understanding....? the paper deals with temperature modeling, that should be the objective/aim \(described in this sentence\)....
Sentence changed and merged with the next one as following:

*"In this study, we aim to make a first step towards a better understanding of rock wall permafrost in Greenland by modelling rock wall temperatures in the mountain area around the town of Sisimiut – 68° N on the West Coast."* L4

L26: and are valird... or are valid only...
Sentence changed to:

*"…or are valid only for sedimentary terrain…"* L25

L41: both approaches, or modeling appproaches...
Sentence changed to:

*"Both approaches.."* L39

L47: no capital letters
Agreed. All other acronym definitions are modified accordingly

L51: data or results?
Sentence changed to:

*"The ERT data allow to interpret the bedrock conditions as frozen/unfrozen,.."* L50

L53: it does not develop!! provides a model of the change sin the electrical resistivity of the subsurface it can also be a 3D model!!!!
Sentence changed to:

*"In particular, this methodology provides a model of ground freezing conditions at a given survey date, which can validate numerical simulations"* L52

L54: what about monitoring ERT? there are plenty of studies about this!
This sentence does not dismiss the existence of monitoring ERT. We do not use monitoring ERT in this study, therefore we do not develop on such methodology.

L57: To reach our objective...
Agreed

L56: the objective of this study is to understand the distribution of.....
The aim of this study is as stated. We do not aim to study permafrost in the whole of Greenland.

L60: material and methods
Removed (there was repetition)

L62: material and methods
Agreed

L63: obtained from teo boreholes \(each one with a maximum depth of 100 m\)
Sentence changed to:

*"We calibrate and test our model with temperature data obtained from two boreholes, each drilled to a depth of 100 m in lowland flat bedrock."* L60

L64: material and methods
Agreed

L65: strange formulation - you gain this with the boreholes right? you mean to extend the brehole information to different ares and gain information about subsurface heterogeneities?
you mean here you used the ERT results to obtain the distribution of temperatures, right? the approch qith references belong to the material and methods
Sentence is changed to:

*"The model is then used to generate rock wall temperatures at high elevation, which we compare to ERT data acquired on the field."* L61

L66: your data set from ERT? or from ? maybe to be more explicit?
Sentence is changed, as above

L67: should not be question 1?
Agreed. Conclusion bullet points are changed accordingly

L68: has this been introduced?
Changed to *"climatic"* L65

L69: is this a conclusion? why this after the research quetions?

Removed

L85: this is positive?
No, it's a negative temperature. See text. The automatic latex formatting here is unfortunate but the issue will be resolved when reformatting in the journal style.

L86: has it been introduced?
Yes, at line 60.

L86: has it been intrduced?
That is the name of the instrument. We now introduce it here but let the editor judge if this is in line with the journal style.

L97: intrduce PRT
Agree

L100: if PRT is introduced before, it would be easier for the readers
Agree

L126: introduce acronym
Agree

L131: I would move these lines to start the paragraph and then the sentence with the reference to the petrophysical model
The previous sentence is introductory explaining the generic method, which is in two steps. Here we describe the first step. We would like to keep such structure.

L164: soil temperature modeling? modeling could be anything - also from geophysica data
Changed to *"Rock temperature modeling"* L160

L195: is applicable or has been appplied in...
Agreed

L201: maybe you want to avoid acronyms in section tittles?
Agreed

L239: imaging results/ trasects?
Agreed

L256: when was the first analysis? maybe rephrasing or delete this and jpiny wwith previous paragraph to keep it continuos.
This whole section describes three products to describe permafrost distribution and evolution: (i) RST maps, (ii) numerical model and (iii) 2D model. These three products are given one paragraph each as they

are provided through different methodologies rather than consequential. Therefore, we would like to keep them as such, instead of merging them.

L262: evaluate?
Agreed

L266: above you cna add the section in parenthesis
Agreed

L375: maybe there is a way to avoid so many "below" and make a better reading?
Yes, sentence changed to:

*"When assessing the RMSE throughout the measurement period, we observed values ranging from a maximum of 0.70 °C at the surface to under 0.10 °C at depths less than 10 m.b.g.s., and further decreasing to under 0.01 °C below 80 m.b.g.s. "* L372

L412: in particular to what?
*"In particular"* refers to the previous sentence, where we introduce discrepancy between ERT and numerical model. In this section we enter the details of this discrepancy. We re-arrange the paragraph structure to make this clearer

L419: you mean not correctly resolved by the ERT or maybe just affecting the ERT results?
Affecting the ERT results, specified now in the text. L416

L420: redundant with lines 410 either there or here needs to be deleted
This has been rephrased to better convey main message:

*"Despite these local differences, the two methods agree on the general pattern of permafrost distribution, as they both indicate discontinuous permafrost across the mountain and a dominance of the SSRD in discerning between frozen and unfrozen conditions."* L418

L444: Hence, or Accordingly, our results reveals that...???
Agreed

L449: ? is this the correct word?
Changed to *"pattern"*

L470: geophysical investigations based on ERT?
Agreed